# Momentum Tracking: Momentum Acceleration for Decentralized Deep Learning on Heterogeneous Data

**Yuki Takezawa**                                                      *yuki-takezawa@ml.ist.i.kyoto-u.ac.jp*
*Kyoto University, Okinawa Institute of Science and Technology*

**Han Bao**                                                                       *bao@i.kyoto-u.ac.jp*
*Kyoto University, Okinawa Institute of Science and Technology*

**Kenta Niwa**                                                            *kenta.niwa.bk@hco.ntt.co.jp*
*NTT Communication Science Laboratories*

**Ryoma Sato**                                                          *r.sato@ml.ist.i.kyoto-u.ac.jp*
*Kyoto University, Okinawa Institute of Science and Technology*

**Makoto Yamada**                                                       *makoto.yamada@oist.jp*
*Okinawa Institute of Science and Technology*

**Reviewed on OpenReview:** *https://openreview.net/forum?id=8koy8QuTZD*

## Abstract

SGD with momentum is one of the key components for improving the performance of neural networks. For decentralized learning, a straightforward approach using momentum is Distributed SGD (DSGD) with momentum (DSGDm). However, DSGDm performs worse than DSGD when the data distributions are statistically heterogeneous. Recently, several studies have addressed this issue and proposed methods with momentum that are more robust to data heterogeneity than DSGDm, although their convergence rates remain dependent on data heterogeneity and deteriorate when the data distributions are heterogeneous. In this study, we propose Momentum Tracking, which is a method with momentum whose convergence rate is proven to be independent of data heterogeneity. More specifically, we analyze the convergence rate of Momentum Tracking in the setting where the objective function is non-convex and the stochastic gradient is used. Then, we identify that it is independent of data heterogeneity for any momentum coefficient $\beta \in [0, 1)$. Through experiments, we demonstrate that Momentum Tracking is more robust to data heterogeneity than the existing decentralized learning methods with momentum and can consistently outperform these existing methods when the data distributions are heterogeneous.

## 1 Introduction

Neural networks have achieved remarkable success in various fields such as image processing (Simonyan & Zisserman, 2015; Chen et al., 2020) and natural language processing (Devlin et al., 2019). To train neural networks, we need to collect large amounts of training data, but it is often difficult to collect large amounts of data such as medical images on one server because of privacy concerns. In such scenarios, decentralized learning has attracted significant attention because it allows us to train neural networks without aggregating all the data onto one server. Recently, decentralized learning has been studied from various perspectives, including data heterogeneity (Tang et al., 2018b; Esfandiari et al., 2021), communication compression (Tang et al., 2018a; Lu & De Sa, 2020; Liu et al., 2021; Takezawa et al., 2022a), and network topologies (Ying et al., 2021; Le Bars et al., 2023).

One of the key components for improving the performance of neural networks is SGD with momentum (SGDm). Whereas SGD updates the model parameters using a stochastic gradient, SGDm updates the model parameters using the moving average of the stochastic gradient, which is called the momentum. Because SGDm can accelerate convergence and improve generalization performance, SGDm has become an indispensable tool, enabling neural networks to achieve high accuracy (He et al., 2016; Cutkosky & Mehta, 2020; Karimireddy et al., 2021; Defazio, 2021). Recently, SGDm has been improved in many studies, and methods such as Adam (Kingma & Ba, 2015) and RAdam (Liu et al., 2020a) have been proposed.

In decentralized learning, the straightforward approach to using the momentum is Distributed SGD (DSGD) with momentum (DSGDm) (Gao & Huang, 2020). When the data distributions held by each node (i.e., the server) are statistically homogeneous, DSGDm works well and can improve the performance as well as SGDm (Lin et al., 2021). However, in real-world decentralized learning settings, the data distributions may be heterogeneous (Hsieh et al., 2020). In such cases, DSGDm performs worse than DSGD (i.e., without momentum) (Yuan et al., 2021).

This is because, when the data distributions are heterogeneous and we use the momentum instead of the stochastic gradient, each model parameter is updated in further different directions and drifts away more easily. As a result, the convergence rate of DSGDm falls below that of DSGD. To address this issue, Lin et al. (2021) and Yuan et al. (2021) modified the update rules of the momentum in DSGDm and proposed methods that are more robust to data heterogeneity than DSGDm. However, their convergence rates remain dependent on data heterogeneity, and our experiments revealed that their performance are degraded when the data distributions are strongly heterogeneous (Sec. 4).

Data heterogeneity for decentralized learning has been well studied from both experimental and theoretical perspectives (Hsieh et al., 2020; Koloskova et al., 2020). Subsequently, many methods including Gradient Tracking (Lorenzo & Scutari, 2016; Nedić et al., 2017) have been proposed and it has been shown that their convergence rates do not depend on data heterogeneity (Tang et al., 2018b; Vogels et al., 2021; Koloskova et al., 2021). However, these studies considered only the case where the momentum was not used, and it remains unclear whether these methods are robust to data heterogeneity when the momentum is applied.

In the convex optimization literature, Xin & Khan (2020) and Carnevale et al. (2022) proposed combining Gradient Tracking with momentum or Adam and analyzed the convergence rates. However, they considered only the case where the objective function is strongly convex and the full gradient is used, which does not hold in the standard deep learning setting, where the objective function is non-convex and only the stochastic gradient is accessible. Hence, their convergence rates are still unknown in the setting where the objective function is non-convex and the stochastic gradient is used and it remains unclear whether their convergence rates are independent of data heterogeneity. Furthermore, they did not discuss data heterogeneity, either theoretically or experimentally.

In this work, we propose a decentralized learning method with momentum, which we call **Momentum Tracking**, whose convergence rate is proven to be independent of data heterogeneity in the setting where the objective function is non-convex and the stochastic gradient is used. More specifically, we identify that the convergence rate of Momentum Tracking is independent of data heterogeneity for any momentum coefficient $\beta \in [0, 1)$. In Table 1, we compare the convergence rate of Momentum Tracking with those of existing methods. To the best of our knowledge, Momentum Tracking is the first decentralized learning method with momentum whose convergence rate has been proven to be independent of data heterogeneity in the setting where the objective function is non-convex and the stochastic gradient is used. Experimentally, we demonstrate that Momentum Tracking is more robust to data heterogeneity than the existing decentralized learning methods with momentum and can consistently outperform these existing methods when the data distributions are heterogeneous.

Table 1: Comparison of the convergence rates. In the "Data-Heterogeneity" column, "✓" indicates that the convergence rate is independent of data heterogeneity, and "(✓)" indicates that it is independent, but there is no discussion about data heterogeneity either theoretically or experimentally. In the "Momentum," "Stochastic," and "Non-Convex" columns, "✓" respectively indicates that the method is accelerated using momentum, the convergence rate is provided when the stochastic gradient is used, and the convergence rate is provided when the objective function is non-convex.

| | Data-Heterogeneity | Momentum | Stochastic | Non-Convex |
|---|---|---|---|---|
| DSGD (Lian et al., 2017) | | | ✓ | ✓ |
| Gradient Tracking (Koloskova et al., 2021) | ✓ | | ✓ | ✓ |
| DSGDm (Gao & Huang, 2020) | | ✓ | ✓ | ✓ |
| QG-DSGDm (Lin et al., 2021) | | ✓ | ✓ | ✓ |
| DecentLaM (Yuan et al., 2021) | | ✓ | ✓ | ✓ |
| ABm (Xin & Khan, 2020) | (✓) | ✓ | | |
| GTAdam (Carnevale et al., 2022) | (✓) | ✓ | | |
| **Momentum Tracking (our work)** | ✓ | ✓ | ✓ | ✓ |

## 2 Preliminaries and Related Work

### 2.1 Decentralized Learning

Let $G = (V, E)$ be an undirected graph that represents the underlying network topology, where $V$ denotes the set of nodes and $E$ denotes the set of edges. Let $N \coloneqq |V|$ be the number of nodes, and we label each node in $V$ by a set of integers $\{1, 2, \cdots, N\}$ for simplicity. We define $\mathcal{N}_i \coloneqq \{j \in V \mid (i, j) \in E\}$ as the set of neighbor nodes of node $i$ and define $\mathcal{N}_i^+ \coloneqq \mathcal{N}_i \cup \{i\}$. In decentralized learning, node $i$ has a local data distribution $\mathcal{D}_i$ and local objective function $f_i : \mathbb{R}^d \to \mathbb{R}$, and can communicate with node $j$ if and only if $(i, j) \in E$. Then, decentralized learning aims to minimize the average of the local objective functions as follows:

$$\min_{\boldsymbol{x} \in \mathbb{R}^d} \left[ f(\boldsymbol{x}) \coloneqq \frac{1}{N} \sum_{i=1}^{N} f_i(\boldsymbol{x}) \right], f_i(\boldsymbol{x}) \coloneqq \mathbb{E}_{\xi_i \sim \mathcal{D}_i} [F_i(\boldsymbol{x}; \xi_i)],$$

where $\boldsymbol{x}$ is the model parameter, $\xi_i$ is the data sample that follows $\mathcal{D}_i$, and local objective function $f_i(\boldsymbol{x})$ is defined as the expectation of $F_i(\boldsymbol{x}; \xi_i)$ over data sample $\xi_i$. In the following, $\nabla F_i(\boldsymbol{x}; \xi_i)$ and $\nabla f_i(\boldsymbol{x}) \coloneqq \mathbb{E}_{\xi_i \sim \mathcal{D}_i}[\nabla F_i(\boldsymbol{x}; \xi_i)]$ denote the stochastic and full gradient respectively.

Distributed SGD (DSGD) (Lian et al., 2017) is one of the most well-known algorithms for decentralized learning. Formally, the update rules of DSGD are defined as follows:

$$\boldsymbol{x}_i^{(r+1)} = \sum_{j \in \mathcal{N}_i^+} W_{ij} \left( \boldsymbol{x}_j^{(r)} - \eta \nabla F_j(\boldsymbol{x}_j^{(r)}; \xi_j^{(r)}) \right), \tag{1}$$

where $\eta > 0$ is the step size and $W_{ij} \in [0, 1]$ is the weight of edge $(i, j)$. Let $\boldsymbol{W} \in [0, 1]^{N \times N}$ be the matrix whose $(i, j)$-element is $W_{ij}$ if $(i, j) \in E$ and 0 otherwise. In general, a mixing matrix is used for $\boldsymbol{W}$ (i.e., $\boldsymbol{W} = \boldsymbol{W}^\top$, $\boldsymbol{W}\mathbf{1} = \mathbf{1}$, and $\boldsymbol{W}^\top \mathbf{1} = \mathbf{1}$). Lian et al. (2018) extended DSGD in the case where each node communicates asynchronously and analyzed the convergence rate. Koloskova et al. (2020) analyzed the convergence rate of DSGD when the network topology changes over time. These results revealed that the convergence rate of DSGD deteriorates and the performance is degraded when the data distributions held by each node are statistically heterogeneous. This is because the local gradients $\nabla f_i$ are different across nodes and each model parameter $\boldsymbol{x}_i$ tends to drift away when the data distributions are heterogeneous. To address this issue, $D^2$ (Tang et al., 2018b), Gradient Tracking (Lorenzo & Scutari, 2016; Nedić et al., 2017), and primal-dual algorithms (Niwa et al., 2020; 2021; Takezawa et al., 2022b) were proposed to correct the local gradient $\nabla f_i$ to the global gradient $\nabla f$. As a different approach, Vogels et al. (2021) proposed a novel averaging method to prevent each model parameter $\boldsymbol{x}_i$ from drifting away. It has been shown that the convergence rates of these methods do not depend on data heterogeneity and do not deteriorate, even when the data distributions are statistically heterogeneous. However, these methods do not consider the case in which momentum is used.

## 2.2 Momentum

The methods with momentum were originally proposed by Polyak (1964), and SGD with momentum (SGDm) has achieved successful results in training neural networks (Simonyan & Zisserman, 2015; He et al., 2016; Wang et al., 2020b). In decentralized learning, a straightforward approach to using the momentum is DSGD with momentum (DSGDm) (Gao & Huang, 2020). The update rules of DSGDm are defined as follows:

$$\boldsymbol{u}_i^{(r+1)} = \beta \boldsymbol{u}_i^{(r)} + \nabla F_i(\boldsymbol{x}_i^{(r)}; \xi_i^{(r)}), \tag{2}$$

$$\boldsymbol{x}_i^{(r+1)} = \sum_{j \in \mathcal{N}_i^+} W_{ij} \left( \boldsymbol{x}_j^{(r)} - \eta \boldsymbol{u}_j^{(r+1)} \right), \tag{3}$$

where $\boldsymbol{u}_i$ is the local momentum of node $i$ and $\beta \in [0, 1)$ is a momentum coefficient. In addition, several variants of DSGDm were studied by Yu et al. (2019); Assran et al. (2019); Wang et al. (2020a); Singh et al. (2021). When the data distributions held by each node are statistically homogeneous, DSGDm works well and can improve accuracy as well as SGDm. However, when the data distributions are statistically heterogeneous, DSGDm leads to poorer performance than DSGD. This is because when the data distributions held by each node are statistically heterogeneous (i.e., $\nabla f_i$ varies significantly across nodes), the difference in the updated value of the model parameter across the nodes (i.e., $\eta \boldsymbol{u}_i$) is amplified by the momentum (Lin et al., 2021).

To address this issue, Yuan et al. (2021) and Lin et al. (2021) proposed methods to modify the update rules of the momentum in DSGDm such that the momentum of each node has close values, which are called DecentLaM and QG-DSGDm, respectively. They further experimentally demonstrated that these methods are more robust to data heterogeneity than DSGDm. However, their convergence rates have been shown to still depend on data heterogeneity and deteriorate when the data distributions are heterogeneous.

## 2.3 Gradient Tracking

One of the most well-known methods whose convergence rate does not depend on data heterogeneity is Gradient Tracking (Lorenzo & Scutari, 2016). Whereas DSGD exchanges only the model parameter $\boldsymbol{x}_i$, Gradient Tracking exchanges the model parameter $\boldsymbol{x}_i$ and local (stochastic) gradient $\nabla f_i$ and then updates the model parameters while estimating global gradient $\nabla f$. Nedić et al. (2017) and Qu & Li (2018) analyzed the convergence rate of Gradient Tracking when the objective function is (strongly) convex and the full gradient is used. Pu & Nedic (2021) analyzed the convergence rate when the objective function is strongly convex and the stochastic gradient is used. Recently, Koloskova et al. (2021) analyzed the convergence rates of Gradient Tracking in the setting where the objective function is non-convex and the stochastic gradient is used. There is also a line of research to combine Gradient Tracking with variance reduction methods (Xin et al., 2022) and communication compression methods (Zhao et al., 2022). They showed that the convergence rate of Gradient Tracking does not depend on data heterogeneity. However, these studies only consider the case without momentum, and the convergence analysis for Gradient Tracking with momentum has not been explored thus far in the aforementioned studies.

# 3 Proposed Method

In this section, we propose **Momentum Tracking**, which is a decentralized learning method with momentum whose convergence rate is proven to be independent of the data heterogeneity in the setting where the objective function is non-convex and the stochastic gradient is used.

## 3.1 Setup

We assume that the following standard assumptions hold:

**Assumption 1.** *There exists a constant $f^\star > -\infty$ that satisfies $f(\boldsymbol{x}) \geq f^\star$ for all $\boldsymbol{x} \in \mathbb{R}^d$.*

**Assumption 2.** *There exists a constant $p \in (0, 1]$ that satisfies for all $\boldsymbol{x}_1, \cdots, \boldsymbol{x}_N \in \mathbb{R}^d$,*

$$\|\boldsymbol{X}\boldsymbol{W} - \bar{\boldsymbol{X}}\|_F^2 \leq (1 - p)\|\boldsymbol{X} - \bar{\boldsymbol{X}}\|_F^2, \tag{4}$$

where $\boldsymbol{X} := (\boldsymbol{x}_1, \cdots, \boldsymbol{x}_N) \in \mathbb{R}^{d \times N}$ and $\bar{\boldsymbol{X}} := \frac{1}{N} \boldsymbol{X} \mathbf{1} \mathbf{1}^\top$.

**Assumption 3.** *There exists a constant $L > 0$ that satisfies for all $i \in V$ and $\boldsymbol{x}, \boldsymbol{y} \in \mathbb{R}^d$,*

$$\|\nabla f_i(\boldsymbol{x}) - \nabla f_i(\boldsymbol{y})\| \leq L \|\boldsymbol{x} - \boldsymbol{y}\|. \tag{5}$$

**Assumption 4.** *There exists a constant $\sigma^2$ that satisfies for all $i \in V$ and $\boldsymbol{x}_i \in \mathbb{R}^d$,*

$$\mathbb{E}_{\xi_i \sim \mathcal{D}_i} \|\nabla F_i(\boldsymbol{x}_i; \xi_i) - \nabla f_i(\boldsymbol{x}_i)\|^2 \leq \sigma^2. \tag{6}$$

Assumptions 1, 2, 3, and 4 are commonly used for decentralized learning algorithms (Lian et al., 2017; Yu et al., 2019; Koloskova et al., 2021; Lin et al., 2021; Lu & De Sa, 2021; Yuan et al., 2022). Additionally, the following assumption, which represents data heterogeneity, is commonly used in the convergence analysis of decentralized learning algorithms (Lian et al., 2017; Yu et al., 2019; Lin et al., 2021).

**Assumption 5.** *There exists a constant $\zeta^2$ that satisfies for all $\boldsymbol{x} \in \mathbb{R}^d$,*

$$\frac{1}{N} \sum_{i=1}^N \|\nabla f_i(\boldsymbol{x}) - \nabla f(\boldsymbol{x})\|^2 \leq \zeta^2.$$

Under Assumption 5, the convergence rates of DSGD (Lian et al., 2017), DSGDm (Gao & Huang, 2020; Yuan et al., 2021), QG-DSGDm (Lin et al., 2021), and DecentLaM (Yuan et al., 2021) were shown to be dependent on data heterogeneity $\zeta^2$ and deteriorate as $\zeta^2$ increases. By contrast, in Sec. 3.3, we prove that Momentum Tracking converges without Assumption 5 and the convergence rate is independent of data heterogeneity $\zeta^2$. In addition, we do not assume the convexity of the objective functions $f(\boldsymbol{x})$ and $f_i(\boldsymbol{x})$. Therefore, $f(\boldsymbol{x})$ and $f_i(\boldsymbol{x})$ are potentially non-convex functions (e.g., the loss functions of neural networks).

### 3.2 Momentum Tracking

In this section, we propose **Momentum Tracking**, which is robust to data heterogeneity and accelerated by the momentum. The update rules of Momentum Tracking are defined as follows:

$$\boldsymbol{u}_i^{(r+1)} = \beta \boldsymbol{u}_i^{(r)} + \nabla F_i(\boldsymbol{x}_i^{(r)}; \xi_i^{(r)}), \tag{7}$$

$$\boldsymbol{x}_i^{(r+1)} = \sum_{j \in \mathcal{N}_i^+} W_{ij} \boldsymbol{x}_j^{(r)} - \eta \left( \boldsymbol{u}_i^{(r+1)} - \boldsymbol{c}_i^{(r)} \right), \tag{8}$$

$$\boldsymbol{c}_i^{(r+1)} = \sum_{j \in \mathcal{N}_i^+} W_{ij} \left( \boldsymbol{c}_j^{(r)} - \boldsymbol{u}_j^{(r+1)} \right) + \boldsymbol{u}_i^{(r+1)}, \tag{9}$$

where $\beta \in [0, 1)$ is a momentum coefficient. The pseudo-code for Momentum Tracking is presented in Sec. A. In Momentum Tracking, $\boldsymbol{c}_i$ corrects the local momentum $\boldsymbol{u}_i$ to the global momentum $\frac{1}{N} \sum_j \boldsymbol{u}_j$ and prevents each model parameter $\boldsymbol{x}_i$ from drifting, even when the data distributions are statistically heterogeneous (i.e., the local momentum $\boldsymbol{u}_i$ varies significantly across nodes).

Because Momentum Tracking is equivalent to Gradient Tracking when $\beta = 0$, Momentum Tracking is a simple extension of Gradient Tracking. Hence, when $\beta = 0$, it has been shown that the convergence rate of Momentum Tracking is independent of data heterogeneity $\zeta^2$ (Koloskova et al., 2021). However, because data heterogeneity is amplified when the momentum is used instead of the stochastic gradient (i.e., $\beta > 0$) (Lin et al., 2021; Yuan et al., 2021), it is unclear whether the convergence rate of Momentum Tracking is independent of data heterogeneity $\zeta^2$ for any $\beta \in [0, 1)$ or for only a restricted range of $\beta$. In Sec. 3.3, we provide the convergence rate of Momentum Tracking and prove that it is independent of $\zeta^2$ for any $\beta \in [0, 1)$.

### 3.3 Convergence Analysis

Under Assumptions 1, 2, 3, and 4, Theorem 1 provides the convergence rate of Momentum Tracking. All proofs are presented in Sec. D.

**Theorem 1.** *Suppose that Assumptions 1, 2, 3, and 4 hold, each model parameter $\boldsymbol{x}_i$ is initialized with the same parameters, and both $\boldsymbol{u}_i$ and $\boldsymbol{c}_i$ are initialized as $\frac{1}{1-\beta}(\nabla F_i(\boldsymbol{x}_i^{(0)}; \xi_i^{(0)}) - \frac{1}{N}\sum_{j=1}^N \nabla F_j(\boldsymbol{x}_j^{(0)}; \xi_j^{(0)}))$. Then, for any $\beta \in [0,1)$ and $R \geq 1$, there exists a step size $\eta$ such that the average parameter $\frac{1}{R}\sum_{r=0}^{R-1}\mathbb{E}\left\|\nabla f(\bar{\boldsymbol{x}}^{(r)})\right\|^2$ generated by Eqs. (7-9) is bounded from above by*

$$\mathcal{O}\left(\sqrt{\frac{r_0\sigma^2 L}{NR}} + \left(\frac{r_0^2\sigma^2 L^2}{p^4 R^2(1-\beta)}\left(1 + \frac{p\beta^2}{1-\beta}\right)\right)^{\frac{1}{3}} + \frac{Lr_0}{(1-\beta)p^2 R}\sqrt{1 + \frac{\beta^2}{(1-\beta^2)^3 p}}\right), \tag{10}$$

*where $\bar{\boldsymbol{x}} := \frac{1}{N}\sum_{i=1}^N \boldsymbol{x}_i$ and $r_0 := f(\bar{\boldsymbol{x}}^{(0)}) - f^\star$.*

**Remark 1.** *Theorem 1 assumes that $\boldsymbol{u}_i$ and $\boldsymbol{c}_i$ are initialized as $\frac{1}{1-\beta}(\nabla F_i(\boldsymbol{x}_i^{(0)}; \xi_i^{(0)}) - \frac{1}{N}\sum_{j=1}^N \nabla F_j(\boldsymbol{x}_j^{(0)}; \xi_j^{(0)}))$. Thus, All-Reduce is required only once before starting the training. If we initialize $\boldsymbol{u}_i$ and $\boldsymbol{c}_i$ as zeros, data heterogeneity at initial parameters $\frac{1}{N}\sum_i \|\nabla f_i(\boldsymbol{x}_i^{(0)}) - \nabla f(\boldsymbol{x}_i^{(0)})\|^2$ appears in the convergence rate, but the same phenomenon occurs in the analysis of Gradient Tracking by Koloskova et al. (2021) (see Sec. B).*

**Remark 2.** *Combinations of Gradient Tracking with the momentum or Adam have also been proposed by Xin & Khan (2020) and Carnevale et al. (2022). However, they considered only the setting in which the objective function is strongly convex and the full gradient is used. By contrast, our study focuses on the deep learning setting. Hence, our proof strategies are completely different from those in these previous studies, and Theorem 1 provides the convergence rate in the setting where the objective function is non-convex and the stochastic gradient is used.*

**Remark 3.** *Koloskova et al. (2021) provided the convergence rate of Gradient Tracking in the setting where the objective function is non-convex and the stochastic gradient is used. However, they did not consider the case where the momentum is used, and it is not trivial to provide the convergence rate of Momentum Tracking from the results in this previous work.*

### 3.4 Discussion

**Comparison with Gradient Tracking:** Theorem 1 indicates that the convergence rate of Momentum Tracking does not depend on data heterogeneity $\zeta^2$ for any $\beta \in [0,1)$ and does not deteriorate even when the data distributions are statistically heterogeneous (i.e., $\zeta^2 > 0$). Therefore, Theorem 1 indicates that Momentum Tracking is theoretically robust to data heterogeneity for any $\beta \in [0,1)$. Although Momentum Tracking is a simple extension of Gradient Tracking, our work is the first to identify that the combination of Gradient Tracking and the momentum converges without being affected by data heterogeneity $\zeta^2$ for any $\beta \in [0,1)$ in the setting where the objective function is non-convex and the stochastic gradient is used.

Although the convergence rate of Momentum Tracking Eq. (10) is minimized when $\beta = 0$, Momentum Tracking does accelerate its convergence with the momentum being used ($\beta > 0$), as experimentally demonstrated in Sec. 4. Indeed, the convergence rates of DSGDm (Gao & Huang, 2020) and QG-DSGDm Lin et al. (2021) have the same issue. Thus, it is still an open question to show the theoretical benefits of using $\beta > 0$.

**Comparison with Existing Algorithms with momentum:** Next, we compare the convergence rate of Momentum Tracking with those of existing decentralized learning algorithms with momentum: DSGDm (Gao & Huang, 2020), DecentLaM (Yuan et al., 2021), and QG-DSGDm (Lin et al., 2021). Here, we only show the convergence rate of QG-DSGDm, but the same discussion holds for the other methods. The convergence rate of QG-DSGDm is as follows:

**Theorem 2** (Lin et al. (2021))**.** *Suppose that Assumptions 1, 2, 3, and 4 hold, and Assumption 5 also holds. Then, for any $\beta \in [0, \frac{p}{21+p}]$ and $R \geq 1$, there exists a step size $\eta$ such that $\frac{1}{R}\sum_{r=0}^{R-1}\mathbb{E}\left\|\nabla f(\bar{\boldsymbol{x}}^{(r)})\right\|^2$ generated by QG-DSGDm is bounded from above by*[1]

$$\mathcal{O}\left(\sqrt{\frac{r_0\sigma^2 L}{NR}} + \left(\frac{r_0^2 L^2(\zeta^2 + \sigma^2)}{p^2 R^2}\right)^{\frac{1}{3}} + \frac{Lr_0}{R}\left(\frac{1}{p} + \frac{1}{1-\beta} + \frac{\beta}{(1-\beta)^3}\right)\right),$$

---

[1]For simplicity, we set the additional hyperparameter $\mu$ for QG-DSGDm to $\beta$.

*where $r_0 := f(\bar{\boldsymbol{x}}^{(0)}) - f^\star$.*

Data heterogeneity $\zeta^2$ appears in the second term. Thus, the convergence rate of QG-DSGDm deteriorates when the data distributions held by each node are statistically heterogeneous. By contrast, the convergence rate of Momentum Tracking Eq. (10) does not depend on data heterogeneity $\zeta^2$. Therefore, Momentum Tracking is more robust to data heterogeneity than QG-DSGDm. Because the convergence rates of DSGDm and DecentLaM also depend on $\zeta^2$, the same discussion holds for DSGDm and DecentLaM. Hence, Momentum Tracking is more robust to data heterogeneity than these methods. To the best of our knowledge, Momentum Tracking is the first decentralized learning method with momentum whose convergence rate has been proven to be independent of data heterogeneity $\zeta^2$ in the setting where the objective function is non-convex and the stochastic gradient is used.

Next, we discuss the range of $\beta$. The convergence rates of QG-DSGDm and DecentLaM provided by Lin et al. (2021) and Yuan et al. (2021) hold only when the range of $\beta$ is restricted. For instance, Theorem 2 assumes that $\beta \le \frac{p}{21+p}(< 0.05)$. However, these restrictions on the range of $\beta$ do not hold in practice. (Typically, $\beta$ is set to 0.9.) Therefore, the convergence rates of QG-DSGDm and DecentLaM are unclear in such practical cases. By contrast, Theorem 1 can provide the convergence rate of Momentum Tracking that holds for any $\beta \in [0, 1)$.

**Comparison with SGDm:** Next, we compare the convergence rate of Momentum Tracking with that of SGDm. In a setting where the objective function is non-convex and the stochastic gradient is used, SGDm has been proven to converge to the stationary point with $\mathcal{O}(1/\sqrt{R})$ (Yan et al., 2018; Liu et al., 2020b). By contrast, Theorem 1 indicates that if the number of rounds $R$ is sufficiently large, Momentum Tracking converges with $\mathcal{O}(1/\sqrt{NR})$. Therefore, Momentum Tracking can achieve a linear speedup with respect to the number of nodes $N$, which is a common and important property in decentralized learning methods (Lian et al., 2018; Koloskova et al., 2020).

## 4 Experiment

In this section, we present the results of an experimental evaluation of Momentum Tracking and demonstrate that Momentum Tracking is more robust to data heterogeneity than the existing decentralized learning methods with momentum. In this section, we focus on test accuracy, and more detailed evaluation about the convergence rate is presented in Sec. C.7.

### 4.1 Setup

**Comparison Methods:** (1) DSGD (Lian et al., 2017): the method described in Sec. 2.1; (2) DSGDm (Gao & Huang, 2020): the method described in Sec. 2.2; (3) QG-DSGDm (Lin et al., 2021): a method in which the update rule of the momentum in DSGDm is modified to be more robust to data heterogeneity than DSGDm; (4) DecentLaM (Yuan et al., 2021): a method in which the update rule of the momentum in DSGDm is modified to be more robust to data heterogeneity; (5) Gradient Tracking (Nedić et al., 2017): a method without momentum that is robust to data heterogeneity; (6) Momentum Tracking: the proposed method described in Sec. 3.

**Dataset and Model:** We evaluated Momentum Tracking using three 10-class image classification tasks: FashionMNIST (Xiao et al., 2017), SVHN (Netzer et al., 2011), and CIFAR-10 (Krizhevsky, 2009). Following the previous work (Niwa et al., 2020), we distributed the data to nodes such that each node was given data of randomly selected $k$ classes. When $k = 10$, the data distributions held by each node can be regarded as statistically homogeneous. When $k < 10$, the data distributions are regarded as statistically heterogeneous. We evaluated the comparison methods by setting $k$ to $\{4, 6, 8, 10\}$ and changing data heterogeneity. Note that a smaller $k$ indicates that the data distributions are more heterogeneous. For the neural network architecture, we used LeNet (LeCun et al., 1998) with group normalization (Wu & He, 2018) in Sec. 4.2. In Sec. 4.3, we present more detailed evaluation by varying the neural network architecture (e.g., VGG-11 (Simonyan & Zisserman, 2015) and ResNet-34 (He et al., 2016)). For each comparison method, we used 10% of the training data for validation and individually tuned the step size. For DSGDm, QG-DSGDm,

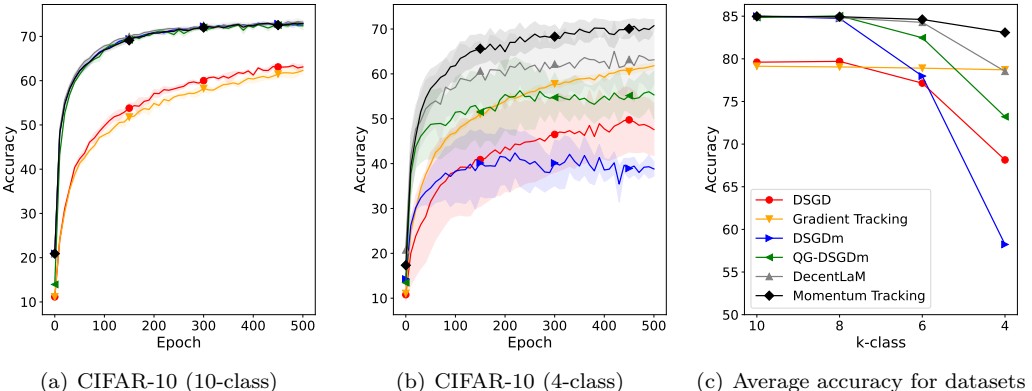

(a) CIFAR-10 (10-class)  (b) CIFAR-10 (4-class)  (c) Average accuracy for datasets

Figure 1: (a) Learning curve on CIFAR-10 with LeNet in the 10-class (i.e., homogeneous) setting. We evaluated the test accuracy per 10 epochs. (b) Learning curve in the 4-class (i.e., heterogeneous) setting. (c) Average test accuracy for all datasets (i.e., FashionMNIST, SVHN, and CIFAR-10).

DecentLaM, and Momentum Tracking, we set $\beta$ to 0.9. All experiments were repeated using three different seed values, and we report their averages. More detailed hyperparameter settings are presented in Sec E.

**Network Topology and Implementation:** Communication efficiency is one of the most important factors in decentralized learning and is determined by the maximum degree of the underlying network topology (Neglia et al., 2019; Wang et al., 2019; Ying et al., 2021). Thus, following these prior works, we present the results of setting the underlying network topology to a ring consisting of eight nodes (i.e., $N = 8$) in Secs. 4.2 and 4.3. In Sec. C.1, we present more detailed evaluation by varying the network topology. All comparison methods were implemented using PyTorch and run on eight GPUs (NVIDIA RTX 3090).

## 4.2 Experimental Results

Table 2 lists the test accuracy for FashionMNIST, SVHN, and CIFAR-10. Fig. 1 (a) and (b) present the learning curves for CIFAR-10 and Fig. 1 (c) presents the average test accuracy for all datasets.

**Comparison of Momentum Tracking and Gradient Tracking:** First, we discuss the results of Momentum Tracking and Gradient Tracking. Table 2 and Fig. 1 indicate that Momentum Tracking achieves a higher accuracy faster than Gradient Tracking and outperforms Gradient Tracking in all settings. When the data distributions are homogeneous (i.e., 10-class), Momentum Tracking outperforms Gradient Tracking by 5.8% on average. When the data distributions are heterogeneous (e.g., 4-class), Momentum Tracking outperforms Gradient Tracking by 4.4% on average. Thus, the results show that Momentum Tracking can consistently outperform Gradient Tracking regardless of data heterogeneity.

**Comparison of Momentum Tracking and DSGDm:** Next, we discuss the results of Momentum Tracking and DSGDm. The results show that when the data distributions are homogeneous (i.e., 10-class), Momentum Tracking and DSGDm are comparable and outperform DSGD and Gradient Tracking. However, when the data distributions are heterogeneous (e.g., 4-class), the test accuracy of DSGDm decreases even more than that of DSGD, and DSGDm underperforms DSGD by 9.9% on average. By contrast, the results indicate that Momentum Tracking consistently outperforms DSGD and Gradient Tracking by 14.9% and 4.4% respectively when the data distributions are heterogeneous. The results indicate that Momentum Tracking is more robust to data heterogeneity than DSGDm and outperforms DSGDm by 24.9% on average.

**Comparison of Momentum Tracking, QG-DSGDm, and DecentLaM:** When the data distributions are homogeneous (i.e., 10-class), Momentum Tracking, QG-DSGDm, and DecentLaM are comparable and outperform DSGD and Gradient Tracking. By contrast, when the data distributions are heterogeneous (e.g., 4-class), Momentum Tracking consistently outperforms QG-DSGDm and DecentLaM by 9.9% and 4.5% respectively, whereas QG-DSGDm and DecentLaM are more robust to data heterogeneity than DSGDm. Hence, these results are consistent with our theoretical analysis, as discussed in Secs. 3.3 and 3.4.

Table 2: Test accuracy on FashionMNIST, SVHN, and CIFAR-10 with LeNet. "$k$-class" means that each node has only the data of randomly selected $k$ classes. Bold font means the highest accuracy.

| | FashionMNIST | | | |
| --- | --- | --- | --- | --- |
| | 10-class | 8-class | 6-class | 4-class |
| DSGD | $85.6 \pm 0.49$ | $85.6 \pm 0.41$ | $82.7 \pm 1.12$ | $78.1 \pm 1.56$ |
| Gradient Tracking | $85.0 \pm 0.49$ | $85.4 \pm 0.26$ | $85.0 \pm 0.37$ | $84.9 \pm 0.22$ |
| DSGDm | $89.5 \pm 0.15$ | $89.3 \pm 0.21$ | $82.1 \pm 3.23$ | $68.7 \pm 5.02$ |
| QG-DSGDm | $\mathbf{89.6 \pm 0.10}$ | $\mathbf{89.5 \pm 0.47}$ | $86.9 \pm 1.59$ | $80.8 \pm 2.94$ |
| DecentLaM | $89.5 \pm 0.14$ | $89.3 \pm 0.36$ | $\mathbf{89.2 \pm 0.41}$ | $84.3 \pm 3.05$ |
| Momentum Tracking | $89.5 \pm 0.36$ | $89.4 \pm 0.05$ | $88.9 \pm 0.47$ | $\mathbf{86.8 \pm 1.56}$ |

| | SVHN | | | |
| --- | --- | --- | --- | --- |
| | 10-class | 8-class | 6-class | 4-class |
| DSGD | $90.1 \pm 0.17$ | $89.5 \pm 0.61$ | $87.6 \pm 1.94$ | $78.8 \pm 8.55$ |
| Gradient Tracking | $90.1 \pm 0.30$ | $89.8 \pm 0.38$ | $89.8 \pm 0.39$ | $89.4 \pm 0.47$ |
| DSGDm | $\mathbf{92.6 \pm 0.35}$ | $92.4 \pm 0.19$ | $88.1 \pm 4.38$ | $67.2 \pm 9.69$ |
| QG-DSGDm | $92.5 \pm 0.22$ | $\mathbf{92.5 \pm 0.17}$ | $90.9 \pm 1.67$ | $83.5 \pm 7.14$ |
| DecentLaM | $92.4 \pm 0.21$ | $92.2 \pm 0.39$ | $92.0 \pm 0.48$ | $88.2 \pm 4.75$ |
| Momentum Tracking | $\mathbf{92.6 \pm 0.32}$ | $92.4 \pm 0.40$ | $\mathbf{92.3 \pm 0.23}$ | $\mathbf{91.7 \pm 0.53}$ |

| | CIFAR-10 | | | |
| --- | --- | --- | --- | --- |
| | 10-class | 8-class | 6-class | 4-class |
| DSGD | $63.1 \pm 0.60$ | $64.1 \pm 0.52$ | $61.2 \pm 1.16$ | $47.6 \pm 5.77$ |
| Gradient Tracking | $62.3 \pm 0.73$ | $62.0 \pm 0.80$ | $61.9 \pm 0.58$ | $61.8 \pm 0.82$ |
| DSGDm | $72.9 \pm 0.41$ | $72.5 \pm 0.20$ | $63.8 \pm 6.24$ | $38.8 \pm 1.61$ |
| QG-DSGDm | $72.4 \pm 0.87$ | $\mathbf{73.1 \pm 0.16}$ | $69.6 \pm 2.42$ | $55.3 \pm 5.30$ |
| DecentLaM | $\mathbf{73.2 \pm 0.36}$ | $72.9 \pm 0.14$ | $71.7 \pm 1.10$ | $63.1 \pm 5.43$ |
| Momentum Tracking | $72.9 \pm 0.59$ | $73.0 \pm 0.49$ | $\mathbf{72.6 \pm 0.41}$ | $\mathbf{70.7 \pm 1.38}$ |

Table 3: Test accuracy on CIFAR-10 with VGG-11 and ResNet-34. "$k$-class" indicates that each node has only the data of randomly selected $k$ classes, and bold font indicates the highest accuracy.

| | CIFAR-10 + VGG-11 | | | CIFAR-10 + ResNet-34 | | |
| --- | --- | --- | --- | --- | --- | --- |
| | 10-class | 4-class | 2-class | 10-class | 4-class | 2-class |
| DSGD | $91.3 \pm 0.12$ | $86.9 \pm 1.75$ | $71.1 \pm 2.82$ | $94.3 \pm 0.13$ | $90.0 \pm 1.65$ | $63.5 \pm 0.90$ |
| Gradient Tracking | $88.1 \pm 0.14$ | $86.3 \pm 0.50$ | $83.0 \pm 0.04$ | $85.9 \pm 0.71$ | $82.6 \pm 0.33$ | $76.2 \pm 0.30$ |
| DSGDm | $\mathbf{92.2 \pm 0.09}$ | $77.3 \pm 4.05$ | $39.6 \pm 5.92$ | $95.8 \pm 0.26$ | $79.0 \pm 3.69$ | $27.7 \pm 2.83$ |
| QG-DSGDm | $92.0 \pm 0.02$ | $89.5 \pm 1.08$ | $77.8 \pm 1.96$ | $95.8 \pm 0.22$ | $94.3 \pm 1.13$ | $79.9 \pm 1.59$ |
| DecentLaM | $92.1 \pm 0.09$ | $\mathbf{90.9 \pm 0.65}$ | $85.2 \pm 0.67$ | $\mathbf{95.9 \pm 0.04}$ | $\mathbf{95.2 \pm 0.51}$ | $89.2 \pm 2.26$ |
| Momentum Tracking | $91.9 \pm 0.06$ | $\mathbf{90.9 \pm 0.60}$ | $\mathbf{87.0 \pm 0.48}$ | $95.0 \pm 0.13$ | $94.4 \pm 0.52$ | $\mathbf{89.9 \pm 0.73}$ |

In summary, when the data distributions are homogeneous, DSGDm, QG-DSGDm, DecentLaM, and Momentum Tracking are comparable and outperform DSGD and Gradient Tracking. When the data distributions are heterogeneous, Momentum Tracking is more robust to data heterogeneity than DSGDm, QG-DSGDm, and DecentLaM, and can outperform all comparison methods.

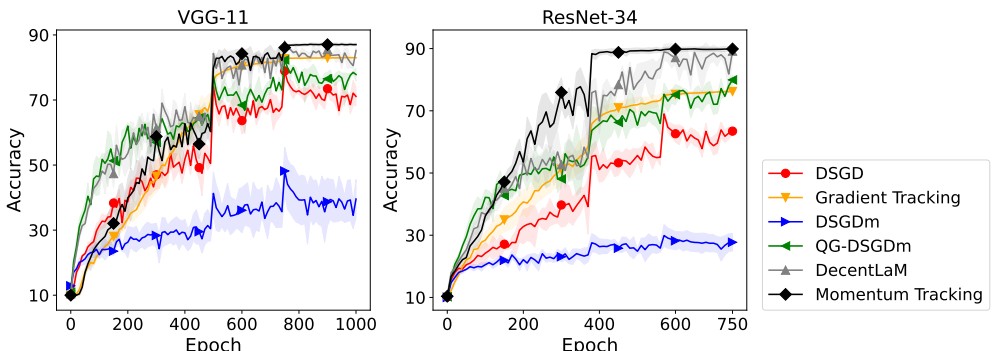

Figure 2: Learning curves for CIFAR-10 with VGG-11 and ResNet-34 in the 2-class setting.

### 4.3 Results with Various Neural Network Architectures

Next, we evaluated Momentum Tracking in more detail by varying the neural network architecture. Table 3 lists the test accuracy with VGG-11 (Simonyan & Zisserman, 2015) and ResNet-34 (He et al., 2016) when we set $k$ to $\{2, 4, 10\}$, and Fig. 2 shows the learning curves.

For both neural network architectures, Table 3 reveals that when the data distributions are homogeneous (i.e., 10-class), Momentum Tracking is comparable with DSGDm, QG-DSGDm, and DecentLaM and outperforms DSGD and Gradient Tracking. By contrast, when the data distributions are heterogeneous (e.g., 2-class), Table 3 and Fig. 2 reveal that Momentum Tracking outperforms all comparison methods for both neural network architectures. In particular, Fig. 2 indicates that DSGDm, QG-DSGDm, and DecentLaM are unstable and continue to oscillate in the final training phase, whereas Momentum Tracking converges stably. These results are consistent with those of LeNet presented in Table 2.

## 5 Conclusion

In this study, we propose Momentum Tracking, which is a method with momentum whose convergence rate is proven to be independent of data heterogeneity. More specifically, we provide the convergence rate of Momentum Tracking in the setting where the objective function is non-convex and the stochastic gradient is used. Our theoretical analysis reveals that the convergence rate of Momentum Tracking is independent of data heterogeneity for any $\beta \in [0, 1)$. Through image classification tasks, we demonstrated that Momentum Tracking can consistently outperform the decentralized learning methods without momentum regardless of data heterogeneity. Moreover, we showed that Momentum Tracking is more to data heterogeneity than existing decentralized learning methods with momentum and can consistently outperform these existing methods when the data distributions are heterogeneous.

## Acknowledgments

Yuki Takezawa, Ryoma Sato, and Makoto Yamada were supported by JSPS KAKENHI Grant Number 23KJ1336, 21J22490, and MEXT KAKENHI Grant Number 20H04243, respectively.

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

## A Pseudo-Codes

The pseudo-codes for Momentum Tracking, QG-DSGDm, and DecentLaM are given in the following, where $\mathbf{Transmit}_{i\to j}(\cdot)$ denotes that node $i$ transmits parameters to node $j$ and $\mathbf{Receive}_{i\leftarrow j}(\cdot)$ denotes that node $i$ receives parameters from node $j$.

---

**Algorithm 1:** Update rules of Momentum Tracking at node $i$.

---

1: **Input:** Step size $\eta > 0$, $\beta \in (0,1]$, mixing matrix $\boldsymbol{W}$. Initialize $\boldsymbol{c}_i$ and $\boldsymbol{u}_i$ to
   $\frac{1}{1-\beta}(\nabla F_i(\boldsymbol{x}_i^{(0)};\xi_i^{(0)}) - \frac{1}{N}\sum_j \nabla F_j(\boldsymbol{x}_j^{(0)};\xi_j^{(0)}))$ for all $i \in V$ and $\boldsymbol{x}_i$ with the same parameter.
2: **for** $r = 0,\cdots,R$ **do**
3:     $\boldsymbol{u}_i^{(r+1)} \leftarrow \beta\boldsymbol{u}_i^{(r)} + \nabla F_i(\boldsymbol{x}_i^{(r)};\xi_i^{(r)})$.
4:     **for** $j \in \mathcal{N}_i$ **do**
5:         $\mathbf{Transmit}_{i\to j}(\boldsymbol{x}_i^{(r)})$ and $\mathbf{Receive}_{i\leftarrow j}(\boldsymbol{x}_j^{(r)})$.
6:         $\mathbf{Transmit}_{i\to j}(\boldsymbol{c}_i^{(r)} - \boldsymbol{u}_i^{(r+1)})$ and $\mathbf{Receive}_{i\leftarrow j}(\boldsymbol{c}_j^{(r)} - \boldsymbol{u}_j^{(r+1)})$.
7:     **end for**
8:     $\boldsymbol{x}_i^{(r+1)} \leftarrow \sum_{j\in\mathcal{N}_i^+} W_{ij}\boldsymbol{x}_j^{(r)} - \eta\left(\boldsymbol{u}_i^{(r+1)} - \boldsymbol{c}_i^{(r)}\right)$.
9:     $\boldsymbol{c}_i^{(r+1)} \leftarrow \sum_{j\in\mathcal{N}_i^+} W_{ij}\left(\boldsymbol{c}_j^{(r)} - \boldsymbol{u}_j^{(r+1)}\right) + \boldsymbol{u}_i^{(r+1)}$.
10: **end for**

---

**Algorithm 2:** Update rules of QG-DSGDm at node $i$.

---

1: **Input:** Step size $\eta > 0$, $\beta,\mu \in (0,1]$, mixing matrix $\boldsymbol{W}$. Initialize $\hat{\boldsymbol{u}}_i$ to zero for all $i \in V$ and $\boldsymbol{x}_i$ with
   the same parameter.
2: **for** $r = 0,\cdots,R$ **do**
3:     $\boldsymbol{u}_i^{(r+1)} \leftarrow \beta\hat{\boldsymbol{u}}_i^{(r)} + \nabla F_i(\boldsymbol{x}_i^{(r)};\xi_i^{(r)})$.
4:     $\boldsymbol{x}_i^{(r+\frac{1}{2})} \leftarrow \boldsymbol{x}_i^{(r)} - \eta\boldsymbol{u}_i^{(r+1)}$
5:     **for** $j \in \mathcal{N}_i$ **do**
6:         $\mathbf{Transmit}_{i\to j}(\boldsymbol{x}_i^{(r+\frac{1}{2})})$ and $\mathbf{Receive}_{i\leftarrow j}(\boldsymbol{x}_j^{(r+\frac{1}{2})})$.
7:     **end for**
8:     $\boldsymbol{x}_i^{(r+1)} \leftarrow \sum_{j\in\mathcal{N}_i^+} W_{ij}\boldsymbol{x}_j^{(r+\frac{1}{2})}$.
9:     $\boldsymbol{d}_i^{(r+1)} \leftarrow \frac{\boldsymbol{x}_i^{(r)} - \boldsymbol{x}_i^{(r+1)}}{\eta}$.
10:     $\hat{\boldsymbol{u}}_i^{(r+1)} \leftarrow \mu\hat{\boldsymbol{u}}_i^{(r)} + (1-\mu)\boldsymbol{d}_i^{(r+1)}$.
11: **end for**

---

**Algorithm 3:** Update rules of DecentLaM at node $i$.

---

1: **Input:** Step size $\eta > 0$, $\beta \in (0,1]$, mixing matrix $\boldsymbol{W}$. Initialize $\boldsymbol{u}_i$ to zero for all $i \in V$ and $\boldsymbol{x}_i$ with the
   same parameter.
2: **for** $r = 0,\cdots,R$ **do**
3:     $\boldsymbol{x}_i^{(r+\frac{1}{2})} \leftarrow \boldsymbol{x}_i^{(r)} - \eta\nabla F_i(\boldsymbol{x}_i^{(r)};\xi_i^{(r)})$.
4:     **for** $j \in \mathcal{N}_i$ **do**
5:         $\mathbf{Transmit}_{i\to j}(\boldsymbol{x}_i^{(r+\frac{1}{2})})$ and $\mathbf{Receive}_{i\leftarrow j}(\boldsymbol{x}_j^{(r+\frac{1}{2})})$.
6:     **end for**
7:     $\hat{\boldsymbol{g}}_i^{(r+1)} \leftarrow \frac{1}{\eta}\boldsymbol{x}_i^{(r)} - \frac{1}{\eta}\sum_{j\in\mathcal{N}_i} W_{ij}\boldsymbol{x}_j^{(r+\frac{1}{2})}$.
8:     $\boldsymbol{u}_i^{(r+1)} \leftarrow \beta\boldsymbol{u}_i^{(r)} + \hat{\boldsymbol{g}}_i^{(r+1)}$.
9:     $\boldsymbol{x}_i^{(r+1)} \leftarrow \boldsymbol{x}_i^{(r)} - \eta\boldsymbol{u}_i^{(r+1)}$.
10: **end for**

---

# B   Additional Discussion about Convergence Rate

## B.1   Comparison with Gradient Tracking

Because Momentum Tracking is equivalent to Gradient Tracking when $\beta = 0$, Theorem 1 also provides the convergence rate of Gradient Tracking. In this section, we compare the convergence rate of Gradient Tracking provided in Theorem 1 to that provided by Koloskova et al. (2021).

From Theorem 1, we get the following statement.

**Corollary 1.** *Suppose that $\beta = 0$ and the assumptions of Theorem 1 hold. Then, for any $R \geq 1$, there exists a step size $\eta$ such that the average parameter $\bar{\boldsymbol{x}} := \frac{1}{N} \sum_i \boldsymbol{x}_i$ generated by Eqs. (7-9) satisfies*

$$\frac{1}{R} \sum_{r=0}^{R-1} \mathbb{E} \left\| \nabla f(\bar{\boldsymbol{x}}^{(r)}) \right\|^2 \leq \mathcal{O} \left( \sqrt{\frac{r_0 \sigma^2 L}{NR}} + \left( \frac{r_0 \sigma L}{p^2 R} \right)^{\frac{2}{3}} + \frac{L r_0}{p^2 R} \right), \tag{11}$$

*where $r_0 := f(\bar{\boldsymbol{x}}^{(0)}) - f^\star$.*

Then, under Assumptions 1, 2, 3, and 4, Koloskova et al. (2021) provided the convergence rate of Gradient Tracking as follows.

**Theorem 3** (Koloskova et al. (2021))**.** *Suppose that Assumptions 1, 2, 3, and 4 hold, each model parameter $\boldsymbol{x}_i$ is initialized with the same parameters, and $\boldsymbol{c}_i$ is initialized as $\boldsymbol{0}$. Then, for any round $R > \frac{2}{p} \log(\frac{50}{p}(1 + \log \frac{1}{p}))$, there exists a step size $\eta$ that satisfies that the average parameter $\bar{\boldsymbol{x}} := \frac{1}{N} \sum_i \boldsymbol{x}_i$ generated by Gradient Tracking satisfies*

$$\frac{1}{R} \sum_{r=0}^{R-1} \left\| \nabla f(\bar{\boldsymbol{x}}^{(r)}) \right\|^2 \leq \tilde{\mathcal{O}} \left( \sqrt{\frac{r_0 \sigma^2 L}{NR}} + \left( \frac{r_0 \sigma L}{(\sqrt{p} c + p \sqrt{N}) R} \right)^{\frac{2}{3}} + \frac{L(r_0 + L \zeta_0^2)}{pcR} \right), \tag{12}$$

*where $\tilde{O}(\cdot)$ hides the polylogarithmic factors, $\zeta_0^2 := \frac{1}{N} \sum_i \| \nabla F_i(\bar{\boldsymbol{x}}^{(0)}; \xi_i) - \frac{1}{N} \sum_j \nabla F_j(\bar{\boldsymbol{x}}^{(0)}; \xi_j) \|^2$, $c := 1 - \min\{\lambda_{min}, 0\}^2$, and $\lambda_{min}$ is the minimum eigenvalue of $\boldsymbol{W}$.*

Comparing the convergence rates in Eqs. (11) and (12), the convergence rate in Eq. (12) is tighter than that in Eq. (11) because $c \geq p$ for any mixing matrix $\boldsymbol{W}$. However, because the convergence rate in Eq. (12) holds only when the number of round $R$ is larger than $\frac{2}{p} \log(\frac{50}{p}(1 + \log \frac{1}{p}))$, Theorem 3 can not describe the behavior of the convergence rate at the beginning of the training. In contrast, Corollary 1 provides the convergence rate for Gradient Tracking that holds for any $R \geq 1$.

## B.2   Comparison with Other Decentralized Learning Methods

Lu & De Sa (2021) and Yuan et al. (2022) proposed DeTAG and MG-DSGD that can achieve optimal convergence rates by using algorithmic techniques such as gradient accumulation and multiple gossip averaging. However, Assumption 5 is necessary for both analyses. Then, as the data heterogeneity becomes large, the convergence rate of DeTAG deteriorates, and the number of multiple gossip averaging increases. The goal of our study is to propose a method with momentum whose convergence rate is independent of data heterogeneity. Thus, we leave it to future work to compare Momentum Tracking with these methods.

# C   Additional Experiments

## C.1   Results with Various Network Topologies

We evaluated Momentum Tracking in more detail by changing the underlying network topology. Table 4 lists the test accuracy of all comparison methods when we set the underlying network topology to be a hypercube or a semantic exponential graph.

Table 4 indicates that when the data distributions held by each node are statistically homogeneous (i.e., 10-class), DSGDm, QG-DSGDm, DecentLaM, and Momentum Tracking are comparable and outperform

DSGD and Gradient Tracking for all network topologies. When the data distributions are heterogeneous (i.e., 4-class), the results show that Momentum Tracking is more robust to data heterogeneity than DSGDm, QG-DSGDm, and DecentLaM and outperforms all comparison methods for all network topologies. Therefore, the results indicate that Momentum Tracking is robust to data heterogeneity regardless of the underlying network topology.

Table 4: Test accuracy on CIFAR-10 with different underlying network topologies.

| | CIFAR-10 | | | |
| | Hypercube | | Semantic Exponential Graph | |
| | 10-class | 4-class | 10-class | 4-class |
|---|---|---|---|---|
| DSGD | $63.3 \pm 0.65$ | $55.9 \pm 4.11$ | $64.0 \pm 0.26$ | $60.7 \pm 1.82$ |
| Gradient Tracking | $61.0 \pm 1.34$ | $60.2 \pm 1.13$ | $62.4 \pm 0.53$ | $62.4 \pm 0.89$ |
| DSGDm | $\mathbf{73.2 \pm 0.09}$ | $45.0 \pm 5.90$ | $\mathbf{73.4 \pm 0.13}$ | $51.5 \pm 7.80$ |
| QG-DSGDm | $73.0 \pm 0.31$ | $62.9 \pm 3.68$ | $\mathbf{73.4 \pm 0.58}$ | $70.2 \pm 1.09$ |
| DecentLaM | $72.9 \pm 0.24$ | $69.1 \pm 4.05$ | $72.9 \pm 0.73$ | $71.2 \pm 1.72$ |
| Momentum Tracking | $72.8 \pm 0.15$ | $\mathbf{72.7 \pm 0.28}$ | $72.7 \pm 0.33$ | $\mathbf{72.9 \pm 0.07}$ |

## C.2 Results with Other Heterogeneous Setting

In Sec. 4, we show the results when the data are distributed such that each node had data of randomly selected $k$ classes. In this section, we show the results in another heterogeneous setting, where the label distributions of each node are determined by Dirichlet distributions Hsu et al. (2019).

Table 5 lists the results when we distributed data using Dirichlet distributions. The results indicate that Momentum Tracking is more robust to the data heterogeneity than DSGDm, QG-DSGDm, and DecentLaM in both cases where we use Dirichlet distributions and where we use $k$-class setting.

Table 5: Test accuracy on CIFAR-10 with different $\alpha$.

| | CIFAR-10 + VGG-11 | |
| | $\alpha = 10$ (homogeneous case) | $\alpha = 0.1$ (heterogeneous case) |
|---|---|---|
| QG-DSGDm | $89.7 \pm 0.07$ | $87.2 \pm 1.54$ |
| DecentLaM | $90.1 \pm 0.28$ | $88.6 \pm 0.99$ |
| Momentum Tracking | $\mathbf{90.5 \pm 0.01}$ | $\mathbf{90.2 \pm 0.37}$ |

## C.3 Initial Value Analysis

In this section, we discuss the initial values of $\boldsymbol{c}_i$ and $\boldsymbol{u}_i$. Table 6 lists the test accuracy for Momentum Tracking when we initialize $\boldsymbol{c}_i$ and $\boldsymbol{u}_i$ to zero and when we initialize $\boldsymbol{c}_i$ and $\boldsymbol{u}_i$ as in Theorem 1. The results indicate that the test accuracy are almost equivalent on both settings. Hence, Theorem 1 requires $\boldsymbol{c}_i$ and $\boldsymbol{u}_i$ to be initialized to $\frac{1}{1-\beta}(\nabla F_i(\boldsymbol{x}_i^{(0)}; \xi_i^{(0)}) - \frac{1}{N}\sum_{j=1}^{N} \nabla F_j(\boldsymbol{x}_j^{(0)}; \xi_j^{(0)}))$. However, in practice, $\boldsymbol{c}_i$ and $\boldsymbol{u}_i$ can be initialized to zeros without any impact on accuracy.

## C.4 Comparison with RelaySum

In this section, we compare Momentum Tracking with RelaySum Vogels et al. (2021), which is one of the methods that are most robust to data heterogeneity, and RelaySum with momentum (RelaySumM). Table 7 lists the accuracy on CIFAR-10 with VGG-11. The results indicate that RelaySumM is more robust to data heterogeneity than Momentum Tracking and outperforms Momentum Tracking in the 2-class setting. However, the convergence rate of RelaySum is proven to be independent of data heterogeneity only when the momentum is not applied, and it remains to be unclear whether the convergence rate of RelaySum is independent of data heterogeneity when the momentum is applied. Thus, it is a clear advantage that the convergence rate of Momentum Tracking is proven to be independent of data heterogeneity for any

Table 6: Test accuracy on FashionMNIST, SVHN, and CIFAR-10 with LeNet. "$k$-class" indicates that each node has only the data of randomly selected $k$ classes.

| | FashionMNIST | | | |
| | 10-class | 8-class | 6-class | 4-class |
|---|---|---|---|---|
| Momentum Tracking | $89.5 \pm 0.36$ | $89.4 \pm 0.05$ | $88.9 \pm 0.47$ | $86.8 \pm 1.56$ |
| Momentum Tracking ($\boldsymbol{c}_i^{(0)} = \boldsymbol{u}_i^{(0)} = \boldsymbol{0}$) | $89.5 \pm 0.38$ | $89.4 \pm 0.04$ | $88.7 \pm 0.63$ | $85.8 \pm 1.53$ |

| | SVHN | | | |
| | 10-class | 8-class | 6-class | 4-class |
|---|---|---|---|---|
| Momentum Tracking | $92.6 \pm 0.32$ | $92.4 \pm 0.40$ | $92.3 \pm 0.23$ | $91.7 \pm 0.53$ |
| Momentum Tracking ($\boldsymbol{c}_i^{(0)} = \boldsymbol{u}_i^{(0)} = \boldsymbol{0}$) | $92.5 \pm 0.34$ | $92.3 \pm 0.50$ | $92.2 \pm 0.29$ | $92.0 \pm 0.81$ |

| | CIFAR-10 | | | |
| | 10-class | 8-class | 6-class | 4-class |
|---|---|---|---|---|
| Momentum Tracking | $72.9 \pm 0.59$ | $73.0 \pm 0.49$ | $72.6 \pm 0.41$ | $70.7 \pm 1.38$ |
| Momentum Tracking ($\boldsymbol{c}_i^{(0)} = \boldsymbol{u}_i^{(0)} = \boldsymbol{0}$) | $72.8 \pm 0.35$ | $72.9 \pm 0.32$ | $73.0 \pm 0.41$ | $70.7 \pm 2.00$ |

momentum coefficient $\beta \in [0, 1)$. The main objective and contribution of our study are not to achieve state-of-the-art, but to propose a method with momentum whose convergence rate is proven to be independent of the data heterogeneity. We believe that our proof is helpful for future research that will attempt to analyze the convergence rates when the momentum is applied (e.g., the convergence rate of RelaySumM).

Table 7: Test accuracy on CIFAR-10 with VGG-11.

| | CIFAR-10 + VGG-11 | |
| | 10-class | 2-class |
|---|---|---|
| DSGD | $91.3 \pm 0.12$ | $71.1 \pm 2.82$ |
| Gradient Tracking | $88.1 \pm 0.14$ | $83.0 \pm 0.04$ |
| RelaySum | $91.1 \pm 0.13$ | $89.0 \pm 0.15$ |
| DSGDm | $\mathbf{92.2 \pm 0.09}$ | $39.6 \pm 5.92$ |
| QG-DSGDm | $92.0 \pm 0.02$ | $77.8 \pm 1.96$ |
| DecentLaM | $92.1 \pm 0.09$ | $85.2 \pm 0.67$ |
| RelaySumM | $92.1 \pm 0.13$ | $\mathbf{89.3 \pm 0.76}$ |
| Momentum Tracking | $91.9 \pm 0.06$ | $87.0 \pm 0.48$ |

## C.5 Comparision with ABm and GTAdam

In this section, we compared Momentum Tracking with ABm (Xin & Khan, 2020) and GTAdam (Carnevale et al., 2022), showing the results in Table 8. The update rules of Momentum Tracking are slightly different from that of ABm and GTAdam, but the results indicate that they can achieve almost the same accuracy. However, as we mention in Remark 2, ABm and GTAdam are proven to be independent of data heterogeneity only in the strongly convex setting. Thus, Momentum Tracking is the first method with momentum whose convergence rate is proven to be independent of data heterogeneity in non-convex and stochastic settings.

## C.6 Learning Curves

In this section, we present the learning curves for the results whose final accuracy are presented in Tables 2 and 3. Figs. 3, 4, and 5 show the learning curves for FashionMNSIT, SVHN, and CIFAR-10, respectively, with LeNet. Figs. 6 and 7 show the learning curves for CIFAR-10 with VGG-11 and ResNet-34, respectively.

Table 8: Test accuracy on CIFAR-10 with LeNet.

|  | 10-class | 4-class |
|---|---|---|
| Momentum Tracking | $72.9 \pm 0.59$ | $70.7 \pm 1.38$ |
| ABm | $71.0 \pm 0.20$ | $71.1 \pm 0.20$ |
| GTAdam | $71.4 \pm 0.56$ | $67.7 \pm 3.20$ |

When the data distributions are statistically homogeneous (i.e., 10-class), the results indicate that DSGDm, QG-DSGDm, DecentLaM, and Momentum Tracking are comparable and can achieve high accuracy faster than DSGD and Gradient Tracking. When the data distributions are statistically heterogeneous (e.g., 2-class and 4-class), the results indicate that the learning curves for Momentum Tracking are more stable than those for DSGDm, QG-DSGDm, and DecentLaM, and Momentum Tracking outperforms all comparison methods. In particular, in Figs. 6 and 7, the accuracy of DSGD, DSGDm, QG-DSGDm, and DecentLaM continue to oscillate in the final training phase in the 2-class setting, whereas the accuracy of Momentum Tracking and Gradient Tracking converge in the 2-class setting as well as in the 10-class setting. Therefore, Momentum Tracking is more robust to data heterogeneity than DSGDm, QG-DSGDm, and DecentLaM.

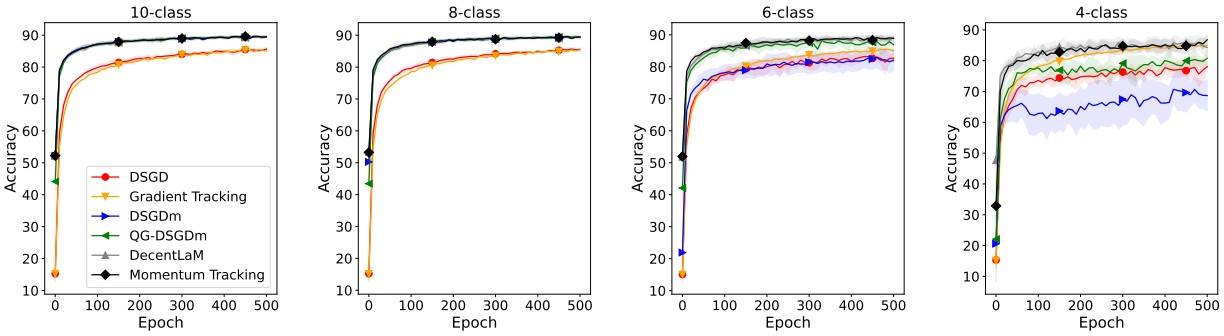

Figure 3: Learning curves on FashionMNIST. The accuracy is evaluated per 10 epochs.

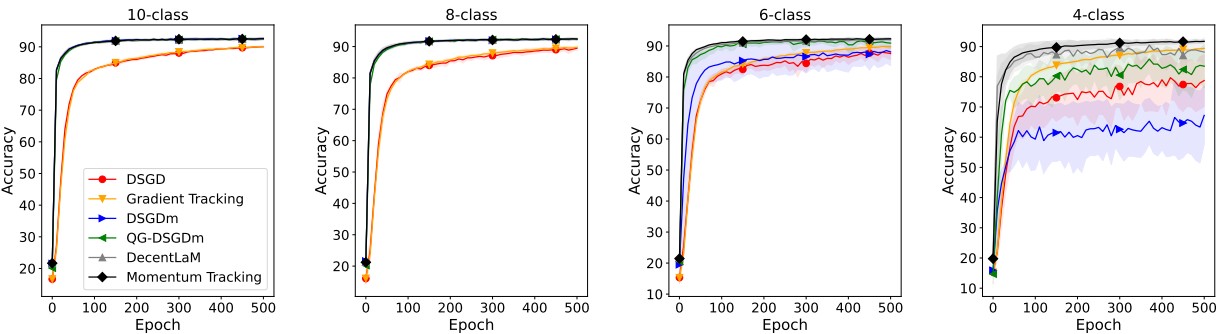

Figure 4: Learning curves on SVHN. The accuracy is evaluated per 10 epochs.

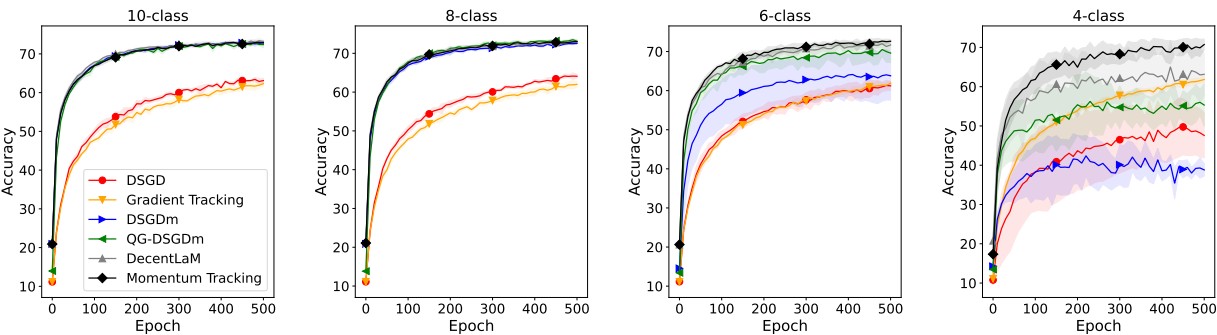

Figure 5: Learning curves on CIFAR-10. The accuracy is evaluated per 10 epochs.

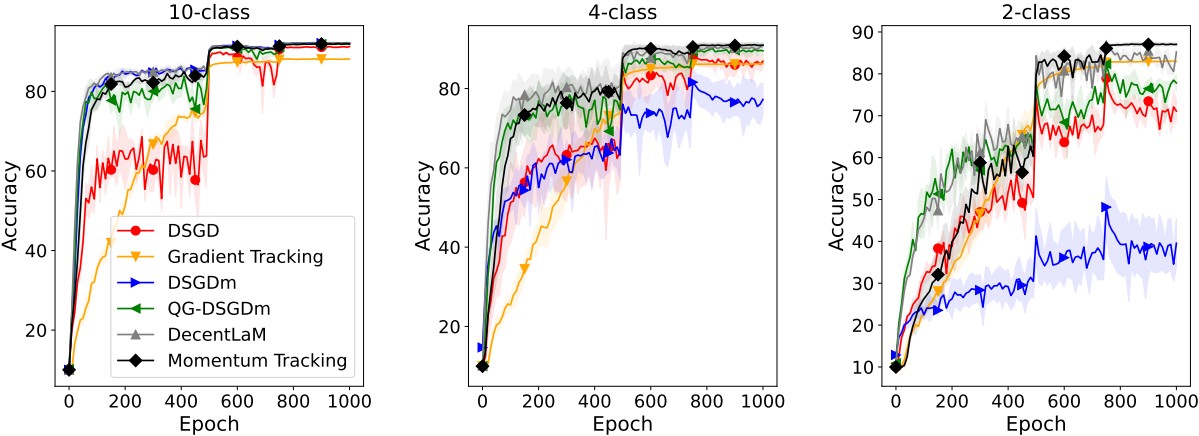

Figure 6: Learning curves on CIFAR-10 with VGG-11. The accuracy is evaluated per 10 epochs.

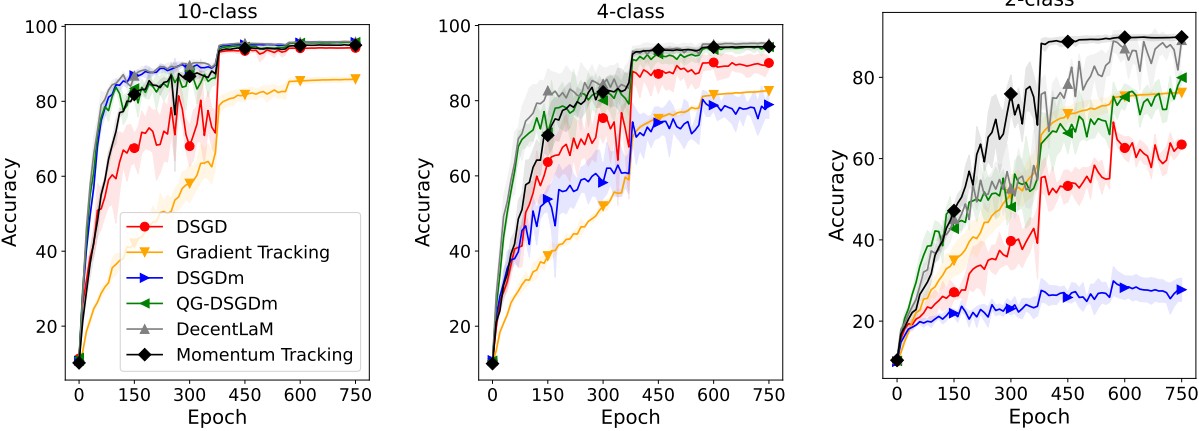

Figure 7: Learning curves on CIFAR-10 with ResNet-34. The accuracy is evaluated per 10 epochs.

### C.7 Synthetic Experiment

In this section, we evaluate the convergence rate in more detail using a synthetic dataset. Following the previous work (Koloskova et al., 2020), we set the dimension of parameter $d$ to 50, the number of nodes $N$ to 25, and the network topology to a ring consisting of $N$ nodes. We then defined the local objective function $f_i(\boldsymbol{x})$ to be $\frac{1}{2}\|\boldsymbol{A}_i\boldsymbol{x} - \boldsymbol{b}_i\|^2$ where $\boldsymbol{A}_i := i/\sqrt{N}$ and $\boldsymbol{b}_i$ are sampled from $\mathcal{N}(\boldsymbol{0}, \zeta^2/i^2\boldsymbol{1})$, and we defined the stochastic gradient $\nabla F_i(\boldsymbol{x}; \xi_i)$ to be $\nabla f_i(\boldsymbol{x}) + \epsilon$ where $\epsilon$ is drawn from $\mathcal{N}(\boldsymbol{0}; \sigma^2/d\boldsymbol{1})$. For all comparison methods, we set the step size $\eta$ to $1.0 \times 10^{-4}$.

Figs. 8 and 9 illustrate $\|\nabla f(\bar{\boldsymbol{x}})\|^2$ with respect to the number of rounds $r$ when we vary data heterogeneity $\zeta^2$ as $\{0, 25, 50\}$ and set $\sigma^2$ to one. The results show that Momentum Tracking converges in the same manner regardless of data heterogeneity $\zeta^2$. On the other hand, for DSGDm, QG-DSGDm, and DecentLaM, $\|\nabla f(\bar{\boldsymbol{x}})\|^2$ increases as data heterogeneity $\zeta^2$ increases. Hence, these results are consistent with our theoretical analysis that the convergence rate of Momentum Tracking is independent of data heterogeneity.

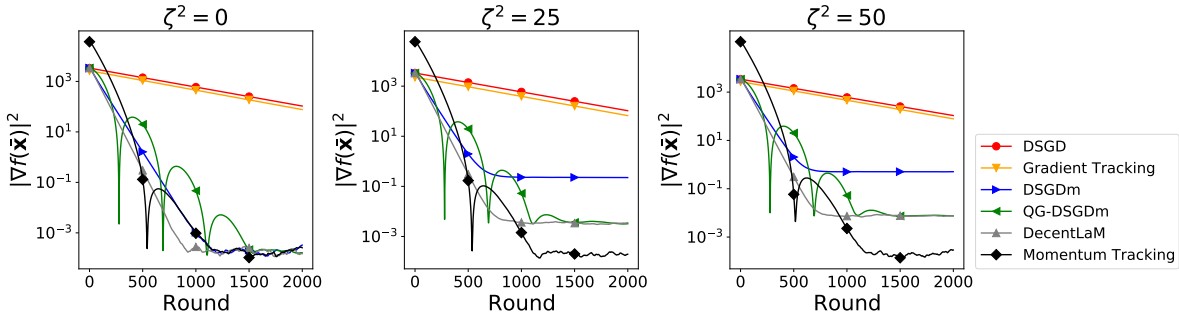

Figure 8: Comparison of the convergence in the initial training phase in the synthetic experiments.

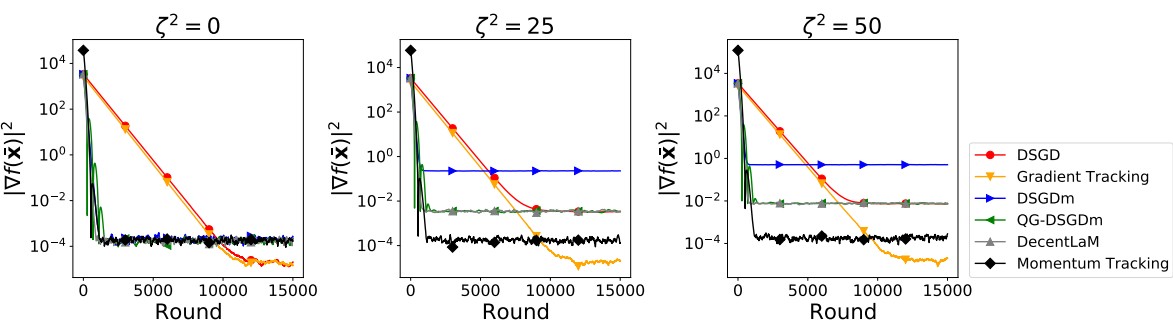

Figure 9: Comparison of the convergence in the synthetic experiments.

# D   Proof of Theorem 1

## D.1   Technical Lemma

**Lemma 1.** *For any $\boldsymbol{x}, \boldsymbol{y} \in \mathbb{R}^d$, $\gamma > 0$, it holds that*

$$\|\boldsymbol{x} + \boldsymbol{y}\|^2 \le (1 + \gamma)\|\boldsymbol{x}\|^2 + (1 + \gamma^{-1})\|\boldsymbol{y}\|^2. \tag{13}$$

**Lemma 2.** *For any $\boldsymbol{a}_1, \cdots, \boldsymbol{a}_N$, it holds that*

$$\left\| \sum_{i=1}^N \boldsymbol{a}_i \right\|^2 \le N \sum_{i=1}^N \|\boldsymbol{a}_i\|^2. \tag{14}$$

**Lemma 3.** *For any $\boldsymbol{x}, \boldsymbol{y} \in \mathbb{R}^d$, $\gamma > 0$, it holds that*

$$2\langle \boldsymbol{x}, \boldsymbol{y} \rangle \le \gamma \|\boldsymbol{x}\|^2 + \gamma^{-1} \|\boldsymbol{y}\|^2. \tag{15}$$

## D.2   Momentum Tracking in Matrix Notation

We define $\boldsymbol{U}^{(r)}$, $\boldsymbol{X}^{(r)}$, $\boldsymbol{C}^{(r)}$, $\nabla F(\boldsymbol{X}^{(r)}; \xi^{(r)})$, and $\nabla f(\boldsymbol{X}^{(r)})$ as follows:

$$\boldsymbol{U}^{(r)} := \left( \boldsymbol{u}_1^{(r)}, \cdots, \boldsymbol{u}_N^{(r)} \right), \quad \boldsymbol{X}^{(r)} := \left( \boldsymbol{x}_1^{(r)}, \cdots, \boldsymbol{x}_N^{(r)} \right), \quad \boldsymbol{C}^{(r)} := \left( \boldsymbol{c}_1^{(r)}, \cdots, \boldsymbol{c}_N^{(r)} \right),$$

$$\nabla F(\boldsymbol{X}^{(r)}; \xi^{(r)}) := \left( \nabla F_1(\boldsymbol{x}_1^{(r)}; \xi_1^{(r)}), \cdots, \nabla F_N(\boldsymbol{x}_N^{(r)}; \xi_N^{(r)}) \right),$$

$$\nabla f(\boldsymbol{X}^{(r)}) := \left( \nabla f_1(\boldsymbol{x}_1^{(r)}), \cdots, \nabla f_N(\boldsymbol{x}_N^{(r)}) \right).$$

Then, the update rule of Momentum Tracking can then be rewritten as follows:

$$\boldsymbol{U}^{(r+1)} = \beta \boldsymbol{U}^{(r)} + \nabla F(\boldsymbol{X}^{(r)}; \xi^{(r)}), \tag{16}$$

$$\boldsymbol{X}^{(r+1)} = \boldsymbol{X}^{(r)} \boldsymbol{W} - \eta(\boldsymbol{U}^{(r+1)} - \boldsymbol{C}^{(r)}), \tag{17}$$

$$\boldsymbol{C}^{(r+1)} = (\boldsymbol{C}^{(r)} - \boldsymbol{U}^{(r+1)}) \boldsymbol{W} + \boldsymbol{U}^{(r+1)}, \tag{18}$$

where $\boldsymbol{U}^{(0)}$ and $\boldsymbol{C}^{(0)}$ are initialized as follows:

$$\boldsymbol{U}^{(0)} = \frac{1}{1 - \beta} (\nabla F(\boldsymbol{X}^{(0)}; \xi^{(0)}) - \frac{1}{N} \nabla F(\boldsymbol{X}^{(0)}; \xi^{(0)}) \boldsymbol{1}\boldsymbol{1}^\top),$$

$$\boldsymbol{C}^{(0)} = \frac{1}{1 - \beta} (\nabla F(\boldsymbol{X}^{(0)}; \xi^{(0)}) - \frac{1}{N} \nabla F(\boldsymbol{X}^{(0)}; \xi^{(0)}) \boldsymbol{1}\boldsymbol{1}^\top).$$

## D.3   Additional Notation

We define the update rules of $\boldsymbol{d}_i$ and $\boldsymbol{e}_i$ as follows:

$$\boldsymbol{d}_i^{(r+1)} = \beta \boldsymbol{d}_i^{(r)} + \nabla f_i(\bar{\boldsymbol{x}}^{(r)}), \tag{19}$$

$$\boldsymbol{e}_i^{(r+1)} = \beta \boldsymbol{e}_i^{(r)} + \nabla f(\bar{\boldsymbol{x}}^{(r)}), \tag{20}$$

where $\boldsymbol{d}_i^{(0)} = \frac{1}{1-\beta}(\nabla f_i(\bar{\boldsymbol{x}}^{(0)}) - \nabla f(\bar{\boldsymbol{x}}^{(0)}))$ and $\boldsymbol{e}_i^{(0)} = \boldsymbol{0}$. Note that it holds that $\bar{\boldsymbol{d}}^{(r)} = \bar{\boldsymbol{e}}^{(r)}$ for any round $r \ge 0$. Then, we define $\boldsymbol{D}$ and $\boldsymbol{E}$ as follows:

$$\boldsymbol{D}^{(r)} := \left( \boldsymbol{d}_1^{(r)}, \cdots, \boldsymbol{d}_N^{(r)} \right), \quad \boldsymbol{E}^{(r)} := \left( \boldsymbol{e}_1^{(r)}, \cdots, \boldsymbol{e}_N^{(r)} \right).$$

The update rules of $\boldsymbol{D}$ and $\boldsymbol{E}$ can be written as follows:

$$\boldsymbol{D}^{(r+1)} = \beta \boldsymbol{D}^{(r)} + \nabla f(\bar{\boldsymbol{X}}^{(r)}), \tag{21}$$

$$\boldsymbol{E}^{(r+1)} = \beta \boldsymbol{E}^{(r)} + \frac{1}{N} \nabla f(\bar{\boldsymbol{X}}^{(r)}) \boldsymbol{1}\boldsymbol{1}^\top, \tag{22}$$

where $\boldsymbol{D}^{(0)}$ and $\boldsymbol{E}^{(0)}$ are initialized as follows:

$$\boldsymbol{D}^{(0)} = \frac{1}{1-\beta}\left(\nabla f(\bar{\boldsymbol{X}}^{(0)}) - \frac{1}{N}\nabla f(\bar{\boldsymbol{X}}^{(0)})\boldsymbol{1}\boldsymbol{1}^{\top}\right),$$

$$\boldsymbol{E}^{(0)} = \boldsymbol{0}.$$

Note that $\boldsymbol{d}_i$, $\boldsymbol{e}_i$, $\boldsymbol{D}$, and $\boldsymbol{E}$ are the only variables used in the proof that do not need to be computed in practice in Alg. 1. We define $\Xi$, $\mathcal{E}$, and $\mathcal{D}$ as follows:

$$\Xi^{(r)} := \frac{1}{N}\mathbb{E}\left\|\boldsymbol{X}^{(r)} - \bar{\boldsymbol{X}}^{(r)}\right\|_F^2,$$

$$\mathcal{E}^{(r)} := \frac{1}{N}\mathbb{E}\left\|\boldsymbol{D}^{(r+1)} - \boldsymbol{C}^{(r)} - \boldsymbol{E}^{(r+1)}\right\|_F^2,$$

$$\mathcal{D}^{(r)} := \frac{1}{N}\mathbb{E}\left\|\boldsymbol{D}^{(r+1)} - \boldsymbol{D}^{(r)} - \boldsymbol{E}^{(r+1)} + \boldsymbol{E}^{(r)}\right\|_F^2.$$

Inspired by Yu et al. (2019), we define $\bar{\boldsymbol{z}}$ as follows:

$$\bar{\boldsymbol{z}}^{(r)} := \begin{cases} \bar{\boldsymbol{x}}^{(r)}, & \text{if } r = 0 \\ \frac{1}{1-\beta}\bar{\boldsymbol{x}}^{(r)} - \frac{\beta}{1-\beta}\bar{\boldsymbol{x}}^{(r-1)}, & \text{otherwise} \end{cases}.$$

In the following, we define $\pm a := a - a = 0$ for any $a$ and $\bar{a} := \frac{1}{N}\sum_{i=1}^N a_i$ for any $a_1, \cdots, a_N$. Then, $\mathbb{E}[\cdot]$ denotes the expectation over all randomness that occurs during training (i.e., $\{\xi_i^{(r)}\}_{i,r}$), and $\mathbb{E}_r[\cdot]$ denotes the expectation over the randomness that occurs at round $r$ (i.e., $\{\xi_i^{(r)}\}_i$).

### D.4 Proof Sketch

In this section, we briefly summarize our proof technique. Our proof is based on the analysis of DSGD Koloskova et al. (2020) that uses the upper bound of the consensus error $\Xi$. We extend their technique to deal with data heterogeneity by decomposing the inequality about the consensus error $\Xi$ into (i) the inequality of the error between global and corrected local momentum $\mathcal{E}$ and (ii) the inequality of the error between update values of global and uncorrected local momentum $\mathcal{D}$. While bounding the latter error is rather straightforward, bounding the former error requires our momentum correction term, which makes the error recursion contractive; otherwise, the error between global and local momentum results in the heterogeneity term in the final convergence error. In the following, we show the proof sketch and explain in more detail how to bound $\Xi$ from above without using $\zeta^2$.

We derive the inequality about the consensus error $\Xi$ as follows (See Lemma 14):

$$\Xi^{(r+1)} \le \left(1 - \frac{p}{2}\right)\Xi^{(r)} + \frac{9}{p}\eta^2\mathcal{E}^{(r)} + \frac{9}{Np}\eta^2\sum_{i=1}^N\left(\mathbb{E}\left\|\mathbf{u}_i^{(r+1)} - \mathbf{d}_i^{(r+1)}\right\|^2 + \mathbb{E}\left\|\mathbf{e}_i^{(r+1)}\right\|^2\right)$$

$\mathcal{E}$ represents the the error between global momentum $\mathbf{e}_i(= \bar{\mathbf{e}})$ and corrected local momentum $(\mathbf{d}_i - \mathbf{c}_i)$. Thus, intuitively, if we remove the tracking term $\mathbf{c}_i$ from Momentum Tracking, $\mathcal{E}$ becomes $\frac{1}{N}\sum_i\mathbb{E}\|\mathbf{d}_i - \mathbf{e}_i\|^2$, which causes the data heterogeneity $\zeta^2$ to appear in the upper bound of $\Xi$. In the following, we explain how to eliminate $\mathcal{E}^{(r)}$ and $\zeta^2$ from the upper bound of $\Xi^{(r+1)}$, which is the most important component of our proof.

To bound $\mathcal{E}$ from above, we derive the following two inequalities (see Lemmas 15 and 16):

$$\mathcal{E}^{(r+1)} \le \left(1 - \frac{p}{2}\right)\mathcal{E}^{(r)} + \frac{18\beta^2}{p}\mathcal{D}^{(r)} + \frac{144L^4}{p}\eta^2\Xi^{(r)} + C_1,$$

$$\mathcal{D}^{(r+1)} \le \frac{2\beta^2}{1+\beta^2}\mathcal{D}^{(r)} + \frac{32L^4\eta^2}{1-\beta^2}\Xi^{(r)} + C_2,$$

where $C_1$ and $C_2$ are variables independent of $\Xi$, $\mathcal{D}$, $\mathcal{E}$, and $\zeta^2$. Here, the most important benefit to adding the tracking term $\mathbf{c}_i$ is that the coefficient of $\mathcal{E}^{(r)}$ becomes less than 1. (i.e., $(1 - \frac{p}{2}) < 1$). Roughly speaking,

the above two inequalities imply that $\mathcal{E}$ and $\mathcal{D}$ become gradually smaller because $(1 - \frac{p}{2}) < 1$ and $\frac{2\beta^2}{1+\beta^2} < 1$ hold for any $\beta \in [0, 1)$.

Next, we derive a new inequality by combining the above three inequalities. We define an auxiliary error term $\Theta$ as follows:

$$\Theta^{(r)} := \Xi^{(r)} + \frac{36}{p^2}\eta^2\mathcal{E}^{(r)} + \frac{A\beta^2}{p^3}\eta^2\mathcal{D}^{(r)},$$

where $A > 0$ is defined in Lemma 17. Then, by combining the above three inequalities, we obtain the following (see Lemma 17):

$$\Theta^{(r+1)} \leq \left(1 - \frac{p}{t}\right)\Theta^{(r)} + C_3,$$

where $C_3$ is a variable independent of $\Xi$, $\mathcal{D}$, $\mathcal{E}$ and $\zeta^2$, and $t \geq 4$ is defined in Lemma 17.

Using $\Xi \leq \Theta$ and applying the above inequality recursively, we obtain (see Lemma 18)

$$\Xi^{(r+1)} \leq \Theta^{(r+1)} \leq \left(1 - \frac{p}{t}\right)^{r+1}\Theta^{(0)} + C_4,$$

where $C_4$ is a variable independent of $\Xi$, $\mathcal{D}$, $\mathcal{E}$, and $\zeta^2$.

Using $(1 - \frac{p}{t}) < 1$, it holds that $\sum_{r=0}^{R}(1 - \frac{p}{t})^{r+1}\Theta^{(0)} \leq \frac{t}{p}\Theta^{(0)}$. Finally, using this inequality and the fact that $C_4$ and $\Theta^{(0)}$ are independent of $\zeta^2$ due to the assumption of initial values, we can eliminate $\mathcal{E}$ from the upper bound of $\Xi$ and derive the upper bound of $\Xi$ that does not contain $\zeta^2$ as follows (see Lemma 19):

$$\frac{4L^2}{R+1}\sum_{r=0}^{R}\Xi^{(r)} \leq \frac{1}{2(R+1)}\sum_{k=0}^{R}\left\|\nabla f(\bar{\mathbf{x}}^{(k)})\right\|^2 + \frac{40L^2t}{(1-\beta)^3p^2}\left(10 + \frac{29}{p} + \frac{864}{p^2}\right)\sigma^2\eta^2.$$

These are the essential techniques to bound $\sum_{r=0}^{R}\Xi^{(r)}$ from above without using $\zeta^2$ and make the convergence rate independent of $\zeta^2$.

### D.5 Useful Lemma

**Lemma 4.** *For any round $r \geq 0$, it holds that $\bar{\boldsymbol{c}}^{(r)} = \boldsymbol{0}$.*

*Proof.* For any round $r \geq 0$, we have

$$\sum_{i=1}^{N} \boldsymbol{c}_i^{(r+1)} = \sum_{i=1}^{N} \sum_{j=1}^{N} W_{ij}(\boldsymbol{c}_j^{(r)} - \boldsymbol{u}_j^{(r+1)}) + \sum_{i=1}^{N} \boldsymbol{u}_i^{(r+1)}$$

$$= \sum_{j=1}^{N} (\boldsymbol{c}_j^{(r)} - \boldsymbol{u}_j^{(r+1)}) \sum_{i=1}^{N} W_{ij} + \sum_{i=1}^{N} \boldsymbol{u}_i^{(r+1)}.$$

Because $\boldsymbol{W}$ is a mixing matrix, we obtain

$$\sum_{i=1}^{N} \boldsymbol{c}_i^{(r+1)} = \sum_{j=1}^{N} \boldsymbol{c}_j^{(r)}.$$

Since we have

$$\sum_{i=1}^{N} \boldsymbol{c}_i^{(0)} = \frac{1}{1-\beta} \sum_{i=1}^{N} \left( \nabla F_i(\boldsymbol{x}_i^{(0)}; \xi_i^{(0)}) - \frac{1}{N} \sum_{j=1}^{N} \nabla F_j(\boldsymbol{x}_j^{(0)}; \xi_j^{(0)}) \right) = \boldsymbol{0},$$

we obtain the statement. $\qquad\square$

**Lemma 5.** *For any round $r \geq 0$, it holds that*

$$\bar{\boldsymbol{x}}^{(r+1)} = \bar{\boldsymbol{x}}^{(r)} - \eta \bar{\boldsymbol{u}}^{(r+1)}.$$

*Proof.* We have

$$\bar{\boldsymbol{x}}^{(r+1)} = \frac{1}{N} \sum_{i=1}^{N} \sum_{j=1}^{N} W_{ij} \boldsymbol{x}_j^{(r)} - \eta(\bar{\boldsymbol{u}}^{(r+1)} - \bar{\boldsymbol{c}}^{(r)})$$

$$= \frac{1}{N} \sum_{j=1}^{N} \boldsymbol{x}_j^{(r)} \sum_{i=1}^{N} W_{ij} - \eta(\bar{\boldsymbol{u}}^{(r+1)} - \bar{\boldsymbol{c}}^{(r)}).$$

The fact that $\boldsymbol{W}$ is a mixing matrix gives us

$$\bar{\boldsymbol{x}}^{(r+1)} = \bar{\boldsymbol{x}}^{(r)} - \eta(\bar{\boldsymbol{u}}^{(r+1)} - \bar{\boldsymbol{c}}^{(r)}).$$

Then, using Lemma 4, we get the statement. $\qquad\square$

**Lemma 6.** *For any round $r \geq 0$, it holds that*

$$\bar{\boldsymbol{z}}^{(r+1)} - \bar{\boldsymbol{z}}^{(r)} = -\frac{\eta}{1-\beta} \frac{1}{N} \sum_{i=1}^{N} \nabla F_i(\boldsymbol{x}_i^{(r)}; \xi_i^{(r)}).$$

*Proof.* For any $r \geq 1$, we have

$$\bar{\boldsymbol{z}}^{(r+1)} - \bar{\boldsymbol{z}}^{(r)} = \frac{1}{1-\beta}(\bar{\boldsymbol{x}}^{(r+1)} - \bar{\boldsymbol{x}}^{(r)}) - \frac{\beta}{1-\beta}(\bar{\boldsymbol{x}}^{(r)} - \bar{\boldsymbol{x}}^{(r-1)})$$

$$= -\frac{\eta}{1-\beta} \bar{\boldsymbol{u}}^{(r+1)} + \frac{\eta\beta}{1-\beta} \bar{\boldsymbol{u}}^{(r)}$$

$$= -\frac{\eta}{1-\beta} \frac{1}{N} \sum_{i=1}^{N} \nabla F_i(\boldsymbol{x}_i^{(r)}; \xi_i^{(r)}),$$

where we use Lemma 5. When $r = 0$, we have

$$
\begin{aligned}
\bar{\boldsymbol{z}}^{(1)} - \bar{\boldsymbol{z}}^{(0)} &= \frac{1}{1-\beta}(\bar{\boldsymbol{x}}^{(1)} - \bar{\boldsymbol{x}}^{(0)}) \\
&= -\frac{\eta}{1-\beta}\bar{\boldsymbol{u}}^{(1)} \\
&= -\frac{\eta}{1-\beta}\frac{1}{N}\sum_{i=1}^{N}\nabla F_i(\boldsymbol{x}_i^{(0)};\xi_i^{(0)}),
\end{aligned}
$$

where we use $\bar{\boldsymbol{u}}^{(0)} = \boldsymbol{0}$. This concludes the proof. $\qquad\square$

**Lemma 7.** *Suppose that Assumptions 1, 2, 3, and 4 hold. For any round $R \geq 0$, it holds that*

$$
\sum_{r=0}^{R}\mathbb{E}\left\|\bar{\boldsymbol{x}}^{(r)} - \bar{\boldsymbol{z}}^{(r)}\right\|^2 \leq \frac{\beta^2\eta^2}{(1-\beta)^4}\sum_{r=0}^{R}\mathbb{E}\left\|\frac{1}{N}\sum_{i=1}^{N}\nabla f_i(\boldsymbol{x}_i^{(r)})\right\|^2 + \frac{\beta^2\sigma^2\eta^2}{N(1-\beta)^4}R.
$$

*Proof.* From Lemma 5, we have

$$
\begin{aligned}
\mathbb{E}\left\|\bar{\boldsymbol{x}}^{(r)} - \bar{\boldsymbol{z}}^{(r)}\right\|^2 &= \mathbb{E}\left\|\frac{\eta\beta}{1-\beta}\bar{\boldsymbol{u}}^{(r)}\right\|^2 \\
&= \frac{\beta^2\eta^2}{(1-\beta)^2}\mathbb{E}\left\|\sum_{k=0}^{r-1}\beta^{r-k-1}\frac{1}{N}\sum_{i=1}^{N}\nabla F_i(\boldsymbol{x}_i^{(k)};\xi_i^{(k)})\right\|^2,
\end{aligned}
$$

for any $r \geq 1$. Defining $s^{(r)} := \sum_{k=0}^{r}\beta^{r-k}$, we obtain

$$
\begin{aligned}
&\mathbb{E}\left\|\bar{\boldsymbol{x}}^{(r)} - \bar{\boldsymbol{z}}^{(r)}\right\|^2 \\
&= \frac{\beta^2\eta^2}{(1-\beta)^2}s^{(r-1)2}\mathbb{E}\left\|\sum_{k=0}^{r-1}\frac{\beta^{r-k-1}}{s^{(r-1)}}\frac{1}{N}\sum_{i=1}^{N}\nabla F_i(\boldsymbol{x}_i^{(k)};\xi_i^{(k)})\right\|^2 \\
&\overset{(a)}{\leq} \frac{\beta^2\eta^2}{(1-\beta)^2}s^{(r-1)}\sum_{k=0}^{r-1}\beta^{r-k-1}\mathbb{E}\left\|\frac{1}{N}\sum_{i=1}^{N}\nabla F_i(\boldsymbol{x}_i^{(k)};\xi_i^{(k)})\right\|^2 \\
&\overset{(6)}{\leq} \frac{\beta^2\eta^2}{(1-\beta)^2}s^{(r-1)}\sum_{k=0}^{r-1}\beta^{r-k-1}\mathbb{E}\left\|\frac{1}{N}\sum_{i=1}^{N}\nabla f_i(\boldsymbol{x}_i^{(k)})\right\|^2 + \frac{\beta^2\sigma^2\eta^2}{N(1-\beta)^2}s^{(r-1)}\sum_{k=0}^{r-1}\beta^{r-k-1},
\end{aligned}
$$

where we use Jensen's inequality in (a). Using $s^{(r-1)} \leq \frac{1}{1-\beta}$, we obtain

$$
\mathbb{E}\left\|\bar{\boldsymbol{x}}^{(r)} - \bar{\boldsymbol{z}}^{(r)}\right\|^2 \leq \frac{\beta^2\eta^2}{(1-\beta)^3}\sum_{k=0}^{r-1}\beta^{r-k-1}\mathbb{E}\left\|\frac{1}{N}\sum_{i=1}^{N}\nabla f_i(\boldsymbol{x}_i^{(k)})\right\|^2 + \frac{\beta^2\sigma^2\eta^2}{N(1-\beta)^4}.
$$

Recursive addition yields

$$
\begin{aligned}
\sum_{r=1}^{R}\mathbb{E}\left\|\bar{\boldsymbol{x}}^{(r)} - \bar{\boldsymbol{z}}^{(r)}\right\|^2 &\leq \frac{\beta^2\eta^2}{(1-\beta)^3}\sum_{r=1}^{R}\sum_{k=0}^{r-1}\beta^{r-k-1}\mathbb{E}\left\|\frac{1}{N}\sum_{i=1}^{N}\nabla f_i(\boldsymbol{x}_i^{(k)})\right\|^2 + \frac{\beta^2\sigma^2\eta^2}{N(1-\beta)^4}R \\
&= \frac{\beta^2\eta^2}{(1-\beta)^3}\sum_{k=0}^{R-1}\mathbb{E}\left\|\frac{1}{N}\sum_{i=1}^{N}\nabla f_i(\boldsymbol{x}_i^{(k)})\right\|^2\sum_{r=k+1}^{R}\beta^{r-k-1} + \frac{\beta^2\sigma^2\eta^2}{N(1-\beta)^4}R \\
&\leq \frac{\beta^2\eta^2}{(1-\beta)^4}\sum_{k=0}^{R-1}\mathbb{E}\left\|\frac{1}{N}\sum_{i=1}^{N}\nabla f_i(\boldsymbol{x}_i^{(k)})\right\|^2 + \frac{\beta^2\sigma^2\eta^2}{N(1-\beta)^4}R,
\end{aligned}
$$

where we use $\sum_{r=k+1}^{R} \beta^{r-k-1} \leq \frac{1}{1-\beta}$ in the last inequality. From the definition of $\bar{z}^{(0)}$, we have $\|\bar{x}^{(0)} - \bar{z}^{(0)}\|^2 = 0$. This yields the statement. □

**Lemma 8.** *Suppose that Assumptions 1, 2, 3, and 4 hold. For any round $r \geq 0$, it holds that*

$$\mathbb{E}\left\|\bar{x}^{(r+1)} - \bar{x}^{(r)}\right\|^2 \leq 4L^2\eta^2\Xi^{(r)} + 2\beta^2\eta^2\mathbb{E}\left\|\bar{u}^{(r)}\right\|^2 + 4\eta^2\mathbb{E}\left\|\nabla f(\bar{x}^{(r)})\right\|^2 + \frac{\sigma^2\eta^2}{N}.$$

*Proof.* From Lemma 5, we have

$$\mathbb{E}_r\left\|\bar{x}^{(r+1)} - \bar{x}^{(r)}\right\|^2 = \eta^2\mathbb{E}_r\left\|\beta\bar{u}^{(r)} + \frac{1}{N}\sum_{i=1}^{N}\nabla F_i(x_i^{(r)}; \xi_i^{(r)})\right\|^2$$

$$\overset{(6)}{\leq} \eta^2\left\|\beta\bar{u}^{(r)} + \frac{1}{N}\sum_{i=1}^{N}\nabla f_i(x_i^{(r)})\right\|^2 + \frac{\sigma^2\eta^2}{N}$$

$$\overset{(13)}{\leq} 2\beta^2\eta^2\left\|\bar{u}^{(r)}\right\|^2 + 2\eta^2\underbrace{\left\|\frac{1}{N}\sum_{i=1}^{N}\nabla f_i(x_i^{(r)})\right\|^2}_{T} + \frac{\sigma^2\eta^2}{N}.$$

Then, $T$ can be bounded from above as follows:

$$T = \left\|\frac{1}{N}\sum_{i=1}^{N}\nabla f_i(x_i^{(r)}) \pm \nabla f_i(\bar{x}^{(r)})\right\|^2$$

$$\overset{(13)}{\leq} 2\left\|\frac{1}{N}\sum_{i=1}^{N}\nabla f_i(x_i^{(r)}) - \nabla f_i(\bar{x}^{(r)})\right\|^2 + 2\left\|\nabla f(\bar{x}^{(r)})\right\|^2$$

$$\overset{(14)}{\leq} \frac{2}{N}\sum_{i=1}^{N}\left\|\nabla f_i(x_i^{(r)}) - \nabla f_i(\bar{x}^{(r)})\right\|^2 + 2\left\|\nabla f(\bar{x}^{(r)})\right\|^2$$

$$\overset{(5)}{\leq} \frac{2L^2}{N}\sum_{i=1}^{N}\left\|x_i^{(r)} - \bar{x}^{(r)}\right\|^2 + 2\left\|\nabla f(\bar{x}^{(r)})\right\|^2.$$

Then, we obtain the statement. □

**Lemma 9.** *For any round $r \geq 0$, it holds that*

$$\mathbb{E}\left\|\bar{e}^{(r+1)}\right\|^2 \leq \frac{1}{1-\beta}\sum_{k=0}^{r}\beta^{r-k}\mathbb{E}\left\|\nabla f(\bar{x}^{(k)})\right\|^2.$$

*Proof.* We have

$$\mathbb{E}\left\|\bar{e}^{(r+1)}\right\|^2 = \mathbb{E}\left\|\sum_{k=0}^{r}\beta^{r-k}\nabla f(\bar{x}^{(k)})\right\|^2,$$

where we use $\bar{e}^{(0)} = \mathbf{0}$. Defining $s^{(r)} := \sum_{k=0}^{r}\beta^{r-k}$, we obtain

$$\mathbb{E}\left\|\bar{e}^{(r+1)}\right\|^2 = s^{(r)2}\mathbb{E}\left\|\sum_{k=0}^{r}\frac{\beta^{r-k}}{s^{(r)}}\nabla f(\bar{x}^{(k)})\right\|^2$$

$$\leq s^{(r)}\sum_{k=0}^{r}\beta^{r-k}\mathbb{E}\left\|\nabla f(\bar{x}^{(k)})\right\|^2,$$

where we use Jensen's inequality. Using $s^{(r)} \leq \frac{1}{1-\beta}$, we obtain the statement. □

**Lemma 10.** *Suppose that Assumptions 1, 2, 3, and 4 hold. For any round $r \geq 0$, it holds that*

$$\frac{1}{N}\mathbb{E}\left\|\boldsymbol{D}^{(r+1)} - \boldsymbol{U}^{(r+1)}\right\|_F^2 \leq \frac{L^2}{1-\beta}\sum_{k=0}^{r}\beta^{r-k}\Xi^{(k)} + \frac{5\sigma^2}{(1-\beta)^3}.$$

*Proof.* We have

$$\mathbb{E}\left\|\boldsymbol{D}^{(r+1)} - \boldsymbol{U}^{(r+1)}\right\|_F^2$$
$$= \mathbb{E}\left\|\sum_{k=0}^{r}\beta^{r-k}(\nabla f(\bar{\boldsymbol{X}}^{(k)}) - \nabla F(\boldsymbol{X}^{(k)};\xi^{(k)})) + \beta^{r+1}(\boldsymbol{D}^{(0)} - \boldsymbol{U}^{(0)})\right\|_F^2.$$

Defining $s^{(r)} := \sum_{k=0}^{r}\beta^{r-k}$, we obtain

$$\mathbb{E}\left\|\boldsymbol{D}^{(r+1)} - \boldsymbol{U}^{(r+1)}\right\|_F^2$$
$$= s^{(r+1)^2}\mathbb{E}\left\|\sum_{k=0}^{r}\frac{\beta^{r-k}}{s^{(r+1)}}(\nabla f(\bar{\boldsymbol{X}}^{(k)}) - \nabla F(\boldsymbol{X}^{(k)};\xi^{(k)})) + \frac{\beta^{r+1}}{s^{(r+1)}}(\boldsymbol{D}^{(0)} - \boldsymbol{U}^{(0)})\right\|_F^2$$
$$\overset{(a)}{\leq} s^{(r+1)}\sum_{k=0}^{r}\beta^{r-k}\mathbb{E}\left\|\nabla f(\bar{\boldsymbol{X}}^{(k)}) - \nabla F(\boldsymbol{X}^{(k)};\xi^{(k)})\right\|_F^2 + s^{(r+1)}\beta^{r+1}\mathbb{E}\left\|\boldsymbol{D}^{(0)} - \boldsymbol{U}^{(0)}\right\|_F^2$$
$$\overset{(14)}{\leq} s^{(r+1)}\sum_{k=0}^{r}\beta^{r-k}\mathbb{E}\left\|\nabla f(\bar{\boldsymbol{X}}^{(k)}) - \nabla F(\boldsymbol{X}^{(k)};\xi^{(k)})\right\|_F^2$$
$$+ \frac{2s^{(r+1)}}{(1-\beta)^2}\beta^{r+1}\mathbb{E}\left\|\nabla f(\bar{\boldsymbol{X}}^{(0)}) - \nabla F(\boldsymbol{X}^{(0)};\xi^{(0)})\right\|_F^2$$
$$+ \frac{2s^{(r+1)}}{(1-\beta)^2}\beta^{r+1}\mathbb{E}\left\|\frac{1}{N}\nabla f(\bar{\boldsymbol{X}}^{(0)})\mathbf{1}\mathbf{1}^\top - \frac{1}{N}\nabla F(\boldsymbol{X}^{(0)};\xi^{(0)})\mathbf{1}\mathbf{1}^\top\right\|_F^2$$
$$\overset{(6)}{\leq} s^{(r+1)}\sum_{k=0}^{r}\beta^{r-k}\mathbb{E}\left\|\nabla f(\bar{\boldsymbol{X}}^{(k)}) - \nabla f(\boldsymbol{X}^{(k)})\right\|_F^2 + s^{(r+1)}\sum_{k=0}^{r}\beta^{r-k}N\sigma^2 + \frac{4s^{(r+1)}}{(1-\beta)^2}\beta^{r+1}N\sigma^2,$$

where we use Jensen's inequality for (a) and use $\boldsymbol{X}^{(0)} = \bar{\boldsymbol{X}}^{(0)}$ for the last inequality. Then, using $s^{(r)} \leq \frac{1}{1-\beta}$, we obtain

$$\mathbb{E}\left\|\boldsymbol{D}^{(r+1)} - \boldsymbol{U}^{(r+1)}\right\|_F^2$$
$$\leq \frac{1}{1-\beta}\sum_{k=0}^{r}\beta^{r-k}\mathbb{E}\left\|\nabla f(\bar{\boldsymbol{X}}^{(k)}) - \nabla f(\boldsymbol{X}^{(k)})\right\|_F^2 + \frac{N\sigma^2}{(1-\beta)^2} + \frac{4N\sigma^2}{(1-\beta)^3}\beta^{r+1}$$
$$\overset{\beta\in[0,1)}{\leq} \frac{1}{1-\beta}\sum_{k=0}^{r}\beta^{r-k}\mathbb{E}\left\|\nabla f(\bar{\boldsymbol{X}}^{(k)}) - \nabla f(\boldsymbol{X}^{(k)})\right\|_F^2 + \frac{5N\sigma^2}{(1-\beta)^3}$$
$$\overset{(5)}{\leq} \frac{L^2}{1-\beta}\sum_{k=0}^{r}\beta^{r-k}\mathbb{E}\left\|\bar{\boldsymbol{X}}^{(k)} - \boldsymbol{X}^{(k)}\right\|_F^2 + \frac{5N\sigma^2}{(1-\beta)^3}.$$

This concludes the proof. $\square$

**Lemma 11.** *Suppose that Assumptions 1, 2, 3, and 4 hold. For any round $r \geq 0$, it holds that*

$$\mathbb{E}\left\|\bar{\boldsymbol{u}}^{(r+1)} - \bar{\boldsymbol{d}}^{(r+1)}\right\|^2 \leq \frac{L^2}{1-\beta}\sum_{k=0}^{r}\beta^{r-k}\Xi^{(k)} + \frac{\sigma^2}{N(1-\beta)^2}$$

*Proof.* We have

$$\mathbb{E}\left\|\bar{\boldsymbol{u}}^{(r+1)} - \bar{\boldsymbol{d}}^{(r+1)}\right\|^2 = \mathbb{E}\left\|\sum_{k=0}^{r} \beta^{r-k}(\frac{1}{N}\sum_{i=1}^{N}\nabla F_i(\boldsymbol{x}_i^{(k)};\xi_i^{(k)}) - \nabla f(\bar{\boldsymbol{x}}^{(k)}))\right\|^2,$$

where we use $\bar{\boldsymbol{u}}^{(0)} = \bar{\boldsymbol{d}}^{(0)} = \boldsymbol{0}$. Defining $s^{(r)} := \sum_{k=0}^{r} \beta^{r-k}$, we obtain

$$\mathbb{E}\left\|\bar{\boldsymbol{u}}^{(r+1)} - \bar{\boldsymbol{d}}^{(r+1)}\right\|^2$$

$$= s^{(r)2}\mathbb{E}\left\|\sum_{k=0}^{r} \frac{\beta^{r-k}}{s^{(r)}}(\frac{1}{N}\sum_{i=1}^{N}\nabla F_i(\boldsymbol{x}_i^{(k)};\xi_i^{(k)}) - \nabla f(\bar{\boldsymbol{x}}^{(k)}))\right\|^2$$

$$\overset{(a)}{\leq} s^{(r)}\sum_{k=0}^{r} \beta^{r-k}\mathbb{E}\left\|\frac{1}{N}\sum_{i=1}^{N}\nabla F_i(\boldsymbol{x}_i^{(k)};\xi_i^{(k)}) - \nabla f(\bar{\boldsymbol{x}}^{(k)})\right\|^2$$

$$\overset{(6)}{\leq} s^{(r)}\sum_{k=0}^{r} \beta^{r-k}\mathbb{E}\left\|\frac{1}{N}\sum_{i=1}^{N}\nabla f_i(\boldsymbol{x}_i^{(k)}) - \nabla f(\bar{\boldsymbol{x}}^{(k)})\right\|^2 + s^{(r)}\sum_{k=0}^{r} \beta^{r-k}\frac{\sigma^2}{N}$$

$$\overset{(14)}{\leq} s^{(r)}\sum_{k=0}^{r} \beta^{r-k}\frac{1}{N}\sum_{i=1}^{N}\mathbb{E}\left\|\nabla f_i(\boldsymbol{x}_i^{(k)}) - \nabla f_i(\bar{\boldsymbol{x}}^{(k)})\right\|^2 + s^{(r)}\sum_{k=0}^{r} \beta^{r-k}\frac{\sigma^2}{N},$$

where we use Jensen's inequality in (a). Then, using $s^{(r)} \leq \frac{1}{1-\beta}$, we obtain

$$\mathbb{E}\left\|\bar{\boldsymbol{u}}^{(r+1)} - \bar{\boldsymbol{d}}^{(r+1)}\right\|^2$$

$$\leq \frac{1}{1-\beta}\sum_{k=0}^{r} \beta^{r-k}\frac{1}{N}\sum_{i=1}^{N}\mathbb{E}\left\|\nabla f_i(\boldsymbol{x}_i^{(k)}) - \nabla f_i(\bar{\boldsymbol{x}}^{(k)})\right\|^2 + \frac{\sigma^2}{N(1-\beta)^2}$$

$$\overset{(5)}{\leq} \frac{L^2}{1-\beta}\sum_{k=0}^{r} \beta^{r-k}\frac{1}{N}\sum_{i=1}^{N}\mathbb{E}\left\|\boldsymbol{x}_i^{(k)} - \bar{\boldsymbol{x}}^{(k)}\right\|^2 + \frac{\sigma^2}{N(1-\beta)^2}.$$

This concludes the proof. $\qquad\square$

**Lemma 12.** *Suppose that Assumptions 1, 2, 3, and 4 hold. For any round $r \geq 0$, it holds that*

$$\mathbb{E}\left\|\bar{\boldsymbol{u}}^{(r+1)}\right\|^2 \leq \frac{2L^2}{1-\beta}\sum_{k=0}^{r} \beta^{r-k}\Xi^{(k)} + \frac{2}{1-\beta}\sum_{k=0}^{r} \beta^{r-k}\left\|\nabla f(\bar{\boldsymbol{x}}^{(k)})\right\|^2 + \frac{2\sigma^2}{N(1-\beta)^2}.$$

*Proof.* We have

$$\mathbb{E}\left\|\bar{\boldsymbol{u}}^{(r+1)}\right\|^2 = \mathbb{E}\left\|\bar{\boldsymbol{u}}^{(r+1)} \pm \bar{\boldsymbol{d}}^{(r+1)}\right\|^2$$

$$\overset{(13)}{\leq} 2\mathbb{E}\left\|\bar{\boldsymbol{u}}^{(r+1)} - \bar{\boldsymbol{d}}^{(r+1)}\right\|^2 + 2\mathbb{E}\left\|\bar{\boldsymbol{d}}^{(r+1)}\right\|^2.$$

From Lemmas 9 and 11, we obtain the statement. $\qquad\square$

### D.6  Main Proof

**Lemma 13** (Descent Lemma)**.** *Suppose that Assumptions 1, 2, 3, and 4 hold. If the step size $\eta$ satisfies*

$$\eta \leq \frac{1-\beta}{4L},$$

*then it holds that for any round $r \geq 0$,*

$$\mathbb{E}f(\bar{\boldsymbol{z}}^{(r+1)}) \leq \mathbb{E}f(\bar{\boldsymbol{z}}^{(r)}) + \frac{L^2\eta}{1-\beta}\mathbb{E}\left\|\bar{\boldsymbol{x}}^{(r)} - \bar{\boldsymbol{z}}^{(r)}\right\|^2 + \frac{L^2\eta}{1-\beta}\Xi^{(r)}$$

$$- \frac{\eta}{4(1-\beta)}\mathbb{E}\left\|\frac{1}{N}\sum_{i=1}^{N}\nabla f_i(\boldsymbol{x}_i^{(r)})\right\|^2 - \frac{\eta}{4(1-\beta)}\mathbb{E}\left\|\nabla f(\bar{\boldsymbol{x}}^{(r)})\right\|^2 + \frac{L\sigma^2\eta^2}{2N(1-\beta)^2}.$$

*Proof.* From Assumption 3 and Lemma 6, we have

$$\mathbb{E}_r f(\bar{\boldsymbol{z}}^{(r+1)})$$

$$\leq f(\bar{\boldsymbol{z}}^{(r)}) + \mathbb{E}_r\langle\nabla f(\bar{\boldsymbol{z}}^{(r)}), \bar{\boldsymbol{z}}^{(r+1)} - \bar{\boldsymbol{z}}^{(r)}\rangle + \frac{L}{2}\mathbb{E}_r\left\|\bar{\boldsymbol{z}}^{(r+1)} - \bar{\boldsymbol{z}}^{(r)}\right\|^2$$

$$= f(\bar{\boldsymbol{z}}^{(r)}) - \frac{\eta}{1-\beta}\left\langle\nabla f(\bar{\boldsymbol{z}}^{(r)}), \frac{1}{N}\sum_{i=1}^{N}\nabla f_i(\boldsymbol{x}_i^{(r)})\right\rangle + \frac{L\eta^2}{2(1-\beta)^2}\mathbb{E}_r\left\|\frac{1}{N}\sum_{i=1}^{N}\nabla F_i(\boldsymbol{x}_i^{(r)}; \xi_i^{(r)})\right\|^2$$

$$\overset{(6)}{\leq} f(\bar{\boldsymbol{z}}^{(r)}) - \frac{\eta}{1-\beta}\left\langle\nabla f(\bar{\boldsymbol{z}}^{(r)}), \frac{1}{N}\sum_{i=1}^{N}\nabla f_i(\boldsymbol{x}_i^{(r)})\right\rangle$$

$$+ \frac{L\eta^2}{2(1-\beta)^2}\left\|\frac{1}{N}\sum_{i=1}^{N}\nabla f_i(\boldsymbol{x}_i^{(r)})\right\|^2 + \frac{L\sigma^2\eta^2}{2N(1-\beta)^2}$$

$$= f(\bar{\boldsymbol{z}}^{(r)}) + \frac{\eta}{1-\beta}\underbrace{\left\langle\nabla f(\bar{\boldsymbol{x}}^{(r)}) - \nabla f(\bar{\boldsymbol{z}}^{(r)}), \frac{1}{N}\sum_{i=1}^{N}\nabla f_i(\boldsymbol{x}_i^{(r)})\right\rangle}_{T_1}$$

$$- \frac{\eta}{1-\beta}\underbrace{\left\langle\nabla f(\bar{\boldsymbol{x}}^{(r)}), \frac{1}{N}\sum_{i=1}^{N}\nabla f_i(\boldsymbol{x}_i^{(r)})\right\rangle}_{T_2} + \frac{L\eta^2}{2(1-\beta)^2}\underbrace{\left\|\frac{1}{N}\sum_{i=1}^{N}\nabla f_i(\boldsymbol{x}_i^{(r)})\right\|^2}_{T_3} + \frac{L\sigma^2\eta^2}{2N(1-\beta)^2}.$$

We can bound $T_1$ from above as follows:

$$T_1 \overset{(15),\gamma=2}{\leq} \left\|\nabla f(\bar{\boldsymbol{x}}^{(r)}) - \nabla f(\bar{\boldsymbol{z}}^{(r)})\right\|^2 + \frac{1}{4}\left\|\frac{1}{N}\sum_{i=1}^{N}\nabla f_i(\boldsymbol{x}_i^{(r)})\right\|^2$$

$$\overset{(5)}{\leq} L^2\left\|\bar{\boldsymbol{x}}^{(r)} - \bar{\boldsymbol{z}}^{(r)}\right\|^2 + \frac{1}{4}\left\|\frac{1}{N}\sum_{i=1}^{N}\nabla f_i(\boldsymbol{x}_i^{(r)})\right\|^2.$$

We can bound $-T_2$ from above as follows:

$$-T_2 = \frac{1}{2}\left\|\nabla f(\bar{\boldsymbol{x}}^{(r)}) - \frac{1}{N}\sum_{i=1}^{N}\nabla f_i(\boldsymbol{x}_i^{(r)})\right\|^2 - \frac{1}{2}\left\|\nabla f(\bar{\boldsymbol{x}}^{(r)})\right\|^2 - \frac{1}{2}\left\|\frac{1}{N}\sum_{i=1}^{N}\nabla f_i(\boldsymbol{x}_i^{(r)})\right\|^2$$

$$\overset{(14)}{\leq} \frac{1}{2}\frac{1}{N}\sum_{i=1}^{N}\left\|\nabla f_i(\bar{\boldsymbol{x}}^{(r)}) - \nabla f_i(\boldsymbol{x}_i^{(r)})\right\|^2 - \frac{1}{2}\left\|\nabla f(\bar{\boldsymbol{x}}^{(r)})\right\|^2 - \frac{1}{2}\left\|\frac{1}{N}\sum_{i=1}^{N}\nabla f_i(\boldsymbol{x}_i^{(r)})\right\|^2$$

$$\overset{(5)}{\leq} \frac{L^2}{2}\frac{1}{N}\sum_{i=1}^{N}\left\|\bar{\boldsymbol{x}}^{(r)} - \boldsymbol{x}_i^{(r)}\right\|^2 - \frac{1}{2}\left\|\nabla f(\bar{\boldsymbol{x}}^{(r)})\right\|^2 - \frac{1}{2}\left\|\frac{1}{N}\sum_{i=1}^{N}\nabla f_i(\boldsymbol{x}_i^{(r)})\right\|^2.$$

Then, we can bound $T_3$ from above as follows:

$$T_3 = \left\| \frac{1}{N} \sum_{i=1}^{N} \nabla f_i(\boldsymbol{x}_i^{(r)}) \pm \nabla f(\bar{\boldsymbol{x}}^{(r)}) \right\|^2$$

$$\overset{(13)}{\leq} 2 \left\| \frac{1}{N} \sum_{i=1}^{N} \nabla f_i(\boldsymbol{x}_i^{(r)}) - \nabla f(\bar{\boldsymbol{x}}^{(r)}) \right\|^2 + 2 \left\| \nabla f(\bar{\boldsymbol{x}}^{(r)}) \right\|^2$$

$$\overset{(14)}{\leq} \frac{2}{N} \sum_{i=1}^{N} \left\| \nabla f_i(\boldsymbol{x}_i^{(r)}) - \nabla f_i(\bar{\boldsymbol{x}}^{(r)}) \right\|^2 + 2 \left\| \nabla f(\bar{\boldsymbol{x}}^{(r)}) \right\|^2$$

$$\overset{(5)}{\leq} \frac{2L^2}{N} \sum_{i=1}^{N} \left\| \boldsymbol{x}_i^{(r)} - \bar{\boldsymbol{x}}^{(r)} \right\|^2 + 2 \left\| \nabla f(\bar{\boldsymbol{x}}^{(r)}) \right\|^2 .$$

By combining them, we obtain

$$\mathbb{E}_r f(\bar{\boldsymbol{z}}^{(r+1)})$$

$$\leq f(\bar{\boldsymbol{z}}^{(r)}) + \frac{L^2 \eta}{1 - \beta} \left\| \bar{\boldsymbol{x}}^{(r)} - \bar{\boldsymbol{z}}^{(r)} \right\|^2 - \frac{\eta}{4(1 - \beta)} \left\| \frac{1}{N} \sum_{i=1}^{N} \nabla f_i(\boldsymbol{x}_i^{(r)}) \right\|^2$$

$$+ \frac{L^2}{1 - \beta} \left( \frac{1}{2} + \frac{L\eta}{1 - \beta} \right) \eta \Xi^{(r)} - \frac{1}{1 - \beta} \left( \frac{1}{2} - \frac{L\eta}{1 - \beta} \right) \eta \left\| \nabla f(\bar{\boldsymbol{x}}^{(r)}) \right\|^2 + \frac{L\sigma^2 \eta^2}{2N(1 - \beta)^2} .$$

Using $\eta \leq \frac{1-\beta}{4L}$, we get the statement. $\qquad \square$

**Lemma 14** (Recursion for $\Xi$). *Suppose that Assumptions 1, 2, 3, and 4 hold. Then, it holds that for any round $r \geq 0$,*

$$\Xi^{(r+1)} \leq (1 - \frac{p}{2}) \Xi^{(r)} + \frac{9}{p} \eta^2 \mathcal{E}^{(r)} + \frac{9}{Np} \eta^2 \mathbb{E} \left\| \boldsymbol{U}^{(r+1)} - \boldsymbol{D}^{(r+1)} \right\|_F^2 + \frac{9}{Np} \eta^2 \mathbb{E} \left\| \boldsymbol{E}^{(r+1)} \right\|_F^2 .$$

*Proof.* Because $\sum_{i=1}^{N} \| \boldsymbol{a}_i - \bar{\boldsymbol{a}} \|^2 \leq \sum_{i=1}^{N} \| \boldsymbol{a}_i \|^2$ for any $\boldsymbol{a}_1, \cdots, \boldsymbol{a}_N \in \mathbb{R}^d$, we have

$$N \Xi^{(r)} = \mathbb{E} \left\| (\boldsymbol{X}^{(r)} - \bar{\boldsymbol{X}}^{(r-1)}) + (\bar{\boldsymbol{X}}^{(r-1)} - \bar{\boldsymbol{X}}^{(r)}) \right\|_F^2 \leq \mathbb{E} \left\| \boldsymbol{X}^{(r)} - \bar{\boldsymbol{X}}^{(r-1)} \right\|_F^2 .$$

Then, we have

$$\left\| \boldsymbol{X}^{(r+1)} - \bar{\boldsymbol{X}}^{(r+1)} \right\|_F^2 \leq \left\| \boldsymbol{X}^{(r+1)} - \bar{\boldsymbol{X}}^{(r)} \right\|_F^2$$

$$= \left\| \boldsymbol{X}^{(r)} \boldsymbol{W} - \eta (\boldsymbol{U}^{(r+1)} - \boldsymbol{C}^{(r)}) - \bar{\boldsymbol{X}}^{(r)} \right\|_F^2$$

$$\overset{(13)}{\leq} (1 + \gamma) \left\| \boldsymbol{X}^{(r)} \boldsymbol{W} - \bar{\boldsymbol{X}}^{(r)} \right\|_F^2 + (1 + \gamma^{-1}) \eta^2 \left\| \boldsymbol{U}^{(r+1)} - \boldsymbol{C}^{(r)} \right\|_F^2$$

$$\overset{(4)}{\leq} (1 + \gamma)(1 - p) \left\| \boldsymbol{X}^{(r)} - \bar{\boldsymbol{X}}^{(r)} \right\|_F^2 + (1 + \gamma^{-1}) \eta^2 \left\| \boldsymbol{U}^{(r+1)} - \boldsymbol{C}^{(r)} \right\|_F^2 .$$

By substituting $\gamma = \frac{p}{2}$ and using $p \leq 1$, we obtain

$$\left\| \boldsymbol{X}^{(r+1)} - \bar{\boldsymbol{X}}^{(r+1)} \right\|_F^2$$

$$\leq (1 - \frac{p}{2}) \left\| \boldsymbol{X}^{(r)} - \bar{\boldsymbol{X}}^{(r)} \right\|_F^2 + \frac{3}{p}\eta^2 \left\| \boldsymbol{U}^{(r+1)} - \boldsymbol{C}^{(r)} \right\|_F^2$$

$$= (1 - \frac{p}{2}) \left\| \boldsymbol{X}^{(r)} - \bar{\boldsymbol{X}}^{(r)} \right\|_F^2 + \frac{3}{p}\eta^2 \left\| \boldsymbol{U}^{(r+1)} \pm \boldsymbol{D}^{(r+1)} \pm \boldsymbol{E}^{(r+1)} - \boldsymbol{C}^{(r)} \right\|_F^2$$

$$\overset{(14)}{\leq} (1 - \frac{p}{2}) \left\| \boldsymbol{X}^{(r)} - \bar{\boldsymbol{X}}^{(r)} \right\|_F^2$$

$$+ \frac{9}{p}\eta^2 \left\| \boldsymbol{U}^{(r+1)} - \boldsymbol{D}^{(r+1)} \right\|_F^2 + \frac{9}{p}\eta^2 \left\| \boldsymbol{D}^{(r+1)} - \boldsymbol{C}^{(r)} - \boldsymbol{E}^{(r+1)} \right\|_F^2 + \frac{9}{p}\eta^2 \left\| \boldsymbol{E}^{(r+1)} \right\|_F^2.$$

This concludes the proof. $\qquad\square$

**Lemma 15** (Recursion for $\mathcal{E}$). *Suppose that Assumptions 1, 2, 3, and 4 hold. Then, it holds that for any round $r \geq 0$,*

$$\mathcal{E}^{(r+1)} \leq (1 - \frac{p}{2})\mathcal{E}^{(r)} + \frac{18\beta^2}{p}\mathcal{D}^{(r)} + \frac{24}{Np}\mathbb{E}\left\| \boldsymbol{U}^{(r+1)} - \boldsymbol{D}^{(r+1)} \right\|_F^2$$

$$+ \frac{144L^4}{p}\eta^2 \Xi^{(r)} + \frac{72\beta^2 L^2}{p}\eta^2 \mathbb{E}\left\| \bar{\boldsymbol{u}}^{(r)} \right\|^2 + \frac{144L^2}{p}\eta^2 \mathbb{E}\left\| \nabla f(\bar{\boldsymbol{x}}^{(r)}) \right\|^2 + \frac{36L^2\sigma^2\eta^2}{Np}.$$

*Proof.* We have

$$\mathbb{E}\left\| \boldsymbol{D}^{(r+2)} - \boldsymbol{C}^{(r+1)} - \boldsymbol{E}^{(r+2)} \right\|_F^2$$

$$= \mathbb{E}\left\| \boldsymbol{D}^{(r+2)} - (\boldsymbol{C}^{(r)} - \boldsymbol{U}^{(r+1)})\boldsymbol{W} - \boldsymbol{U}^{(r+1)} - \boldsymbol{E}^{(r+2)} \pm \boldsymbol{D}^{(r+1)} \pm \boldsymbol{D}^{(r+1)}\boldsymbol{W} \pm \boldsymbol{E}^{(r+1)} \right\|_F^2$$

$$\overset{(13),(14)}{\leq} (1 + \gamma)\mathbb{E}\left\| (\boldsymbol{D}^{(r+1)} - \boldsymbol{C}^{(r)})\boldsymbol{W} - \boldsymbol{E}^{(r+1)} \right\|_F^2$$

$$+ 2(1 + \gamma^{-1})\mathbb{E}\left\| (\boldsymbol{U}^{(r+1)} - \boldsymbol{D}^{(r+1)})(\boldsymbol{W} - \boldsymbol{I}) \right\|_F^2$$

$$+ 2(1 + \gamma^{-1})\mathbb{E}\left\| \boldsymbol{D}^{(r+2)} - \boldsymbol{D}^{(r+1)} + \boldsymbol{E}^{(r+1)} - \boldsymbol{E}^{(r+2)} \right\|_F^2$$

$$\overset{(4)}{\leq} (1 + \gamma)(1 - p)\mathbb{E}\left\| \boldsymbol{D}^{(r+1)} - \boldsymbol{C}^{(r)} - \boldsymbol{E}^{(r+1)} \right\|_F^2$$

$$+ 2(1 + \gamma^{-1})\mathbb{E}\left\| (\boldsymbol{U}^{(r+1)} - \boldsymbol{D}^{(r+1)})(\boldsymbol{W} - \boldsymbol{I}) \right\|_F^2$$

$$+ 2(1 + \gamma^{-1})\mathbb{E}\left\| \boldsymbol{D}^{(r+2)} - \boldsymbol{D}^{(r+1)} + \boldsymbol{E}^{(r+1)} - \boldsymbol{E}^{(r+2)} \right\|_F^2,$$

where we use Lemma 4 and $\boldsymbol{E}^{(r+1)} = \frac{1}{N}\boldsymbol{D}^{(r+1)}\boldsymbol{1}\boldsymbol{1}^\top$ in the last inequality. Then, we have

$$\mathbb{E}\left\| \boldsymbol{D}^{(r+2)} - \boldsymbol{C}^{(r+1)} - \boldsymbol{E}^{(r+2)} \right\|_F^2$$

$$\overset{(a)}{\leq} (1 + \gamma)(1 - p)\mathbb{E}\left\| \boldsymbol{D}^{(r+1)} - \boldsymbol{C}^{(r)} - \boldsymbol{E}^{(r+1)} \right\|_F^2$$

$$+ 2(1 + \gamma^{-1})\mathbb{E}\left\| \boldsymbol{U}^{(r+1)} - \boldsymbol{D}^{(r+1)} \right\|_F^2 \|\boldsymbol{W} - \boldsymbol{I}\|_{\text{op}}^2$$

$$+ 2(1 + \gamma^{-1})\mathbb{E}\left\| \boldsymbol{D}^{(r+2)} - \boldsymbol{D}^{(r+1)} + \boldsymbol{E}^{(r+1)} - \boldsymbol{E}^{(r+2)} \right\|_F^2$$

$$\overset{(b)}{\leq} (1 + \gamma)(1 - p)\mathbb{E}\left\| \boldsymbol{D}^{(r+1)} - \boldsymbol{C}^{(r)} - \boldsymbol{E}^{(r+1)} \right\|_F^2 + 8(1 + \gamma^{-1})\mathbb{E}\left\| \boldsymbol{U}^{(r+1)} - \boldsymbol{D}^{(r+1)} \right\|_F^2$$

$$+ 2(1 + \gamma^{-1})\mathbb{E}\left\| \boldsymbol{D}^{(r+2)} - \boldsymbol{D}^{(r+1)} + \boldsymbol{E}^{(r+1)} - \boldsymbol{E}^{(r+2)} \right\|_F^2,$$

where $\|\cdot\|_{\mathrm{op}}$ denotes the operator norm. In (a), we use the following definition of the operator norm: $\|\boldsymbol{W} - \boldsymbol{I}\|_{\mathrm{op}} := \sup_{\hat{\boldsymbol{v}} \in \mathbb{R}^d \setminus \{\boldsymbol{0}\}} \frac{\|(\boldsymbol{W}-\boldsymbol{I})\hat{\boldsymbol{v}}\|}{\|\hat{\boldsymbol{v}}\|} \geq \frac{\|(\boldsymbol{W}-\boldsymbol{I})\boldsymbol{v}\|}{\|\boldsymbol{v}\|}$ for any $\boldsymbol{v} \in \mathbb{R}^d \setminus \{\boldsymbol{0}\}$. In (b), we use Gershgorin circle theorem and the fact that $\boldsymbol{W}$ is a mixing matrix. Substituting $\gamma = \frac{p}{2}$, we obtain

$$\mathbb{E}\left\|\boldsymbol{D}^{(r+2)} - \boldsymbol{C}^{(r+1)} - \boldsymbol{E}^{(r+2)}\right\|_F^2$$
$$\leq (1 - \frac{p}{2})\mathbb{E}\left\|\boldsymbol{D}^{(r+1)} - \boldsymbol{C}^{(r)} - \boldsymbol{E}^{(r+1)}\right\|_F^2 + \frac{24}{p}\mathbb{E}\left\|\boldsymbol{U}^{(r+1)} - \boldsymbol{D}^{(r+1)}\right\|_F^2$$
$$+ \frac{6}{p}\underbrace{\mathbb{E}\left\|\boldsymbol{D}^{(r+2)} - \boldsymbol{D}^{(r+1)} + \boldsymbol{E}^{(r+1)} - \boldsymbol{E}^{(r+2)}\right\|_F^2}_{T}.$$

Then, we can bound $T$ from above by expanding $\boldsymbol{D}^{(r+2)}$, $\boldsymbol{D}^{(r+1)}$, $\boldsymbol{E}^{(r+2)}$, and $\boldsymbol{E}^{(r+1)}$ as follows:

$$T \overset{(14)}{\leq} 3\beta^2 \mathbb{E}\left\|\boldsymbol{D}^{(r+1)} - \boldsymbol{D}^{(r)} + \boldsymbol{E}^{(r)} - \boldsymbol{E}^{(r+1)}\right\|_F^2 + 3\mathbb{E}\left\|\nabla f(\bar{\boldsymbol{X}}^{(r+1)}) - \nabla f(\bar{\boldsymbol{X}}^{(r)})\right\|_F^2$$
$$+ 3\mathbb{E}\left\|\frac{1}{N}\nabla f(\bar{\boldsymbol{X}}^{(r)})\boldsymbol{1}\boldsymbol{1}^\top - \frac{1}{N}\nabla f(\bar{\boldsymbol{X}}^{(r+1)})\boldsymbol{1}\boldsymbol{1}^\top\right\|_F^2$$
$$\overset{(5)}{\leq} 3\beta^2 \mathbb{E}\left\|\boldsymbol{D}^{(r+1)} - \boldsymbol{D}^{(r)} + \boldsymbol{E}^{(r)} - \boldsymbol{E}^{(r+1)}\right\|_F^2 + 6L^2 \mathbb{E}\left\|\bar{\boldsymbol{X}}^{(r+1)} - \bar{\boldsymbol{X}}^{(r)}\right\|_F^2.$$

Using Lemma 8, we obtain

$$T \leq 3\beta^2 \mathbb{E}\left\|\boldsymbol{D}^{(r+1)} - \boldsymbol{D}^{(r)} + \boldsymbol{E}^{(r)} - \boldsymbol{E}^{(r+1)}\right\|_F^2$$
$$+ N(24L^4\eta^2\Xi^{(r)} + 12\beta^2 L^2\eta^2\mathbb{E}\left\|\bar{\boldsymbol{u}}^{(r)}\right\|^2 + 24L^2\eta^2\mathbb{E}\left\|\nabla f(\bar{\boldsymbol{x}}^{(r)})\right\|^2 + \frac{6L^2\sigma^2\eta^2}{N}).$$

This concludes the proof. $\qquad\square$

**Lemma 16** (Recursion for $\mathcal{D}$). *Suppose that Assumptions 1, 2, 3, and 4 hold. Then, it holds that for any round $r \geq 0$,*

$$\mathcal{D}^{(r+1)} \leq \frac{2\beta^2}{1+\beta^2}\mathcal{D}^{(r)} + \frac{32L^4\eta^2}{1-\beta^2}\Xi^{(r)} + \frac{16L^2\beta^2\eta^2}{1-\beta^2}\mathbb{E}\left\|\bar{\boldsymbol{u}}^{(r)}\right\|^2$$
$$+ \frac{32L^2\eta^2}{1-\beta^2}\mathbb{E}\left\|\nabla f(\bar{\boldsymbol{x}}^{(r)})\right\|^2 + \frac{8L^2\sigma^2\eta^2}{N(1-\beta^2)}.$$

*Proof.* We have

$$\mathbb{E}\left\|\boldsymbol{D}^{(r+2)} - \boldsymbol{D}^{(r+1)} - \boldsymbol{E}^{(r+2)} + \boldsymbol{E}^{(r+1)}\right\|_F^2$$
$$\overset{(13),(14)}{\leq} (1+\gamma)\beta^2 \mathbb{E}\left\|\boldsymbol{D}^{(r+1)} - \boldsymbol{D}^{(r)} + \boldsymbol{E}^{(r)} - \boldsymbol{E}^{(r+1)}\right\|_F^2$$
$$+ 2(1+\gamma^{-1})\mathbb{E}\left\|\nabla f(\bar{\boldsymbol{X}}^{(r+1)}) - \nabla f(\bar{\boldsymbol{X}}^{(r)})\right\|_F^2$$
$$+ 2(1+\gamma^{-1})\mathbb{E}\left\|\frac{1}{N}\nabla f(\bar{\boldsymbol{X}}^{(r)})\boldsymbol{1}\boldsymbol{1}^\top - \frac{1}{N}\nabla f(\bar{\boldsymbol{X}}^{(r+1)})\boldsymbol{1}\boldsymbol{1}^\top\right\|_F^2$$
$$\overset{(5)}{\leq} (1+\gamma)\beta^2 \mathbb{E}\left\|\boldsymbol{D}^{(r+1)} - \boldsymbol{D}^{(r)} + \boldsymbol{E}^{(r)} - \boldsymbol{E}^{(r+1)}\right\|_F^2 + 4L^2(1+\gamma^{-1})\mathbb{E}\left\|\bar{\boldsymbol{X}}^{(r+1)} - \bar{\boldsymbol{X}}^{(r)}\right\|_F^2.$$

Substituting $\gamma = \frac{1-\beta^2}{1+\beta^2}$, we obtain

$$\mathbb{E}\left\|\boldsymbol{D}^{(r+2)} - \boldsymbol{D}^{(r+1)} - \boldsymbol{E}^{(r+2)} + \boldsymbol{E}^{(r+1)}\right\|_F^2$$
$$\leq \frac{2\beta^2}{1+\beta^2}\mathbb{E}\left\|\boldsymbol{D}^{(r+1)} - \boldsymbol{D}^{(r)} + \boldsymbol{E}^{(r)} - \boldsymbol{E}^{(r+1)}\right\|_F^2 + \frac{8L^2}{1-\beta^2}\mathbb{E}\left\|\bar{\boldsymbol{X}}^{(r+1)} - \bar{\boldsymbol{X}}^{(r)}\right\|_F^2.$$

Using Lemma 8, we obtain

$$\mathbb{E}\left\|\boldsymbol{D}^{(r+2)} - \boldsymbol{D}^{(r+1)} - \boldsymbol{E}^{(r+2)} + \boldsymbol{E}^{(r+1)}\right\|_F^2$$

$$\leq \frac{2\beta^2}{1+\beta^2}\mathbb{E}\left\|\boldsymbol{D}^{(r+1)} - \boldsymbol{D}^{(r)} + \boldsymbol{E}^{(r)} - \boldsymbol{E}^{(r+1)}\right\|_F^2$$

$$+ N\Big(\frac{32L^4\eta^2}{1-\beta^2}\Xi^{(r)} + \frac{16L^2\beta^2\eta^2}{1-\beta^2}\mathbb{E}\left\|\bar{\boldsymbol{u}}^{(r)}\right\|^2 + \frac{32L^2\eta^2}{1-\beta^2}\mathbb{E}\left\|\nabla f(\bar{\boldsymbol{x}}^{(r)})\right\|^2 + \frac{8L^2\sigma^2\eta^2}{N(1-\beta^2)}\Big).$$

This concludes the proof.

$\square$

**Lemma 17** (Recursion for $\Xi$, $\mathcal{E}$, and $\mathcal{D}$). *We define $t \in \mathbb{R}$ and $A \in \mathbb{R}$ as follows:*

$$t := \frac{2\beta^2 p}{1-\beta^2} + 4, \quad A := \frac{648}{1 - \frac{p}{t} - \frac{2\beta^2}{1+\beta^2}}.$$

*Note that it holds that $t \geq 4$ and $A > 0$. Suppose that Assumptions 1, 2, 3, and 4 hold, and step size $\eta$ satisfies*

$$\eta \leq \frac{p}{8L\sqrt{324 + \frac{2A\beta^2}{1-\beta^2}}}.$$

*Then, it holds that*

$$\Xi^{(r+1)} + \frac{36}{p^2}\eta^2\mathcal{E}^{(r+1)} + \frac{A\beta^2}{p^3}\eta^2\mathcal{D}^{(r+1)}$$

$$\leq (1 - \frac{p}{t})(\Xi^{(r)} + \frac{36}{p^2}\eta^2\mathcal{E}^{(r)} + \frac{A\beta^2}{p^3}\eta^2\mathcal{D}^{(r)})$$

$$+ \frac{1}{p}\eta^2\mathbb{E}\left\|\nabla f(\bar{\boldsymbol{x}}^{(r)})\right\|^2 + \frac{1}{Np}\left(9 + \frac{864}{p^2}\right)\eta^2\mathbb{E}\left\|\boldsymbol{U}^{(r+1)} - \boldsymbol{D}^{(r+1)}\right\|_F^2$$

$$+ \frac{L^2}{p^3}\left(2592\beta^2 + \frac{16A\beta^4}{1-\beta^2}\right)\eta^4\mathbb{E}\left\|\bar{\boldsymbol{u}}^{(r)}\right\|^2 + \frac{9}{Np}\eta^2\mathbb{E}\left\|\boldsymbol{E}^{(r+1)}\right\|_F^2 + \frac{\sigma^2\eta^2}{p}.$$

*Proof.* From Lemmas 14 and 15, we have

$$\Xi^{(r+1)} \leq (1 - \frac{p}{2})\Xi^{(r)} + \frac{9}{p}\eta^2\mathcal{E}^{(r)} + \frac{9}{Np}\eta^2\mathbb{E}\left\|\boldsymbol{U}^{(r+1)} - \boldsymbol{D}^{(r+1)}\right\|_F^2 + \frac{9}{Np}\eta^2\mathbb{E}\left\|\boldsymbol{E}^{(r+1)}\right\|_F^2,$$

$$\frac{36}{p^2}\eta^2\mathcal{E}^{(r+1)} \leq (1 - \frac{p}{2})\frac{36}{p^2}\eta^2\mathcal{E}^{(r)} + \frac{648\beta^2}{p^3}\eta^2\mathcal{D}^{(r)} + \frac{5184L^4}{p^3}\eta^4\Xi^{(r)} + \frac{2592\beta^2 L^2}{p^3}\eta^4\mathbb{E}\left\|\bar{\boldsymbol{u}}^{(r)}\right\|^2$$

$$+ \frac{864}{Np^3}\eta^2\mathbb{E}\left\|\boldsymbol{U}^{(r+1)} - \boldsymbol{D}^{(r+1)}\right\|_F^2 + \frac{5184L^2}{p^3}\eta^4\mathbb{E}\left\|\nabla f(\bar{\boldsymbol{x}}^{(r)})\right\|^2 + \frac{1296L^2\sigma^2\eta^4}{Np^3}.$$

Then, from Lemma 16, we have

$$\frac{A\beta^2}{p^3}\eta^2\mathcal{D}^{(r+1)} \leq \frac{2A\beta^4}{(1+\beta^2)p^3}\eta^2\mathcal{D}^{(r)} + \frac{32A\beta^2 L^4}{(1-\beta^2)p^3}\eta^4\Xi^{(r)} + \frac{16AL^2\beta^4}{(1-\beta^2)p^3}\eta^4\mathbb{E}\left\|\bar{\boldsymbol{u}}^{(r)}\right\|^2$$

$$+ \frac{32A\beta^2 L^2}{(1-\beta^2)p^3}\eta^4\mathbb{E}\left\|\nabla f(\bar{\boldsymbol{x}}^{(r)})\right\|^2 + \frac{8A\beta^2 L^2\sigma^2}{N(1-\beta^2)p^3}\eta^4.$$

Using $\eta^2 \leq \frac{p^2}{L^2}$ and $\eta^2 \leq \frac{p^2}{128L^2(162 + \frac{A\beta^2}{1-\beta^2})}$, we have

$$\left((1 - \frac{p}{2}) + \frac{5184L^4}{p^3}\eta^4 + \frac{32A\beta^2 L^4}{(1-\beta^2)p^3}\eta^4\right)\Xi^{(r)} \leq \left((1 - \frac{p}{2}) + \frac{L^2}{4p}\eta^2\right)\Xi^{(r)} \leq (1 - \frac{p}{4})\Xi^{(r)}.$$

In addition, we have

$$\left(\frac{9}{p}\eta^2 + (1 - \frac{p}{2})\frac{36}{p^2}\eta^2\right)\mathcal{E}^{(r)} = (1 - \frac{p}{4})\frac{36}{p^2}\eta^2\mathcal{E}^{(r)},$$

$$\left(\frac{648\beta^2}{p^3}\eta^2 + \frac{2A\beta^4}{(1+\beta^2)p^3}\eta^2\right)\mathcal{D}^{(r)} = \left(\frac{648}{A} + \frac{2\beta^2}{1+\beta^2}\right)\frac{A\beta^2}{p^3}\eta^2\mathcal{D}^{(r)} = (1 - \frac{p}{t})\frac{A\beta^2}{p^3}\eta^2\mathcal{D}^{(r)}.$$

Then, using $t \geq 4$, we obtain

$$\Xi^{(r+1)} + \frac{36}{p^2}\eta^2\mathcal{E}^{(r+1)} + \frac{A\beta^2}{p^3}\eta^2\mathcal{D}^{(r+1)}$$

$$\leq (1 - \frac{p}{t})(\Xi^{(r)} + \frac{36}{p^2}\eta^2\mathcal{E}^{(r)} + \frac{A\beta^2}{p^3}\eta^2\mathcal{D}^{(r)})$$

$$+ \frac{L^2}{p^3}\left(\frac{32A\beta^2}{1-\beta^2} + 5184\right)\eta^4\mathbb{E}\left\|\nabla f(\bar{\boldsymbol{x}}^{(r)})\right\|^2$$

$$+ \frac{1}{Np}\left(9 + \frac{864}{p^2}\right)\eta^2\mathbb{E}\left\|\boldsymbol{U}^{(r+1)} - \boldsymbol{D}^{(r+1)}\right\|_F^2$$

$$+ \frac{9}{Np}\eta^2\mathbb{E}\left\|\boldsymbol{E}^{(r+1)}\right\|_F^2$$

$$+ \frac{L^2}{p^3}\left(2592\beta^2 + \frac{16A\beta^4}{1-\beta^2}\right)\eta^4\mathbb{E}\left\|\bar{\boldsymbol{u}}^{(r)}\right\|^2$$

$$+ \frac{L^2\sigma^2}{Np^3}\left(1296 + \frac{8A\beta^2}{1-\beta^2}\right)\eta^4.$$

Using $\eta^2 \leq \frac{p^2}{32L^2(162 + \frac{A\beta^2}{1-\beta^2})}$, we have

$$\frac{L^2}{p^3}\left(\frac{32A\beta^2}{1-\beta^2} + 5184\right)\eta^4\mathbb{E}\left\|\nabla f(\bar{\boldsymbol{x}}^{(r)})\right\|^2 \leq \frac{1}{p}\eta^2\mathbb{E}\left\|\nabla f(\bar{\boldsymbol{x}}^{(r)})\right\|^2.$$

Using $\eta^2 \leq \frac{p^2}{8L^2(162 + \frac{A\beta^2}{1-\beta^2})}$, we obtain

$$\frac{L^2\sigma^2}{Np^3}\left(1296 + \frac{8A\beta^2}{1-\beta^2}\right)\eta^4 \leq \frac{\sigma^2\eta^2}{Np} \leq \frac{\sigma^2\eta^2}{p}.$$

This concludes the proof. $\qquad\square$

**Lemma 18.** *We define $t \in \mathbb{R}$ and $A \in \mathbb{R}$ as follows:*

$$t := \frac{2\beta^2 p}{1 - \beta^2} + 4, \quad A := \frac{648}{1 - \frac{p}{t} - \frac{2\beta^2}{1+\beta^2}}.$$

*Under the same assumptions as those in Lemma 17, it holds that*

$$\sum_{r=0}^{R}\Xi^{(r)} \leq \frac{t}{p}\sum_{k=0}^{R}\Psi^{(k)} + \frac{145t\sigma^2\eta^2}{(1-\beta)^2 p^3}R.$$

*where $\Psi^{(r)}$ is defined as follows:*

$$\Psi^{(r)} := \frac{1}{p}\eta^2\mathbb{E}\left\|\nabla f(\bar{\boldsymbol{x}}^{(r)})\right\|^2 + \frac{1}{Np}\left(9 + \frac{864}{p^2}\right)\eta^2\mathbb{E}\left\|\boldsymbol{U}^{(r+1)} - \boldsymbol{D}^{(r+1)}\right\|_F^2$$

$$+ \frac{L^2}{p^3}\left(2592\beta^2 + \frac{16A\beta^4}{1-\beta^2}\right)\eta^4\mathbb{E}\left\|\bar{\boldsymbol{u}}^{(r)}\right\|^2 + \frac{9}{Np}\eta^2\mathbb{E}\left\|\boldsymbol{E}^{(r+1)}\right\|_F^2.$$

*Proof.* We define $\Theta^{(r)} := \Xi^{(r)} + \frac{36}{p^2}\eta^2\mathcal{E}^{(r)} + \frac{A\beta^2}{p^3}\eta^2\mathcal{D}^{(r)}$. From Lemma 17, we obtain

$$\Theta^{(r+1)} \le (1 - \frac{p}{t})\Theta^{(r)} + \Psi^{(r)} + \frac{\sigma^2\eta^2}{p}$$

$$\le (1 - \frac{p}{t})^{r+1}\Theta^{(0)} + \sum_{k=0}^{r}(1 - \frac{p}{t})^{r-k}\Psi^{(k)} + \sum_{k=0}^{r}(1 - \frac{p}{t})^{r-k}\frac{\sigma^2\eta^2}{p}.$$

Using $\sum_{k=0}^{r}(1 - \frac{p}{t})^{r-k} \le \frac{t}{p}$, we obtain

$$\Theta^{(r+1)} \le (1 - \frac{p}{t})^{r+1}\Theta^{(0)} + \sum_{k=0}^{r}(1 - \frac{p}{t})^{r-k}\Psi^{(k)} + \frac{t\sigma^2\eta^2}{p^2}.$$

Then, for any $R \ge 1$, we obtain

$$\sum_{r=1}^{R}\Theta^{(r)} \le \sum_{r=1}^{R}(1 - \frac{p}{t})^{r}\Theta^{(0)} + \sum_{r=1}^{R}\sum_{k=0}^{r-1}(1 - \frac{p}{t})^{r-k-1}\Psi^{(k)} + \frac{t\sigma^2\eta^2}{p^2}R$$

$$= \sum_{r=1}^{R}(1 - \frac{p}{t})^{r}\Theta^{(0)} + \sum_{k=0}^{R-1}\Psi^{(k)}\sum_{r=k+1}^{R}(1 - \frac{p}{t})^{r-k-1} + \frac{t\sigma^2\eta^2}{p^2}R$$

$$\le \frac{t}{p}\Theta^{(0)} + \frac{t}{p}\sum_{k=0}^{R-1}\Psi^{(k)} + \frac{t\sigma^2\eta^2}{p^2}R,$$

where we use $\sum_{r=1}^{R}(1 - \frac{p}{t})^{r} \le \frac{t}{p}$ and $\sum_{r=k+1}^{R}(1 - \frac{p}{t})^{r-k-1} \le \frac{t}{p}$ in the last inequality. Then, from the definition of $\mathcal{E}^{(r)}$, we have

$$\mathcal{E}^{(0)} = \frac{1}{N}\mathbb{E}\left\|\boldsymbol{D}^{(1)} - \boldsymbol{C}^{(0)} - \boldsymbol{E}^{(1)}\right\|_F^2$$

$$= \frac{1}{(1-\beta)^2}\frac{1}{N}\mathbb{E}\left\|\nabla f(\bar{\boldsymbol{X}}^{(0)}) - \frac{1}{N}\nabla f(\bar{\boldsymbol{X}}^{(0)})\mathbf{1}\mathbf{1}^\top - \nabla F(\boldsymbol{X}^{(0)};\xi^{(0)}) + \frac{1}{N}\nabla F(\boldsymbol{X}^{(0)};\xi^{(0)})\mathbf{1}\mathbf{1}^\top\right\|_F^2$$

$$\overset{(14)}{\le} \frac{2}{(1-\beta)^2}\frac{1}{N}\mathbb{E}\left\|\nabla f(\bar{\boldsymbol{X}}^{(0)}) - \nabla F(\boldsymbol{X}^{(0)};\xi^{(0)})\right\|_F^2$$

$$+ \frac{2}{(1-\beta)^2}\frac{1}{N}\mathbb{E}\left\|\frac{1}{N}\nabla f(\bar{\boldsymbol{X}}^{(0)})\mathbf{1}\mathbf{1}^\top - \frac{1}{N}\nabla F(\boldsymbol{X}^{(0)};\xi^{(0)})\mathbf{1}\mathbf{1}^\top\right\|_F^2$$

$$\overset{(6)}{\le} \frac{4}{(1-\beta)^2}\sigma^2,$$

where we use $\boldsymbol{X}^{(0)} = \bar{\boldsymbol{X}}^{(0)}$ in the last inequality. From the definition of $\mathcal{D}^{(r)}$, we have

$$\mathcal{D}^{(0)} = \frac{1}{N}\mathbb{E}\left\|(\beta - 1)\boldsymbol{D}^{(0)} + \nabla f(\bar{\boldsymbol{X}}^{(0)}) - \frac{1}{N}\nabla f(\bar{\boldsymbol{X}}^{(0)})\mathbf{1}\mathbf{1}^\top\right\|_F^2 = 0.$$

Then using $\boldsymbol{X}^{(0)} = \bar{\boldsymbol{X}}^{(0)}$ (i.e., $\Xi^{(0)} = 0$), we have

$$\Theta^{(0)} \le \frac{144\sigma^2\eta^2}{(1-\beta)^2 p^2}.$$

Here, the above upper bounds of $\mathcal{E}^{(0)}$ and $\mathcal{D}^{(0)}$ are attributed to how we choose the initial values $\boldsymbol{u}_i^{(0)}$, $\boldsymbol{c}_i^{(0)}$, $\boldsymbol{d}_i^{(0)}$, and $\boldsymbol{e}_i^{(0)}$ for $i \in V$. Then, combining them, we obtain

$$\sum_{r=1}^{R}\Theta^{(r)} \le \frac{t}{p}\sum_{k=0}^{R-1}\Psi^{(k)} + \frac{144t\sigma^2\eta^2}{(1-\beta)^2 p^3} + \frac{t\sigma^2\eta^2}{p^2}R$$

$$\le \frac{t}{p}\sum_{k=0}^{R-1}\Psi^{(k)} + \frac{145t\sigma^2\eta^2}{(1-\beta)^2 p^3}R,$$

where we use $p \in (0,1]$, $\beta \in [0,1)$, and $R \geq 1$ in the last inequality. Then, using $\Theta^{(r)} \geq \Xi^{(r)}$ and $\Xi^{(0)} = 0$, we obtain the statement. $\qquad\square$

**Lemma 19.** *We define $t \in \mathbb{R}$ and $A \in \mathbb{R}$ as follows:*

$$t := \frac{2\beta^2 p}{1 - \beta^2} + 4, \quad A := \frac{648}{1 - \frac{p}{t} - \frac{2\beta^2}{1 + \beta^2}}.$$

*Note that it holds that $t \geq 4$ and $A > 0$. Suppose that the same assumptions as those in Lemma 17 hold. Then, if step size $\eta$ satisfies*

$$\eta \leq \min\{\frac{p}{4L\sqrt{324 + \frac{2A\beta^2}{1 - \beta^2}}}, \frac{(1 - \beta)p}{2L\sqrt{t(5 + \frac{432}{p^2})}}, \frac{(1 - \beta)p}{8L\sqrt{5t}}\},$$

*it holds that*

$$4L^2 \sum_{r=0}^{R} \Xi^{(r)} \leq \frac{1}{2} \sum_{k=0}^{R} \left\| \nabla f(\bar{\boldsymbol{x}}^{(k)}) \right\|^2 + \frac{40L^2 t}{(1 - \beta)^3 p^2} \left( 10 + \frac{29}{p} + \frac{864}{p^2} \right) \sigma^2 \eta^2 (R + 1).$$

*Proof.* We define $\Psi^{(r)}$ as follows:

$$\Psi^{(r)} := \frac{1}{p} \eta^2 \mathbb{E} \left\| \nabla f(\bar{\boldsymbol{x}}^{(r)}) \right\|^2 + \frac{1}{Np} \left( 9 + \frac{864}{p^2} \right) \eta^2 \mathbb{E} \left\| \boldsymbol{U}^{(r+1)} - \boldsymbol{D}^{(r+1)} \right\|_F^2$$
$$+ \frac{L^2}{p^3} \left( 2592\beta^2 + \frac{16A\beta^4}{1 - \beta^2} \right) \eta^4 \mathbb{E} \left\| \bar{\boldsymbol{u}}^{(r)} \right\|^2 + \frac{9}{Np} \eta^2 \mathbb{E} \left\| \boldsymbol{E}^{(r+1)} \right\|_F^2.$$

Using $\bar{\boldsymbol{e}}^{(r)} = \boldsymbol{e}_i^{(r)}$ and Lemmas 9, 10, and 12, we obtain

$$\Psi^{(r)} \leq \frac{1}{p} \eta^2 \mathbb{E} \left\| \nabla f(\bar{\boldsymbol{x}}^{(r)}) \right\|^2$$
$$+ \eta^2 \frac{9}{(1 - \beta)p} \sum_{k=0}^{r} \beta^{r-k} \left\| \nabla f(\bar{\boldsymbol{x}}^{(k)}) \right\|^2$$
$$+ \frac{2L^2}{(1 - \beta)p^3} \left( 2592\beta^2 + \frac{16A\beta^4}{1 - \beta^2} \right) \eta^4 \left( \sum_{k=0}^{r-1} \beta^{r-k-1} \left\| \nabla f(\bar{\boldsymbol{x}}^{(k)}) \right\|^2 \right)$$
$$+ \frac{L^2}{(1 - \beta)p} \left( 9 + \frac{864}{p^2} \right) \eta^2 \left( \sum_{k=0}^{r} \beta^{r-k} \Xi^{(k)} \right)$$
$$+ \frac{2L^4}{(1 - \beta)p^3} \left( 2592\beta^2 + \frac{16A\beta^4}{1 - \beta^2} \right) \eta^4 \left( \sum_{k=0}^{r-1} \beta^{r-k-1} \Xi^{(k)} \right)$$
$$+ \frac{5}{(1 - \beta)^3 p} \left( 9 + \frac{864}{p^2} \right) \sigma^2 \eta^2$$
$$+ \frac{2L^2}{N(1 - \beta)^2 p^3} \left( 2592\beta^2 + \frac{16A\beta^4}{1 - \beta^2} \right) \sigma^2 \eta^4,$$

for any round $r \geq 1$. Then, we obtain

$$
\begin{aligned}
\sum_{r=1}^{R} \Psi^{(r)} \leq & \frac{1}{p} \eta^2 \sum_{r=1}^{R} \mathbb{E} \left\| \nabla f(\bar{\boldsymbol{x}}^{(r)}) \right\|^2 \\
& + \eta^2 \frac{9}{(1-\beta)p} \sum_{r=1}^{R} \sum_{k=0}^{r} \beta^{r-k} \left\| \nabla f(\bar{\boldsymbol{x}}^{(k)}) \right\|^2 \\
& + \frac{2L^2}{(1-\beta)p^3} \left( 2592\beta^2 + \frac{16A\beta^4}{1-\beta^2} \right) \eta^4 \left( \sum_{r=1}^{R} \sum_{k=0}^{r-1} \beta^{r-k-1} \left\| \nabla f(\bar{\boldsymbol{x}}^{(k)}) \right\|^2 \right) \\
& + \frac{L^2}{(1-\beta)p} \left( 9 + \frac{864}{p^2} \right) \eta^2 \left( \sum_{r=1}^{R} \sum_{k=0}^{r} \beta^{r-k} \Xi^{(k)} \right) \\
& + \frac{2L^4}{(1-\beta)p^3} \left( 2592\beta^2 + \frac{16A\beta^4}{1-\beta^2} \right) \eta^4 \left( \sum_{r=1}^{R} \sum_{k=0}^{r-1} \beta^{r-k-1} \Xi^{(k)} \right) \\
& + \frac{5}{(1-\beta)^3 p} \left( 9 + \frac{864}{p^2} \right) \sigma^2 \eta^2 R \\
& + \frac{2L^2}{N(1-\beta)^2 p^3} \left( 2592\beta^2 + \frac{16A\beta^4}{1-\beta^2} \right) \sigma^2 \eta^4 R.
\end{aligned}
$$

Then, we obtain

$$
\begin{aligned}
\sum_{r=1}^{R} \Psi^{(r)} \leq & \frac{1}{p} \eta^2 \sum_{r=1}^{R} \mathbb{E} \left\| \nabla f(\bar{\boldsymbol{x}}^{(r)}) \right\|^2 \\
& + \eta^2 \frac{9}{(1-\beta)p} \sum_{k=0}^{R} \left\| \nabla f(\bar{\boldsymbol{x}}^{(k)}) \right\|^2 \sum_{r=\max\{1,k\}}^{R} \beta^{r-k} \\
& + \frac{2L^2}{(1-\beta)p^3} \left( 2592\beta^2 + \frac{16A\beta^4}{1-\beta^2} \right) \eta^4 \left( \sum_{k=0}^{R-1} \left\| \nabla f(\bar{\boldsymbol{x}}^{(k)}) \right\|^2 \sum_{r=k+1}^{R} \beta^{r-k-1} \right) \\
& + \frac{L^2}{(1-\beta)p} \left( 9 + \frac{864}{p^2} \right) \eta^2 \left( \sum_{k=0}^{R} \Xi^{(k)} \sum_{r=\max\{1,k\}}^{R} \beta^{r-k} \right) \\
& + \frac{2L^4}{(1-\beta)p^3} \left( 2592\beta^2 + \frac{16A\beta^4}{1-\beta^2} \right) \eta^4 \left( \sum_{k=0}^{R-1} \Xi^{(k)} \sum_{r=k+1}^{R} \beta^{r-k-1} \right) \\
& + \frac{5}{(1-\beta)^3 p} \left( 9 + \frac{864}{p^2} \right) \sigma^2 \eta^2 R \\
& + \frac{2L^2}{N(1-\beta)^2 p^3} \left( 2592\beta^2 + \frac{16A\beta^4}{1-\beta^2} \right) \sigma^2 \eta^4 R.
\end{aligned}
$$

Using $\sum_{r=k+1}^{R} \beta^{r-k-1} \le \frac{1}{1-\beta}$ and $\sum_{r=\max\{1,k\}}^{R} \beta^{r-k} \le \frac{1}{1-\beta}$, we obtain

$$\sum_{r=1}^{R} \Psi^{(r)} \le \frac{1}{p}\eta^2 \sum_{r=1}^{R} \mathbb{E}\left\|\nabla f(\bar{\boldsymbol{x}}^{(r)})\right\|^2$$

$$+ \eta^2 \frac{9}{(1-\beta)^2 p} \sum_{k=0}^{R} \left\|\nabla f(\bar{\boldsymbol{x}}^{(k)})\right\|^2$$

$$+ \frac{2L^2\beta^2}{(1-\beta)^2 p^3}\left(2592 + \frac{16A\beta^2}{1-\beta^2}\right)\eta^4 \left(\sum_{k=0}^{R-1}\left\|\nabla f(\bar{\boldsymbol{x}}^{(k)})\right\|^2\right)$$

$$+ \frac{L^2}{(1-\beta)^2 p}\left(9 + \frac{864}{p^2}\right)\eta^2 \left(\sum_{k=0}^{R}\Xi^{(k)}\right)$$

$$+ \frac{2L^4\beta^2}{(1-\beta)^2 p^3}\left(2592 + \frac{16A\beta^2}{1-\beta^2}\right)\eta^4 \left(\sum_{k=0}^{R-1}\Xi^{(k)}\right)$$

$$+ \frac{5}{(1-\beta)^3 p}\left(9 + \frac{864}{p^2}\right)\sigma^2\eta^2 R$$

$$+ \frac{2L^2\beta^2}{N(1-\beta)^2 p^3}\left(2592 + \frac{16A\beta^2}{1-\beta^2}\right)\sigma^2\eta^4 R.$$

Then, using $\bar{\boldsymbol{u}}^{(0)} = \boldsymbol{0}$ and Lemmas 9 and 10, we have

$$\Psi^{(0)} \le \frac{1}{p}\eta^2 \mathbb{E}\left\|\nabla f(\bar{\boldsymbol{x}}^{(0)})\right\|^2 + \frac{9}{(1-\beta)p}\eta^2 \left\|\nabla f(\bar{\boldsymbol{x}}^{(0)})\right\|^2 + \frac{5}{(1-\beta)^3 p}\left(9 + \frac{864}{p^2}\right)\sigma^2\eta^2.$$

Then, we obtain

$$\sum_{r=0}^{R} \Psi^{(r)} \le \frac{1}{p}\eta^2 \sum_{r=0}^{R} \mathbb{E}\left\|\nabla f(\bar{\boldsymbol{x}}^{(r)})\right\|^2$$

$$+ \eta^2 \frac{18}{(1-\beta)^2 p} \sum_{k=0}^{R} \left\|\nabla f(\bar{\boldsymbol{x}}^{(k)})\right\|^2$$

$$+ \frac{2L^2\beta^2}{(1-\beta)^2 p^3}\left(2592 + \frac{16A\beta^2}{1-\beta^2}\right)\eta^4 \left(\sum_{k=0}^{R-1}\left\|\nabla f(\bar{\boldsymbol{x}}^{(k)})\right\|^2\right)$$

$$+ \frac{L^2}{(1-\beta)^2 p}\left(9 + \frac{864}{p^2}\right)\eta^2 \left(\sum_{k=0}^{R}\Xi^{(k)}\right)$$

$$+ \frac{2L^4\beta^2}{(1-\beta)^2 p^3}\left(2592 + \frac{16A\beta^2}{1-\beta^2}\right)\eta^4 \left(\sum_{k=0}^{R-1}\Xi^{(k)}\right)$$

$$+ \frac{5}{(1-\beta)^3 p}\left(9 + \frac{864}{p^2}\right)\sigma^2\eta^2 (R+1)$$

$$+ \frac{2L^2\beta^2}{N(1-\beta)^2 p^3}\left(2592 + \frac{16A\beta^2}{1-\beta^2}\right)\sigma^2\eta^4 R.$$

Using $\eta^2 \le \frac{p^2}{32L^2\left(162 + \frac{A\beta^2}{1-\beta^2}\right)}$, we have

$$\frac{2L^2\beta^2}{(1-\beta)^2 p^3}\left(2592 + \frac{16A\beta^2}{1-\beta^2}\right)\eta^4 \left(\sum_{k=0}^{R-1}\left\|\nabla f(\bar{\boldsymbol{x}}^{(k)})\right\|^2\right) \le \frac{\beta^2}{(1-\beta)^2 p}\eta^2 \left(\sum_{k=0}^{R-1}\left\|\nabla f(\bar{\boldsymbol{x}}^{(k)})\right\|^2\right).$$

Using $\eta^2 \leq \frac{p^2}{32L^2(162 + \frac{A\beta^2}{1-\beta^2})}$, we have

$$\frac{2L^4\beta^2}{(1-\beta)^2 p^3} \left(2592 + \frac{16A\beta^2}{1-\beta^2}\right) \eta^4 \left(\sum_{k=0}^{R-1} \Xi^{(k)}\right) \leq \frac{L^2\beta^2}{(1-\beta)^2 p} \eta^2 \left(\sum_{k=0}^{R-1} \Xi^{(k)}\right).$$

Using $\eta^2 \leq \frac{p^2}{32L^2(162 + \frac{A\beta^2}{1-\beta^2})}$, we have

$$\frac{2L^2\beta^2}{N(1-\beta)^2 p^3} \left(2592 + \frac{16A\beta^2}{1-\beta^2}\right) \sigma^2 \eta^4 R \leq \frac{\beta^2}{N(1-\beta)^2 p} \sigma^2 \eta^2 R.$$

Then, using $\beta \in [0, 1)$ and $N \geq 1$, we obtain

$$\sum_{r=0}^{R} \Psi^{(r)} \leq \frac{1}{p}\eta^2 \sum_{r=0}^{R} \mathbb{E}\left\|\nabla f(\bar{\boldsymbol{x}}^{(r)})\right\|^2$$
$$+ \eta^2 \frac{19}{(1-\beta)^2 p} \sum_{k=0}^{R} \left\|\nabla f(\bar{\boldsymbol{x}}^{(k)})\right\|^2$$
$$+ \frac{L^2}{(1-\beta)^2 p} \left(10 + \frac{864}{p^2}\right) \eta^2 \left(\sum_{k=0}^{R} \Xi^{(k)}\right)$$
$$+ \frac{5}{(1-\beta)^3 p} \left(10 + \frac{864}{p^2}\right) \sigma^2 \eta^2 (R+1).$$

Using $\beta \in [0, 1)$ and Lemma 18, we obtain

$$\sum_{r=0}^{R} \Xi^{(r)} \leq \frac{20t}{(1-\beta)^2 p^2} \eta^2 \sum_{k=0}^{R} \left\|\nabla f(\bar{\boldsymbol{x}}^{(k)})\right\|^2$$
$$+ \frac{tL^2}{(1-\beta)^2 p^2} \left(10 + \frac{864}{p^2}\right) \eta^2 \left(\sum_{k=0}^{R} \Xi^{(k)}\right)$$
$$+ \frac{5t}{(1-\beta)^3 p^2} \left(10 + \frac{864}{p^2}\right) \sigma^2 \eta^2 (R+1) + \frac{145t\sigma^2\eta^2}{(1-\beta)^2 p^3} R$$
$$\leq \frac{20t}{(1-\beta)^2 p^2} \eta^2 \sum_{k=0}^{R} \left\|\nabla f(\bar{\boldsymbol{x}}^{(k)})\right\|^2$$
$$+ \frac{tL^2}{(1-\beta)^2 p^2} \left(10 + \frac{864}{p^2}\right) \eta^2 \left(\sum_{k=0}^{R} \Xi^{(k)}\right)$$
$$+ \frac{5t}{(1-\beta)^3 p^2} \left(10 + \frac{29}{p} + \frac{864}{p^2}\right) \sigma^2 \eta^2 (R+1).$$

Then, using $\eta^2 \leq \frac{(1-\beta)^2 p^2}{4tL^2(5 + \frac{432}{p^2})}$, we obtain

$$\frac{1}{2}\sum_{r=0}^{R} \Xi^{(r)} \leq \frac{20t}{(1-\beta)^2 p^2} \eta^2 \sum_{k=0}^{R} \left\|\nabla f(\bar{\boldsymbol{x}}^{(k)})\right\|^2 + \frac{5t}{(1-\beta)^3 p^2} \left(10 + \frac{29}{p} + \frac{864}{p^2}\right) \sigma^2 \eta^2 (R+1).$$

Multiplying $8L^2$, we obtain

$$4L^2 \sum_{r=0}^{R} \Xi^{(r)} \leq \frac{160L^2 t}{(1-\beta)^2 p^2} \eta^2 \sum_{k=0}^{R} \left\|\nabla f(\bar{\boldsymbol{x}}^{(k)})\right\|^2 + \frac{40L^2 t}{(1-\beta)^3 p^2} \left(10 + \frac{29}{p} + \frac{864}{p^2}\right) \sigma^2 \eta^2 (R+1).$$

Using $\eta^2 \leq \frac{(1-\beta)^2 p^2}{320L^2 t}$, we obtain the statement. $\qquad\square$

**Lemma 20.** *We define $t \in \mathbb{R}$ as follows:*

$$t := \frac{2\beta^2 p}{1 - \beta^2} + 4.$$

*Suppose that the assumptions of Lemma 19 hold. Then, if step size $\eta$ satisfies*

$$\eta \leq \frac{(1 - \beta)^2}{2\sqrt{2}L},$$

*it holds that*

$$\frac{1}{2(R+1)} \sum_{r=0}^{R} \mathbb{E} \left\| \nabla f(\bar{\boldsymbol{x}}^{(r)}) \right\|^2 \leq \frac{4(1-\beta)}{\eta(R+1)} \left( f(\bar{\boldsymbol{z}}^{(0)}) - f^\star \right) + \frac{2L\sigma^2\eta}{N(1-\beta)}$$
$$+ \frac{L^2}{(1-\beta)^3} \left( \frac{40t}{p^2} \left( 10 + \frac{29}{p} + \frac{864}{p^2} \right) + \frac{4\beta^2}{N(1-\beta)} \right) \sigma^2 \eta^2.$$

*Proof.* Using Lemma 13 and Assumption 1, we have

$$\sum_{r=0}^{R} \mathbb{E} \left\| \nabla f(\bar{\boldsymbol{x}}^{(r)}) \right\|^2 \leq \frac{4(1-\beta)}{\eta} \left( f(\bar{\boldsymbol{z}}^{(0)}) - f^\star \right) + 4L^2 \sum_{r=0}^{R} \mathbb{E} \left\| \bar{\boldsymbol{x}}^{(r)} - \bar{\boldsymbol{z}}^{(r)} \right\|^2 + 4L^2 \sum_{r=0}^{R} \Xi^{(r)}$$
$$- \sum_{r=0}^{R} \mathbb{E} \left\| \frac{1}{N} \sum_{i=1}^{N} \nabla f_i(\boldsymbol{x}_i^{(r)}) \right\|^2 + \frac{2L\sigma^2\eta}{N(1-\beta)}(R+1).$$

From Lemma 7, we have

$$4L^2 \sum_{r=0}^{R} \mathbb{E} \left\| \bar{\boldsymbol{x}}^{(r)} - \bar{\boldsymbol{z}}^{(r)} \right\|^2 \leq \frac{4L^2\beta^2\eta^2}{(1-\beta)^4} \sum_{r=0}^{R} \mathbb{E} \left\| \frac{1}{N} \sum_{i=1}^{N} \nabla f_i(\boldsymbol{x}_i^{(r)}) \right\|^2 + \frac{4L^2\beta^2\sigma^2\eta^2}{N(1-\beta)^4} R.$$

Combining them yields

$$\sum_{r=0}^{R} \mathbb{E} \left\| \nabla f(\bar{\boldsymbol{x}}^{(r)}) \right\|^2$$
$$\leq \frac{4(1-\beta)}{\eta} \left( f(\bar{\boldsymbol{z}}^{(0)}) - f^\star \right) + 4L^2 \sum_{r=0}^{R} \Xi^{(r)} - \left( 1 - \frac{4L^2\beta^2\eta^2}{(1-\beta)^4} \right) \sum_{r=0}^{R} \mathbb{E} \left\| \frac{1}{N} \sum_{i=1}^{N} \nabla f_i(\boldsymbol{x}_i^{(r)}) \right\|^2$$
$$+ \frac{2L\sigma^2\eta}{N(1-\beta)}(R+1) + \frac{4L^2\beta^2\sigma^2\eta^2}{N(1-\beta)^4} R.$$

Using $\eta^2 \leq \frac{(1-\beta)^4}{8L^2}$ and $\beta < 1$, we obtain

$$\sum_{r=0}^{R} \mathbb{E} \left\| \nabla f(\bar{\boldsymbol{x}}^{(r)}) \right\|^2$$
$$\leq \frac{4(1-\beta)}{\eta} \left( f(\bar{\boldsymbol{z}}^{(0)}) - f^\star \right) + 4L^2 \sum_{r=0}^{R} \Xi^{(r)} + \frac{2L\sigma^2\eta}{N(1-\beta)}(R+1) + \frac{4L^2\beta^2\sigma^2\eta^2}{N(1-\beta)^4} R.$$

Using Lemma 19, we obtain

$$\frac{1}{2} \sum_{r=0}^{R} \mathbb{E} \left\| \nabla f(\bar{\boldsymbol{x}}^{(r)}) \right\|^2$$
$$\leq \frac{4(1-\beta)}{\eta} \left( f(\bar{\boldsymbol{z}}^{(0)}) - f^\star \right) + \frac{2L\sigma^2\eta}{N(1-\beta)}(R+1)$$
$$+ \frac{L^2}{(1-\beta)^3} \left( \frac{40t}{p^2} \left( 10 + \frac{29}{p} + \frac{864}{p^2} \right) + \frac{4\beta^2}{N(1-\beta)} \right) \sigma^2 \eta^2 (R+1).$$

This concludes the proof. $\qquad\square$

**Lemma 21.** *We define $t \in \mathbb{R}$ and $A \in \mathbb{R}$ as follows:*

$$t := \frac{2\beta^2 p}{1 - \beta^2} + 4, \quad A := \frac{648}{1 - \frac{p}{t} - \frac{2\beta^2}{1 + \beta^2}}.$$

*Then, it holds that*

$$\frac{(1 - \beta)^2 p^2}{16L\sqrt{\frac{7836\beta^2}{(1 - \beta^2)^3 p} + 282}} \leq \min\left\{\frac{1 - \beta}{4L}, \frac{p}{8L\sqrt{324 + \frac{2A\beta^2}{1 - \beta^2}}}, \frac{(1 - \beta)p^2}{2L\sqrt{t(5p^2 + 432)}} \frac{(1 - \beta)p}{8L\sqrt{5t}}, \frac{(1 - \beta)^2}{2\sqrt{2}L}\right\}.$$

*Proof.* Because $\sqrt{\frac{7836\beta^2}{(1 - \beta^2)^3 p} + 282} > 1$, $p \in (0, 1]$, and $\beta \in [0, 1)$, we have

$$\frac{(1 - \beta)^2 p^2}{16L\sqrt{\frac{7836\beta^2}{(1 - \beta^2)^3 p} + 282}} \leq \min\left\{\frac{1 - \beta}{4L}, \frac{(1 - \beta)^2}{2\sqrt{2}L}\right\}.$$

From $p \leq 1$, we have

$$t - 3 = \frac{2\beta^2 p}{1 - \beta^2} + 1 \geq \frac{1 + \beta^2}{1 - \beta^2}p = \frac{p}{1 - \frac{2\beta^2}{1 + \beta^2}}.$$

Then, we obtain

$$1 - \frac{p}{t} - \frac{2\beta^2}{1 + \beta^2} \geq \frac{p}{t - 3} - \frac{p}{t} = \frac{3p}{t(t - 3)} \geq \frac{3p}{t^2}.$$

Then, we obtain

$$A \leq \frac{216t^2}{p}.$$

Using the above inequality, we obtain

$$\frac{A\beta^2}{1 - \beta^2} + 162 \leq \frac{216\beta^2 t^2}{p(1 - \beta^2)} + 162.$$

From the definition of $t$, we obtain

$$\frac{A\beta^2}{1 - \beta^2} + 30t + 162 \leq \frac{216\beta^2 t^2}{p(1 - \beta^2)} + 30t + 162$$

$$= \frac{216\beta^2}{p(1 - \beta^2)}\left(\frac{2\beta^2 p}{1 - \beta^2} + 4\right)^2 + 30\left(\frac{2\beta^2 p}{1 - \beta^2} + 4\right) + 162$$

$$= \frac{216\beta^2}{p(1 - \beta^2)}\left(\frac{4\beta^4 p^2}{(1 - \beta^2)^2} + \frac{16\beta^2 p}{1 - \beta^2} + 16\right) + 30\left(\frac{2\beta^2 p}{1 - \beta^2} + 4\right) + 162$$

$$\leq \frac{7836\beta^2}{p(1 - \beta^2)^3} + 282,$$

where we use $\beta \in [0, 1)$ and $p \in (0, 1]$ in the last inequality. Then, we obtain

$$\frac{(1 - \beta)^2 p^2}{16L\sqrt{\frac{7836\beta^2}{(1 - \beta^2)^3 p} + 282}} \leq \frac{(1 - \beta)^2 p^2}{16L\sqrt{\frac{A\beta^2}{1 - \beta^2} + 30t + 162}}$$

$$\leq \min\left\{\frac{p}{8L\sqrt{324 + \frac{2A\beta^2}{1 - \beta^2}}}, \frac{(1 - \beta)p^2}{2L\sqrt{t(5p^2 + 432)}} \frac{(1 - \beta)p}{8L\sqrt{5t}},\right\}.$$

This concludes the proof. $\square$

**Lemma 22** (Convergence Rate for Non-convex Case). *Suppose that Assumptions 1, 2, 3, and 4 hold. Then, for any $R \geq 1$, there exists a step size $\eta$ such that it holds that*

$$\frac{1}{R} \sum_{r=0}^{R-1} \mathbb{E} \left\| \nabla f(\bar{\boldsymbol{x}}^{(r)}) \right\|^2$$
$$\leq \mathcal{O} \left( \sqrt{\frac{r_0 \sigma^2 L}{NR}} + \left( \frac{r_0^2 \sigma^2 L^2}{p^4 R^2 (1-\beta)} \left( 1 + \frac{p\beta^2}{1-\beta} \right) \right)^{\frac{1}{3}} + \frac{Lr_0}{(1-\beta)p^2 R} \sqrt{1 + \frac{\beta^2}{(1-\beta^2)^3 p}} \right),$$

*where $r_0 := f(\bar{\boldsymbol{x}}^{(0)}) - f^\star$.*

*Proof.* From Lemmas 20 and 21, if the step size $\eta$ satisfies the following:

$$\eta \leq \frac{(1-\beta)^2 p^2}{16L\sqrt{\frac{7836\beta^2}{(1-\beta^2)^3 p} + 282}},$$

then we have

$$\frac{1}{2(R+1)} \sum_{r=0}^{R} \mathbb{E} \left\| \nabla f(\bar{\boldsymbol{x}}^{(r)}) \right\|^2$$
$$\leq \frac{4}{\tilde{\eta}(R+1)} \left( f(\bar{\boldsymbol{z}}^{(0)}) - f^\star \right) + \frac{2L\sigma^2 \tilde{\eta}}{N} + \underbrace{\frac{L^2}{1-\beta} \left( \frac{40t}{p^2} \left( 10 + \frac{29}{p} + \frac{864}{p^2} \right) + \frac{4\beta^2}{N(1-\beta)} \right)}_{T} \sigma^2 \tilde{\eta}^2,$$

where we define $\tilde{\eta} := \frac{\eta}{1-\beta}$. Then, we can bound $T$ from above as follows:

$$T = \frac{L^2}{1-\beta} \left( \frac{40}{p^2} \left( \frac{2\beta^2 p}{1-\beta^2} + 4 \right) \left( 10 + \frac{29}{p} + \frac{864}{p^2} \right) + \frac{4\beta^2}{N(1-\beta)} \right)$$
$$\overset{p \in (0,1]}{\leq} \frac{L^2}{1-\beta} \left( \frac{36120}{p^4} \left( \frac{2\beta^2 p}{1-\beta^2} + 4 \right) + \frac{4\beta^2}{N(1-\beta)} \right)$$
$$\overset{p \in (0,1], \beta \in [0,1)}{\leq} \frac{36120 L^2}{(1-\beta)p^4} \left( \frac{3\beta^2 p}{1-\beta} + 4 \right).$$

Then, we obtain

$$\frac{1}{2(R+1)} \sum_{r=0}^{R} \mathbb{E} \left\| \nabla f(\bar{\boldsymbol{x}}^{(r)}) \right\|^2$$
$$\leq \frac{4}{\tilde{\eta}(R+1)} \left( f(\bar{\boldsymbol{z}}^{(0)}) - f^\star \right) + \frac{2L\sigma^2 \tilde{\eta}}{N} + \frac{36120 L^2}{(1-\beta)p^4} \left( \frac{3\beta^2 p}{1-\beta} + 4 \right) \sigma^2 \tilde{\eta}^2.$$

Using Lemma 17 in the previous work (Koloskova et al., 2020), we obtain the statement. □

# E   Hyperparameter Settings

Tables 9, 10, 11, 12, and 13 list the hyperparameter settings for each dataset. We evaluated the performance of each comparison method for different step sizes and selected the step size that achieved the highest accuracy on the validation dataset.

Table 9: Experimental settings for FashionMNIST.

| | |
|---|---|
| Neural network architecture | LeNet (LeCun et al., 1998) |
| Normalization | Group normalization (Wu & He, 2018) |
| Step size | $\{0.005, 0.001, 0.0005\}$ |
| L2 penalty | 0.001 |
| Batch size | 100 |
| Data augmentation | RandomCrop |
| Total number of epochs | 500 |

Table 10: Experimental settings for SVHN.

| | |
|---|---|
| Neural network architecture | LeNet (LeCun et al., 1998) |
| Normalization | Group normalization (Wu & He, 2018) |
| Step size | $\{0.005, 0.001, 0.0005\}$ |
| L2 penalty | 0.001 |
| Batch size | 100 |
| Data augmentation | RandomCrop |
| Total number of epochs | 500 |

Table 11: Experimental settings for CIFAR-10.

| | |
|---|---|
| Neural network architecture | LeNet (LeCun et al., 1998) |
| Normalization | Group normalization (Wu & He, 2018) |
| Step size | $\{0.005, 0.001, 0.0005\}$ |
| L2 penalty | 0.001 |
| Batch size | 100 |
| Data augmentation | RandomCrop, RandomHorizontalFlip |
| Total number of epochs | 500 |

Table 12: Experimental settings for CIFAR-10 with VGG-11.

| | |
|---|---|
| Neural network architecture | VGG-11 (Simonyan & Zisserman, 2015) |
| Normalization | Group normalization (Wu & He, 2018) |
| Step size | $\{0.5, 0.1, 0.05, 0.01, 0.005\}$ |
| Step size decay | /10 at epoch 500 and 750. |
| L2 penalty | 0.001 |
| Batch size | 100 |
| Data augmentation | RandomCrop, RandomHorizontalFlip, RandomErasing |
| Total number of epochs | 1000 |

Table 13: Experimental settings for CIFAR-10 with ResNet-34.

| | |
|---|---|
| Neural network architecture | ResNet-34 (He et al., 2016) |
| Normalization | Group normalization (Wu & He, 2018) |
| Step size | $\{0.5, 0.1, 0.05, 0.01, 0.005\}$ |
| Step size decay | /10 at epoch 375 and 563. |
| L2 penalty | 0.001 |
| Batch size | 100 |
| Data augmentation | RandomCrop, RandomHorizontalFlip, RandomErasing |
| Total number of epochs | 750 |

