# OpenReview forum: "Momentum Tracking: Momentum Acceleration for Decentralized Deep Learning on Heterogeneous Data"
_TMLR — Accepted by TMLR_

### Review · Reviewer_KSER · 2023-06-08

**Summary Of Contributions:**


This work considers a stochastic decentralized optimization problem. In particular, the paper focuses on a combination of two standard optimization techniques: gradient tracking (for decentralized optimization over a network) and Polyak’s momentum. Such combination has been already proposed previously in the literature (as acknowledged by the authors). The main theoretical contribution of this work is the refined analysis of such combination, which allows to remove a strong gradient similarity assumption (present in the previous works combining gradient tracking and momentum). The obtained rates of convergence recover those for non-momentum variant of gradient tracking by Koloskova et al 2021. The paper also offers an extensive numerical comparison of this method with other existing alternatives on a number of deep learning tasks.

My main concern about this work is the lack of comparison to existing works in decentralized optimization and the lack of technical novelty. In particular, it seems questionable to me that removing gradient similarity/bounded gradients assumption is a serious challenge in this setting given that such assumption is not present in Koloskova et al 2021 (for gradient tracking without momentum) and it was removed in (Liu  et al 2020) for momentum method (in the centralized setting).


**Audience:**

Yes

**Claims And Evidence:**

Yes

**Requested Changes:**

Mentioned above.

**Strengths And Weaknesses:**

**Strengths:**

1. The work extends the analysis of (Koloskova et al 2021) to incorporate the momentum step based on analysis of momentum in (Yu et al 2019) and (Liu et al 2020). This extension is done (essentially) for free meaning that the convergence rate is worse only by a numerical constant compared to a non-momentum version of Koloskova et al 2021.
2. The paper provides a brief proof sketch of their result in the appendix. The proof of the theorem is well written and the (proof) sketch helps to understand it better. The statement of the main theorem makes sense to me. I have also checked a few key lemma in the appendix, which seem to be correct.
3. The paper includes an extensive experimental comparison of their momentum tracking method to those known in the literature. The key message of experiments is clear: momentum helps in practice and the proposed scheme is robust to the increase in the level of heterogeneity of data.


**Weaknesses:**

1. The analysis in this work is heavily based on the one by (Koloskova et al, 2020) and the authors subsequently compare to this early result. However, it is unclear to me why the authors chose to base on this analysis given that there were seemingly a number of improvements and refinements of this analysis since then. In particular, the closely related works (Lu et al, 2021) and (Yuan et al, 2022b) are completely ignored and not even cited. These works seem to have better convergence rates, and the first one in particular does not require gradient similarity assumption. Other related works are missing, e.g., (Batiste Le Bars et al, 2022).
2. The literature review related to Polyak’s momentum is also far from being complete. For instance, (Defazio, 2021), (Karimireddy et al, 2021), (Cutkosky et al, 2020, 2021).
3. The difference of algorithms and analysis with those in (Lin et al, 2021) and (Yuan et al, 2020) are not discussed (QG-SGDm and DecentLaM). These works have similar convergence guarantees in the exact same setting, but require stronger data similarity assumptions. Is it the limitation of their analysis or the methods? Are their methods different from the one proposed in this work?

**The writing and presentation** of the paper can be significantly improved.
1. There are a number of repetitions of sentences or even entire paragraphs throughout the work. For example, this paragraph is repeated twice (in Section 1 and 2):
“In the convex optimization literature, Xin & Khan (2020) and Carnevale et al. (2022) proposed combining Gradient Tracking with momentum or Adam and analyzed the convergence rates...”

This sentence is repeated twice (in section 1 and 2): “When the data distributions held by each node are statistically homogeneous, DSGDm works well and can improve the performance as well as SGDm.”

This sentence is almost identical to the first sentence of the abstract: “One of the key components for improving the performance of neural networks is SGD with momentum acceleration (SGDm).”

I also noticed a number of unnecessary repetitions in Section 4: “when the data distributions are homogeneous, DSGDm, QG-DSGDm, DecentLaM, and Momen- tum Tracking are comparable and outperform DSGD and Gradient Tracking.”

Same for this sentence: “ a method in which the update rule of the momentum in DSGDm is modified to be more robust to data heterogeneity than DSGDm”. Moreover, it is not clarified how the method “is modified to be robust to data heterogeneity”.

2. Some word choices are not very precise/rigorous: For example: “When the data distributions held by each node (i.e., the server) are statistically homogeneous, DSGDm works well and can improve the performance as well as SGDm.” Improve in which sense? Theoretically/practically? Why the node is the same thing as the server?
On page 4, authors claim that the setting they consider is “standard DL setting”. This is not precise given that the loss in most of DL architectures (including those considered in this paper) is not smooth.

3. The authors repeatedly use the word “acceleration” in the paper, which seems to be an over claim since there is no proven acceleration (in any sense) of the proposed method compared to (non momentum variant) gradient tracking method. Moreover, I would refrain from using this word to avoid confusion with acceleration in the sense of Nesterov since there is a large body of works trying to prove that Polyak’s momentum has accelerated rate compared to gradient descent beyond strongly convex quadratics. But in many important cases this question still remains an open problem.


Yucheng Lu and Christopher De Sa. Optimal complexity in decentralized training. In Interna- tional Conference on Machine Learning (ICML), pages 7111–7123. PMLR, 2021.

Kun Yuan, Xinmeng Huang, Yiming Chen, Xiaohan Zhang, Yingya Zhang, Pan Pan . Revisiting Optimal Convergence Rate for Smooth and Non-convex Stochastic Decentralized Optimization. 2022b.

Batiste Le Bars, Aurélien Bellet, Marc Tommasi, Erick Lavoie, Anne-Marie Kermarrec . Refined Convergence and Topology Learning for Decentralized SGD with Heterogeneous Data. 2022.

Aaron Defazio. Momentum via primal averaging: Theoretical insights and learning rate schedules for non-convex optimization. arXiv preprint arXiv:2010.00406, 2021.

Sai Praneeth Karimireddy, Martin Jaggi, Satyen Kale, Mehryar Mohri, Sashank J. Reddi, Sebastian U. Stich, and Ananda Theertha Suresh. Mime: Mimicking centralized stochastic algorithms in federated learning. arXiv preprint arXiv:2008.03606, 2021.

Ashok Cutkosky and Harsh Mehta. Momentum improves normalized SGD. In International Conference on Machine Learning, page 2260–2268. PMLR, 2020.

Ashok Cutkosky and Harsh Mehta. High-probability bounds for non-convex stochastic optimization with heavy tails. pages 4883–4895, 2021.

---

> ### Author Response · Authors · 2023-07-25
> **Response to Reviewer KSER (1/2)**
>
> > The analysis in this work is heavily based on the one by (Koloskova et al, 2020) and the authors subsequently
> compare to this early result. However, it is unclear to me why the authors chose to base on this analysis given that there
> were seemingly a number of improvements and refinements of this analysis since then. In particular, the closely related
> works (Lu et al, 2021) and (Yuan et al, 2022b) are completely ignored and not even cited. These works seem to have
> better convergence rates, and the first one in particular does not require gradient similarity assumption. Other related
> works are missing, e.g., (Batiste Le Bars et al, 2022).
>
> Yuan et al. (2022) and Lu and De Sa (2021) provided the lower bounds of the convergence rate of decentralized learning
> methods and proposed novel methods whose convergence rates are almost the same as these lower bounds. To achieve
> these lower bounds, their proposed methods use algorithmic techniques such as gradient accumulation and multiple
> gossip averaging, which are not used in Momentum Tracking, and their proof strategies are designed to analyze these
> techniques. Therefore, their proof strategies are not applicable to the convergence analysis of Momentum Tracking, and
> we analyzed it based on the analysis of (Koloskova et al., 2020).
>
> Following the reviewer’s suggestion, we cited these works in Secs. 1 and 3.1 in the revised manuscript
>
> > The literature review related to Polyak’s momentum is also far from being complete. For instance, (Defazio, 2021), (Karimireddy et al, 2021), (Cutkosky et al, 2020, 2021).
>
> We cited these works in the revised manuscript in Sec. 1.
>
> > The difference of algorithms and analysis with those in (Lin et al, 2021) and (Yuan et al, 2020) are not discussed
> (QG-SGDm and DecentLaM). These works have similar convergence guarantees in the exact same setting, but require
> stronger data similarity assumptions. Is it the limitation of their analysis or the methods? Are their methods different
> from the one proposed in this work?
>
> In Figs. 8 and 9, we compared Momentum Tracking with QG-DSGDm and DecentLaM with convex optimization
> problems. The results indicate that the convergence rates of QG-DSGDm and DecentLaM become slow when data heterogeneity becomes large, whereas the convergence rate of Momentum Tracking is independent of data heterogeneity. Therefore, we can conclude that it is not the limitation of convergence analysis but the limitation of methods that the convergence rates of QG-DSGDm and DecentLaM provided by Lin et al. (2021) and Yuan et al. (2021) depend on data heterogeneity.

---

> ### Author Response · Authors · 2023-07-25
> **Response to Reviewer KSER (2/2)**
>
> > There are a number of repetitions of sentences or even entire paragraphs throughout the work. [...]
>
> Following the reviewer’s suggestion, we have omitted the paragraph and sentences in Secs. 2.3 and 4.3.
>
> > Some word choices are not very precise/rigorous: For example: “When the data distributions held by each node (i.e.,
> the server) are statistically homogeneous, DSGDm works well and can improve the performance as well as SGDm.”
> Improve in which sense? Theoretically/practically? [...]
>
> We have rewritten this sentence in Sec. 2.2 to say that DSGDm can experimentally work well when the data distributions are homogeneous.
>
> > Some word choices are not very precise/rigorous: [...] On page 4, authors claim that the setting they consider is
> “standard DL setting”. This is not precise given that the loss in most of DL architectures (including those considered in
> this paper) is not smooth.
>
> We have rephrased the term “standard deep learning setting” with the term “setting where the objective function is
> non-convex and the stochastic gradient is used.”
>
> > [...] “ a method in which the update rule of the momentum in DSGDm is modified to be more robust to data heterogeneity than DSGDm”. Moreover, it is not clarified how the method “is modified to be robust to data heterogeneity”.
>
> We revised the sentence in Sec. 2.2 to explain the ideas of QG-DSGDm and DecentLaM.
>
> > The authors repeatedly use the word “acceleration” in the paper, which seems to be an over claim since there is no proven acceleration (in any sense) of the proposed method compared to (non momentum variant) gradient tracking
> method. Moreover, I would refrain from using this word to avoid confusion with acceleration in the sense of Nesterov
> since there is a large body of works trying to prove that Polyak’s momentum has accelerated rate compared to gradient
> descent beyond strongly convex quadratics. But in many important cases this question still remains an open problem.
>
> Following the reviewer’s suggestion, we have replaced the term “momentum acceleration” with the term “momentum” in the revised manuscript.
>
> ### References
> Lu, Y. and De Sa, C. (2021). Optimal complexity in decentralized training. In ICML.
>
> Yuan et al., (2022). Revisiting optimal convergence rate for smooth and non-convex stochastic decentralized optimization. In NeurIPS.
>
> Koloskova et al., (2020). A unified theory of decentralized SGD with changing topology and local updates. In ICML.

---

### Review · Reviewer_8a5E · 2023-06-10

**Summary Of Contributions:**

This work proposed Momentum Tracking, a decentralized learning method with momentum acceleration whose convergence rate is proven to be independent of the data heterogeneity in the standard deep learning setting (stochastic, smooth and nonconvex objective).

**Audience:**

Yes

**Broader Impact Concerns:**

NA, mainly theoretic work.

**Claims And Evidence:**

Yes

**Requested Changes:**

Since ABm (Xin & Khan, 2020) and GTAdam (Carnevale et al., 2022) already proposes the idea of combining Gradient Tracking with momentum or Adam, is there any algorithmic modification to this idea proposed in this work? Some algorithmic comparison between Momentum Tracking and ABm/GTAdam is necessary.

**Strengths And Weaknesses:**

Strengths:
- This paper is clearly written. The comparison with existing methods and the literature review are solid.
- Thorough discussion about the theorem, such as the initialization requirement, tuning of $\beta$, comparison with existing theoretic results, which is appreciated and clearly demonstrates the theoretic contribution.
- It is always nice to see a theory-inspired algorithmic modification to be effective and have practical benefits, as shown in the synthetic and real world experiments.

Weaknesses:
- The contribution is kind of incremental. Since ABm (Xin & Khan, 2020) and GTAdam (Carnevale et al., 2022) already proposes the idea of combining Gradient Tracking with momentum or Adam, is there any algorithmic modification to this idea proposed in this work? Some algorithmic comparison between Momentum Tracking and ABm/GTAdam is necessary. I'd like to see some algorithmic novelty. However, it is totally fine if the contribution is mainly theoretic, just might be of smaller practical interest in this case.

---

> ### Author Response · Authors · 2023-07-25
> **Response to Reviewer 8a5E**
>
> > Since ABm (Xin \& Khan, 2020) and GTAdam (Carnevale et al., 2022) already proposes the idea of combining Gradient Tracking with momentum or Adam, is there any algorithmic modification to this idea proposed in this work? Some algorithmic comparison between Momentum Tracking and ABm/GTAdam is necessary.
>
> We compared Momentum Tracking with ABm (Xin \& Khan, 2020) and GTAdam(Carnevale et al., 2022) with CIFAR-10 and LeNet, showing the results in the following table.
> The update rules of Momentum Tracking are slightly different from that of ABm and GTAdam,
> but the results indicate that they can achieve almost the same accuracy.
>
> |                   | 10-class        | 4-class         |
> |-------------------|-----------------|-----------------|
> | Momentum Tracking | $72.9 \pm 0.59$  | $70.7 \pm 1.38$ |
> | ABm               | $71.0 \pm 0.20$ | $71.1 \pm 0.20$ |
> | GTAdam            | $71.4 \pm 0.56$ | $67.7 \pm 3.20$ |
>
> However, as the reviewer mentioned in the review,
> the main contribution of our work is that we proposed the method with momentum whose convergence rate is proven to be independent of data heterogeneity in the non-convex and stochastic setting,
> and we would like to emphasize that Xin \& Khan, 2020 and Carnevale et al., 2022 provided the convergence analysis only on the strongly convex setting.
> We have added these results and discussion in Sec. C.5.

---

### Review · Reviewer_WFz3 · 2023-07-12

**Summary Of Contributions:**

The authors introduce a novel approach to decentralized learning, incorporating momentum acceleration, that they've called 'Momentum Tracking'.
Under standard assumptions for decentralized optimization (see Assumptions 1-5 below),
they provided a convergence rate for the smooth non-convex regime of the form
$\mathcal{O}\left(\sqrt{\frac{r_0 \sigma^2 L}{N R}}+\left(\frac{r_0^2 \sigma^2 L^2}{p^4 R^2(1-\beta)}\left(1+\frac{p \beta^2}{1-\beta}\right)\right)^{\frac{1}{3}}+\frac{L r_0}{(1-\beta) p^2 R} \sqrt{1+\frac{\beta^2}{\left(1-\beta^2\right)^3 p}}\right)$
, which is independent of the data heterogeneity parameter $\zeta^2$ and holds for any momentum coefficient $\beta$ within the range of 0 to 1.

**Assumption 1**. There exists a constant $f^{\star}>-\infty$ that satisfies $f(\boldsymbol{x}) \geq f^{\star}$ for all $\boldsymbol{x} \in \mathbb{R}^d$.

**Assumption 2**. There exists a constant $p \in(0,1]$ that satisfies for all $\boldsymbol{x}_1, \cdots, \boldsymbol{x}_N \in \mathbb{R}^d$,
$$
\|\boldsymbol{X} \boldsymbol{W}-\overline{\boldsymbol{X}}\|_F^2 \leq(1-p)\|\boldsymbol{X}-\overline{\boldsymbol{X}}\|_F^2
$$
where $\boldsymbol{X}:=\left(\boldsymbol{x}_1, \cdots, \boldsymbol{x}_N\right) \in \mathbb{R}^{d \times N}$ and $\overline{\boldsymbol{X}}:=\frac{1}{N} \boldsymbol{X} \mathbf{1 1}{ }^{\top}$.

**Assumption 3**. There exists a constant $L>0$ that satisfies for all $i \in V$ and $\boldsymbol{x}, \boldsymbol{y} \in \mathbb{R}^d$,
$$
\left\|\nabla f_i(\boldsymbol{x})-\nabla f_i(\boldsymbol{y})\right\| \leq L\|\boldsymbol{x}-\boldsymbol{y}\|
$$

**Assumption 4**. There exists a constant $\sigma^2$ that satisfies for all $i \in V$ and $\boldsymbol{x_i} \in \mathbb{R}^d$,
$$
\mathbb{E}_{\xi_i \sim \mathcal{D}_i}\left\|\nabla F_i\left(\boldsymbol{x}_i ; \xi_i\right)-\nabla f_i\left(\boldsymbol{x}_i\right)\right\|^2 \leq \sigma^2
$$

**Assumption 5**. There exists a constant $\zeta^2$ that satisfies for all $\boldsymbol{x} \in \mathbb{R}^d$,
$$
\frac{1}{N} \sum_{i=1}^N\left\|\nabla f_i(\boldsymbol{x})-\nabla f(\boldsymbol{x})\right\|^2 \leq \zeta^2
$$

Comparison with existing methods, as shown in their study (Table 1), further solidifies the author's contributions: this is the first method that addresses Data-Heterogeneity, Momentum, Stochastic estimators, and Non-Convexity among considered baselines.

In terms of practical application, the authors provide empirical evidence to support their theoretical results, showing that Momentum Tracking displays greater robustness to data heterogeneity compared to current decentralized learning methods with momentum acceleration.

**Audience:**

Yes

**Broader Impact Concerns:**

This work is primarily theoretical in nature. Hence, there is no identifiable potential for negative societal impact arising from this work.

**Claims And Evidence:**

Yes

**Requested Changes:**

1) I recommend the authors revisit their choice of line colors in the figures. For instance, in Figures 1 and 2, the differentiation between black and dark blue colors, representing DecentLaL and Momentum Tracking respectively, might pose a challenge for some viewers.
2) For similar considerations, I would recommend that the authors employ marked lines in their figures, each representing a specific method. This would significantly assist individuals with color vision deficiency.
3) Further, the reasons behind selecting the ring topology in the experimental section of the paper's main part could benefit from a more explicit explanation.

**Strengths And Weaknesses:**

**Strengths:**
1) The whole paper is clearly written, with its principal claims effectively outlined, making the text easy to read;
2) The literature review contains pertinent papers related to the topic. The authors did a great job providing a comprehensive literature overview, perfectly fitting their paper to the study of decentralized methods;
3) The authors propose a novel algorithm suitable for non-convex objectives that makes it possible to combine a momentum trick (which is a standard one for training deep learning modes) and gradient tracking - a famous approach to removing gradient heterogeneity from the iteration complexity for decentralized methods.
4) Besides the solid theoretical results, the authors proposed an extensive set of experiments comparing the proposed method against multiple baselines with\without momentum and with\without gradient tracking. In all cases (except one (a comparison between "Momentum Tracking" and "RelaySum") ), the proposed method for a homogeneous setting shows compatible results. In contrast, for a heterogeneous one, in terms of the accuracy metric, the proposed "Momentum Tracking" considerably outperforms existing baselines.

**Weaknesses:**
1) Some potential updates can be considered in the revised version of this Paper. See section **Requested changes**;
 2) When $\beta=0$ (i.e., when momentum tracking reduces to gradient tracking), resulting complexity
   $\mathcal{O}\left(\sqrt{\frac{r_0 \sigma^2 L}{N R}}+\left(\frac{r_0 \sigma L}{p^2 R}\right)^{\frac{2}{3}}+\frac{L r_0}{p^2 R}\right)$
   is worse than the one
   $\tilde{\mathcal{O}}\left(\sqrt{\frac{r_0 \sigma^2 L}{N R}}+\left(\frac{r_0 \sigma L}{(\sqrt{p} c+p \sqrt{N}) R}\right)^{\frac{2}{3}}+\frac{L\left(r_0+L \zeta_0^2\right)}{p c R}\right)$
   from [1] by Koloskova et al. However, Koloskova et al. obtained this complexity under a stricter condition
 $R>\frac{2}{p} \log \left(\frac{50}{p}(1+\right.$ $\left.\log \frac{1}{p}\right)$ )
whereas the former complexity works for any $R$.

[1] Anastasia Koloskova, Tao Lin, and Sebastian U Stich. An improved analysis of gradient tracking for decentralized machine learning. In Advances in Neural Information Processing Systems, 2021.

---

> ### Author Response · Authors · 2023-07-25
> **Response to Reviewer WFz3**
>
> > I recommend the authors revisit their choice of line colors in the figures. [...]
>
> Following the reviewer's suggestion, we changed the colors in the figures.
>
> > For similar considerations, I would recommend that the authors employ marked lines in their figures, each representing a specific method. [...]
>
> Following the reviewer's suggestion, we changed the figures.
>
> > Further, the reasons behind selecting the ring topology in the experimental section of the paper's main part could benefit from a more explicit explanation.
>
> Communication efficiency is one of the most important factors in decentralized learning and is determined by the
> maximum degree of the underlying network topology (Neglia et al., 2019; Wang et al., 2019; Ying et al., 2021). Because
> the maximum degree of the ring is only two, the ring is communication efficient topology and is widely used in
> decentralized learning, e.g., (Lian et al., 2017; Tang et al., 2018). Following these prior works, we evaluated Momentum
> Tracking with the ring in Sec. 4. We added these explanations in Sec. 4.1 in the revised manuscript. Additionally, we
> evaluated Momentum Tracking with various network topologies in Sec. C, showing that Momentum Tracking can
> outperform the existing methods in all network topologies.
>
> ### References
> Lian et al., (2017). Can decentralized algorithms outperform
> centralized algorithms? a case study for decentralized parallel stochastic gradient descent. In Advances in Neural
> Information Processing Systems.
>
> Neglia et al., (2019). The role of network topology for distributed machine
> learning. In IEEE Conference on Computer Communications.
>
> Tang et al., (2018). d2: Decentralized training over decentralized data. In
> International Conference on Machine Learning.
>
> Wang et al., (2019). Matcha: Speeding up decentralized sgd via matching
> decomposition sampling. In Indian Control Conference.
>
> Ying et al., (2021). Exponential graph is provably efficient for
> decentralized deep training. In Advances in Neural Information Processing Systems.

---

### Author Response · Authors · 2023-07-25
**Response to All Reviewers**

We thank all reviewers for their constructive comments. In the revised manuscript, revised or added contents are shown in blue. Then, omitted contents are displayed in the strikethrough style. We will remove these contents in the camera-ready version.

---

### Decision · Action_Editors · 2023-08-28

**Recommendation:** Accept with minor revision

**Comment:**

Several suggestions were made by Reviewers KSER and WFz3, and some of them were not taken into account yet. I propose a minor revision where this is taken int account.

Further, the authors study decentralized nonconvex stochastic optimization, and their aim is to obtain a rate independent of any data heterogeneity term. However, methods that do that exist already, e.g., Koloskova et al (2021), which the authors cite and compare to, but also later works that improve previous results, such as

Li et al, BEER: Fast O(1/T ) Rate for Decentralized Nonconvex Optimization with Communication Compression, NeurIPS 22,

which the authors do not cite nor compare to. The above work does not use momentum, but momentum is not needed from a theoretical point of view as it does not improve the rate. I'd welcome a comparison with the above work.

**Audience:**

ML researchers interested in Polyak momentum and decentralized optimization.

**Claims And Evidence:**

Yes. The key claims of the paper are supported with theorems and proofs, and with numerical experiments.